# ChaosEater: Fully Automating Chaos Engineering with Large Language Models

## Abstract

Chaos Engineering (CE) is an engineering technique aimed at improving the resiliency of distributed systems. It involves artificially injecting specific failures into a distributed system and observing its behavior in response. Based on the observation, the system can be proactively improved to handle those failures. Recent CE tools realize the automated execution of predefined CE experiments. However, defining these experiments and reconfiguring the system after the experiments still remain manual. To reduce the costs of the manual operations, we propose ChaosEater, a *system* for automating the entire CE operations with Large Language Models (LLMs). It pre-defines the general flow according to the systematic CE cycle and assigns subdivided operations within the flow to LLMs. We assume systems based on Infrastructure as Code (IaC), wherein the system configurations and artificial failures are managed through code. Hence, the LLMs' operations in our *system* correspond to software engineering tasks, including requirement definition, code generation and debugging, and testing. We validate our *system* through case studies on both small and large systems. The results demonstrate that our *system* significantly reduces both time and monetary costs while completing a reasonable CE cycle. Our code is available in the Supplementary Material.

## 1 Introduction

Modern software-based services, such as streaming, e-commerce, and conversational AI platforms, are implemented as distributed systems, where each service is divided into smaller services according to specific functionalities. These small services (i.e., functions), along with the communication network that connects them, constitute the entire service. This design, known as microservice architecture (Bucchiarone et al., 2020), enables scalable and continuous deployment while supporting the integration of heterogeneous technologies. On the other hand, the complex dependencies among small services can lead to unexpected, chaotic behavior in the entire system from even minor failures. However, proactively predicting and addressing such complex behavior is challenging.

To address this and improve the resiliency of distributed systems, numerous organizations, including Netflix, Amazon, and Microsoft, have recently adopted Chaos Engineering (CE) (Basiri et al., 2016; 2019). Its concept is that *rather than predicting the chaotic behavior, let's observe it directly by artificially injecting the failures into the system*. Based on the observation, we can proactively rebuild a new system that is resilient to the assumed failures. Systematically, CE cycles through four phases for a system:

1. **Hypothesis**: Define steady states (i.e., normal behavior) of the system and injected failures. Then, make a hypothesis that *the steady states are maintained in the system even when the failures occur*.
2. **(Chaos) Experiment**: Inject the failures into the system while logging the system's response behavior.
3. **Analysis**: Analyze the logged data and check if the hypothesis is satisfied. If so, this CE cycle is finished here. If not, move to (4).
4. **Improvement**: Reconfigure the system to satisfy the hypothesis. The reconfigured system is tested again in (2) and (3), i.e., repeat (2) to (4) until the hypothesis is satisfied.

In recent years, several CE tools (Netflix, 2012; Amazon Web Services, 2021; Chaos Mesh, 2021; Microsoft, 2023) have advanced the automation of chaos-experiment execution. Moreover, monitor-

ing tools (Prometheus, 2012; Grafana Labs, 2021) enable automating metric collection, aggregation, and threshold-based testing during chaos experiments. Hence, the *experiment* and *analysis* phases have been mostly automated. However, defining a hypothesis in the *hypothesis* phase, planning a chaos experiment to test the hypothesis in the *experiment* phase, and reconfiguring the system in the *improvement* phase still remain manual. These manual operations require a complex set of skills, including domain knowledge in networking and CE, the ability to interpret system configurations, logs, and error messages, as well as creative problem-solving for requirement definition, planning, and system reconfiguration. Consequently, while the costs of these operations remain high, their automation has not been achieved yet with existing algorithmic approaches.

We believe that Large Language Models (LLMs) are the key to overcoming this challenge. LLMs have recently shown promising capabilities across a wide range of tasks, including natural language processing, coding, and operations for networking (Zhao et al., 2023; Jiang et al., 2024; Ahmed et al., 2024; Piovesan et al., 2024). In more recent years, LLMs have also provided promising performance on software engineering (SE) benchmarks (Yang et al., 2024; Cognition Labs, 2024). In the context of software-based systems configured by the Infrastructure as Code (IaC) paradigm, where the system configurations and artificial failures are managed through code, CE operations can be regarded as SE tasks. The *hypothesis* phase corresponds to a requirement definition to determine the resilience required for the system. In the *experiment* phase, planning an experiment corresponds to the design of testing, and running the experiment requires coding. The *analysis* corresponds to the verification of the tests. Lastly, the *improvevment* phase corresponds to code debugging. Considering their general capabilities, domain knowledge in networking, and potential in SE, it is also expected that the automation of the entire CE cycle can be achieved with LLMs.

Here, we propose CHAOSEATER, a *system* for automating the entire CE cycle with LLMs. It pre-defines the general flow according to the systematic CE cycle and assigns subdivided CE operations within this flow to LLMs. The flow (i.e., CE cycle) then progresses by sequentially performing subdivided operations using the assigned LLMs, while processing data inputs and outputs through rule-based algorithms. Our *system* assumes CE for IaC-based systems, specifically Kubernetes (K8s) (Kubernetes, 2014) systems. In this paper, we present the flow design, rule-based algorithms, CE operations performed by LLMs, and the integration of the latter two to achieve the designed flow. In evaluation, we validate our *system* through case studies on both a small system and a large system. The results demonstrate that our *system* significantly reduces both time and monetary costs while completing a reasonable CE cycle. Lastly, we discuss the broader impacts, limitations, and future directions of our *system* based on this study.

The main contributions of this paper are organized as follows:

- We are the first to propose a *system* for automating the entire CE cycle with LLMs, which reduces time and monetary costs in a CE cycle. This proposal would be a starting point towards the full automation of system resilience improvement.
- We publicly release all code for our *system*. This release provides development practices for constructing complex systems that combine LLMs and rule-based algorithms.
- We validate our *system* through case studies and discuss its broader impacts, limitations, and future directions. The results demonstrate the new potential of LLMs in CE, while our discussion provides insights for advancing the full automation of CE.

## 2 PROPOSED SYSTEM: CHAOSEATER

In this section, we describe the technical details of CHAOSEATER. Figure 1 shows its simplified system diagram. It takes as input instructions for the CE cycle (optional) and a folder containing K8s manifests (Kubernetes, 2014) and a Skaffold configuration file (Google, 2019). In short, K8s manifests are system configuration files that define the resources (i.e., small services) that constitute a system, while a Skaffold configuration file defines the process to automatically deploy those resources in a K8s cluster. It then conducts a CE cycle for those inputs through five divided phases: pre-processing, *hypothesis*, *experiment*, *analysis*, *improvement*, and post-processing phases. Finally, it outputs a summary of the completed CE cycle and a modified folder containing K8s manifests that have been modified to satisfy the hypothesis defined in the *hypothesis* phase and their Skaffold configuration file.

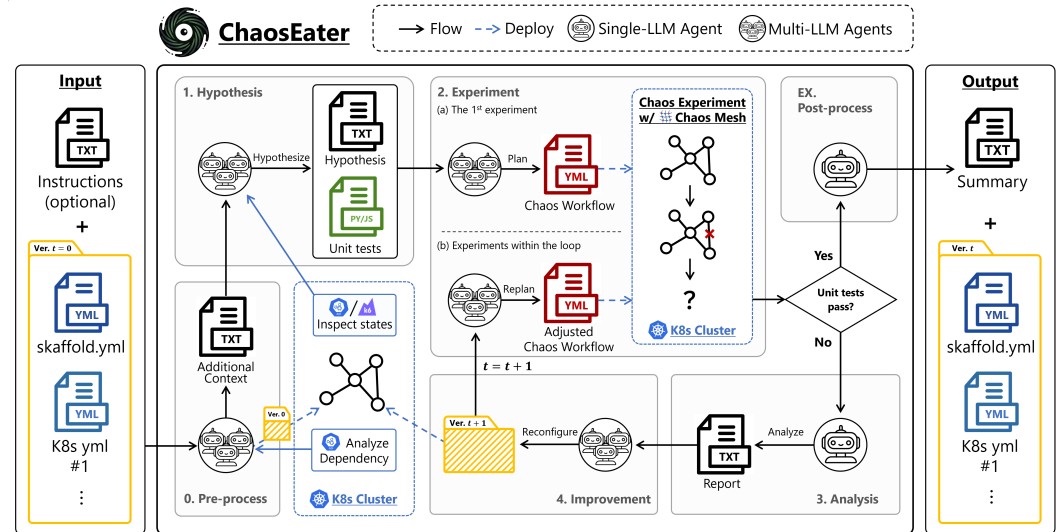

Figure 1: A simplified system diagram of CHAOSEATER and its input and output. CHAOSEATER autonomously completes the systematic CE cycle using internal agents and IaC tools. Note that only the representative inputs and outputs of agents are illustrated within the diagram. The two K8s clusters within the diagram refer to the same one.

To ensure that the LLMs perform as intended, our *system* fixes the general flow according to the systematic CE cycle. It then guides LLMs by assigning them subdivided CE operations within this flow. Hereafter, we define each LLM assigned a CE operation as an agent. Our *system* prepares prompt templates for each agent,[1] which include placeholders that can be dynamically filled with text. Therefore, once a user inputs the data, prompts for each agent are dynamically generated according to that data, and the internal agents autonomously complete the flow (i.e., CE cycle). To facilitate data processing within our *system*, all agents output JSON data. This is achieved by instructing agents in their input prompts to output text in JSON format, and then parsing the output text as JSON data. Our *system* uses the JSON output instruction and parser of LangChain (LangChain, 2023). Our *system* has 27 agents, 21 system prompts, 26 user prompts, and two AI prompts.[2] In the following sections, we describe the details of our *system*'s internal process from input to output, breaking it down into the five phases. See Appendix B for all our *system*'s prompt templates.

## 2.1 PHASE 0: PRE-PROCESSING

Given user inputs, our *system* first deploys the user's system to the K8s cluster by running the Skaffold configuration file. Then, the agents sequentially process the user inputs as follows:

1. Summarize each of the input K8s manifests separately.
2. Identify potential issues for resiliency and redundancy in the K8s manifests.
3. Assume a possible application of the K8s manifests.
4. Summarize user instructions for the CE cycle if provided. At the same time, filter out suspicious prompts, e.g., jailbreak prompts.

This phase is for deploying the user's system and explicitly filling in the implicit context of the user's input. In the subsequent phases, this added context will also be provided as input.

---

[1]Instead of simply appending previous data and agent outputs to the conversation history to create the next agent's prompt, we create a new conversation for each agent every time and embed the organized previous data and agent outputs within it. However, the verification loop, which will be discussed later, is an exception.

[2]As our *system* uses chat models, prompts with three different roles—system, human, and AI—are required.

## 2.2 PHASE 1: HYPOTHESIS

The *hypothesis* phase defines the system's resiliency for an assumed failure scenario. Following the principles of CE (Basiri et al., 2016), our *system* first defines steady states and then defines failure injections.

**Steady-state definition**  Steady states are the expected, normal behaviors of a system. Each steady state is defined by a pair of a state value and a threshold, and a steady state is considered satisfied when the state value meets the threshold. Therefore, the state values must be measurable outputs of the system, such as the number of active resources, error rates, and response time. Given the pre-processed user inputs, the agents define steady states as follows:

1. Select measurable states critical to maintaining the system's application. If any weak configurations are identified from the K8s manifests, their related states are preferentially selected.
2. Select tools to inspect the states. K8s API and k6 (Grafana Labs, 2021) are supported. Then, write the corresponding inspection scripts and inspect the current (normal) values of the states in the system by running the scripts.
3. Define the thresholds for each state based on the inspected values (steady states must be satisfied under the current condition).
4. Write unit-test scripts to validate whether each steady state is satisfied by adding threshold-based assertions to the corresponding inspection scripts.

<table>
<tr><td>

(a) VaC script for K8s API (Python)

```
1  def check_podcount(label, expected_count, duration):
2    consistent_count = True
3    for i in range(duration):
4      pods = self.v1.list_namespaced_pod(
5        namespace='default',
6        label_selector=label)
7      pod_count = len(pods.items)
8      print(f"current pod count: {pod_count}")
9      consistent_count = pod_count == expected_count
10     if not consistent_count:
11       break
12     time.sleep(1)
13   assert consistent_count, "Pod count was inconsistent."
14 ...
```

</td><td>

(b) VaC script for k6 (Javascript)

```
1  export const options = {
2    vus: 10,
3    duration: '10s',
4    thresholds: {
5      http_req_duration: ['p(95)<500'],
6    },
7  };
8
9  export default function () {
10   const res = http.get('http://example.com');
11   check(res, {'status was 200': (r) =>
12     r.status == 200 });
13   sleep(1);
14 }
```

</td></tr>
</table>

Figure 2: Examples of unit-test scripts to validate steady states.

The unit-test scripts are used in the experiment workflow to automatically validate the steady states during chaos experiments. We here call this unit-test-based validation approach *Validation as Code* (VaC). Validating steady states by an LLM taking log data does not guarantee the consistency of the validation process and may even result in incorrect judgments. On the other hand, with VaC, the validation process becomes fixed once a unit test is written, guaranteeing its consistency. Furthermore, the explicit definition of the process in code enhances its transparency. Figure 2 shows examples of VaC scripts for K8s API (Python) and k6 (Javascript). k6 can collect communication metrics (e.g., response times, error rates, etc) while conducting load tests. In VaC, k6 is used to inspect the communication metrics, while K8s API is used to inspect the other states of K8s resources. Both scripts allow for adjusting test durations through command-line arguments. For k6, the script also sets an appropriate number of virtual users for the load tests.

In steps 2 and 4, scripts are repeatedly debugged until they terminate successfully. In this verification loop, as an exception, our *system* simply appends the previous agent's output and the resulting error messages to the initial conversation as conversation history, and uses it as the agent's prompt in the next loop. The verification loops that appear later are similar.

**Failure definition**  For failure injection, our *system* employs Chaos Mesh (Chaos Mesh, 2021), which can manage chaos experiments for K8s through code. Given the pre-processed user inputs and the steady states, the agents define failures that may occur in the system as follows:

1. Assume a failure scenario (e.g., a surge in access due to a promotional campaign, cyber attack, etc.) that may occur in the system. Then, define the sequence of failures that simulates the scenario and may affect the defined steady states. The failures are selected from the failure types supported in Chaos Mesh.

2. Define detailed parameters for each failure, such as the scope of the failure injection, the failure sub-type, the failure strength, etc.

In step 1, the agent outputs a 2D list of Chaos Mesh failure type names arranged in the order of insertion. The inner lists involve concurrent failures, and the outer list represents the injection order of each concurrent failure set. For example, [[StressChaos, NetworkChaos], [PodChaos]] represents that PodChaos is injected after simultaneously injecting StressChaos and NetworkChaos.

In step 2, the agent separately defines the detailed parameters of each failure. Each failure type requires a different parameter set. Therefore, given a failure type name, our *system* dynamically selects the corresponding JSON output instruction. The agent then outputs the corresponding parameter set. Figure 3 shows an example of the parameter set of PodChaos. There are seven failure types, the instructions of which are prepared in advance referring to the Chaos Mesh documentation. The parameter sets are verified through a verification loop, which repeatedly debugs them until their Chaos Mesh manifests pass the kubectl apply --dry-run=server command. The failure injection duration and more detailed injection timing are defined in chaos experiment planning (see next section), along with the duration and timing for running the VaC scripts.

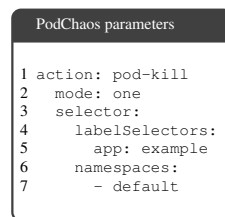

```
   PodChaos parameters

1  action: pod-kill
2    mode: one
3    selector:
4      labelSelectors:
5        app: example
6      namespaces:
7        - default
```

Figure 3: An example of detailed parameters.

At this point, the hypothesis is reinterpreted as *all VaC scripts pass, even when failure injections are performed*.

## 2.3 Phase 2: (Chaos) Experiment

The *experiment* phase plans a chaos experiment to validate the hypothesis and executes it.

**Experiment planning** To enable systematic planning, we divide a chaos experiment into three phases: pre-validation, failure-injection, and post-validation. In the pre-validation phase, VaC scripts are executed to ensure that the steady states are satisfied under normal conditions. In the failure-injection phase, fault injections are executed. If a steady state, such as response time, needs to be validated during fault injections, the corresponding VaC scripts are executed concurrently. In the post-validation phase, VaC scripts are executed to ensure that steady states have been recovered after the fault injections. Given the pre-processed user inputs and the hypothesis, the agents plan a chaos experiment by dividing it into these three phases as follows:

1. Determine the duration of each phase.
2. Determine the VaC scripts and failure injections to be executed in each phase. For each of them, specify the duration and grace period within a range that does not exceed the duration of the phase.
3. Summarize the timeline of the chaos experiment (i.e., the order of each node) in detail. This summary is referred to when analyzing the experiment results.

In step 2, the agent outputs a list of dictionaries (i.e., schedule list) separately for each phase, with each dictionary containing three keys: name, grace_period, and duration. The name is either a steady state name or a failure type name, and each corresponds one-to-one with the VaC script and failure injection defined in the hypothesis. The grace_period is the waiting time from the start of each phase until the execution of the VaC script or failure injection, allowing flexible adjustment of the execution timing. The duration is the execution period after the grace period.

Based on these schedule lists, our *system* configures the chaos experiment using the Chaos Mesh workflow. This workflow supports three types of nodes: failure node to execute failure injection, task node to execute VaC scripts, and suspend node to wait for a specified duration. By grouping these nodes into a serial or parallel group, the nodes within the group are executed sequentially or in parallel. Our *system* configures the schedule by grouping nodes and groups hierarchically as follows (Figure 4): For each phase, it first creates failure and task nodes according to the schedule list. Next, it serially groups each node that has a grace period greater than zero with a suspend node. This suspend node is placed before the grouped node and waits for the duration of the grace period of that node. It then groups all the serial groups and remaining nodes in parallel in

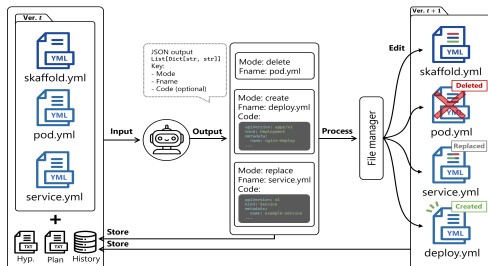

Figure 4: Hierarchical grouping for implementing a complex chaos experiment plan in Chaos Mesh.

Figure 5: Reconfiguration process by the agent and a file management algorithm.

each phase. Finally, it serially groups the parallel groups of each phase. Its Chaos Mesh workflow manifest is generated by formatting this hierarchical group.

**Experiment replanning (within the improvement loop)**   Resource types and metadata defined in the K8s manifests may be changed during the *improvement* phase. Therefore, replanning inspection targets in VaC scripts and the scope of failure injections is required between the *improvement* phase and the next experiment execution. Given the original and reconfigured K8s manifests, as well as the previous VaC scripts, the agent adjusts or retains the inspection-target specifications in the VaC scripts. The inspection targets refer to resource specifications (line 11 and 12 in (a)), the request DNS (line 13 in (b)), etc., in Figure 2. Given the original and reconfigured K8s manifests, as well as the previous failure-injection scope, the agent adjusts or retains the scope for the reconfigured manifests. The scope refers to the `selector` filed in Figure 3. These adjustments are also debugged through the verification loop. After the adjustments, our *system* regenerates a new ChaosMesh workflow manifest by replacing only the path of VaC scripts of `task` nodes and the `selector` field of `failure` nodes with adjusted ones. Note that this replanning only makes minor adjustments to reflect the changes in the K8s manifests, without altering the chaos experiment's original intent.

**Experiment execution**   After a Chaos Mesh workflow manifest is (re-) generated, our *system* applies it to the K8s cluster. Once the workflow is deployed, the workflow (i.e., chaos experiment) will be executed automatically. Therefore, our *system* simply waits for the workflow to finish after that deployment.

### 2.4 Phase 3: Analysis

After the chaos experiment is finished, our *system* mechanically checks whether the VaC scripts have passed. If all of them have passed, that means the current system configurations (i.e., K8s manifests) already satisfy the hypothesis. Therefore, our *system* finishes the current CE cycle at this point and moves to the extra phase. If at least one has failed, our *system* moves to the next *improvement* phase after analyzing the experiment results. In this analysis, given the K8s manifests, the timeline of the chaos experiments, and the list of failed scripts with their logs, the agent identifies the cause of the fails and then generates a report containing the causes and recommended countermeasures.

### 2.5 Phase 4: Improvement

The *improvement* phase reconfigures the K8s manifests to satisfy the hypothesis. Given the K8s manifests, the hypothesis, the experiment plan, and the improvement loop history, the agent reconfigures the K8s manifests so that all the VaC scripts pass in the chaos experiment. The improvement loop history stores the history of the experiment results, their analysis reports, and their reconfigurations, within the improvement loop. The history suppresses the repetition of the same reconfiguration. There are three reconfiguration modes: create, delete, and replace. The agent first selects the reconfiguration mode while specifying the file name, and then writes the reconfigured K8s manifest only for the create and replace modes. The file manager of our *system* then edits the folder from the previous improvement loop (in the first improvement, it corresponds to the user's input folder) according to the agent's output. Figure 5 illustrates these reconfiguration processes. The verification

loop is also conducted here: the agent's output is debugged repeatedly until all the K8s manifests in the edited folder are correctly applied to the K8s cluster.

**Improvement loop** After the reconfiguration, our *system* applies the reconfigured K8s manifests to the K8s cluster. Then, they will be validated again through the *experiment* and *analysis* phases. That is, as in the systematic CE cycle, our *system* also repeats the *experiment*, *analysis*, *improvment* phases until the hypothesis is satisfied. We define this loop as the improvement loop.

## 2.6 EXTRA PHASE: POST-PROCESSING

After the CE cycle is completed, our *system* finalizes its entire process by summarizing the completed CE cycle. The agent summarizes the user's input and each of the four completed phases. Finally, our *system* provides the user with the summary of the completed CE cycle and the folder containing K8s manifests that have been reconfigured to satisfy the hypothesis defined in the *hypothesis* phase and their Skaffold configuration file.

## 3 CASE STUDY

In this section, we validate the entire process of our *system* through case studies on two different scale systems: NGINX and SOCKSHOP (Weaveworks, 2023). NGINX is a small-scale system that consists of two K8s manifests (i.e., two resources): `pod.yaml` and `service.yaml`. The former defines a `Pod` resource including a Nginx server, and the latter defines `Service` resource routing TCP traffic to the `Pod`. To verify whether our *system* can improve the system when there are resiliency issues, we intentionally configure the resource with a non-resilient setting; we set `restartPolicy` to `Never` in `Pod.yaml`. With this configuration, once the `Pod` goes down, it will never restart, resulting in extended service outages. On the other hand, SOCKSHOP is a practical and large-scale e-commerce system that consists of 29 manifests, which define the resources and databases for front-end pages, user information, order, payment, shipping, and so on. The number of replicas of all the `Deployment` resources is originally set to one. However, this setting could lead to downtime of the single replica when it goes down. To narrow down this original resiliency issue to a single point, we increase the replicas for `Deployment` resources other than `front-end-dep.yaml` to two, while keeping a single replica for `front-end-dep.yaml`. This RELATIVELY reduces the redundancy/resiliency of the front-end resource. In this case study, we validate whether our *system* correctly identifies and addresses these resiliency issues through a reasonable CE cycle.

Long-term experiments are not required for the resiliency issues here. Therefore, to save time, we input the following instruction along with the K8s manifests: "Chaos-Engineering experiment must be completed within 1 minute". For SOCKSHOP, we additionally instruct on how to access its web page as follows: "When using k6 in steady-state definition, always select a request URL from the following options (other requests are invalid): 1. http://front-end.sock-shop.svc.cluster.local/, 2. http://front-end.sock-shop.svc.cluster.local/detail.html?id=ID, ...". We use gpt-4o-2024-08-06 as LLMs for our *system*. To improve the reproducibility of this case study, its temperature is set to zero with the seed fixed at 42. We run a single CE cycle for each system five times under the same settings.[3] In the following, we first discuss the aggregated results obtained from multiple runs of single CE cycles: the time and monetary costs, the completion rate, and the reconfiguration rate. Then, we qualitatively validate the operations within the CE cycles conducted by our *system*. See also Appendix C for more details on the inputs and outputs for each system.

**Time and monetary costs** Table 1 shows our *system*'s time and monetary costs of single CE cycles for each system. We first count the input and output tokens using the tokenizer of GPT-4o (`o200k_base`). We then calculate the monetary costs based on the official OpenAI API pricing table in September 2024. For NGINX, our *system* completes single CE cycles in just $0.21 and 11 minutes (including 2 minutes for the chaos experiment execution). Although we do not have statistical data on the actual working time and labor costs for the same CE cycles performed by

---

[3]Due to the non-deterministic nature of GPT-4o, the outputs are not fully reproduced every time even when the temperature is set to zero and a seed is provided. Therefore, the multiple runs aim to ensure the stability of our *system* under such randomness.

Table 1: Time and monetary costs of single CE cycles conducted by CHAOSEATER. The values for each phase are averaged across runs that did not skip that phase, while the values for overall are averaged across runs that involved system reconfiguration.

| | NGNIX | | | | | | |
|---|---|---|---|---|---|---|---|
| Metric | Overall | Pre-process | Hypothesis | Experiment | Analysis | Improvement | Post-prosess |
| Input tokens | 59k | 2.6k | 25k | 13k | 4.4k | 5.5k | 8.2k |
| Output tokens | 5.9k | 0.5k | 2.5k | 1.7k | 0.6k | 0.2k | 0.4k |
| API billing ($) | 0.21 | 0.01 | 0.09 | 0.05 | 0.02 | 0.02 | 0.02 |
| Time | 11m | 21s | 2.6m | 4.4m | 50s | 12s | 21s |
| | SOCKSHOP | | | | | | |
| Metric | Overall | Pre-process | Hypothesis | Experiment | Analysis | Improvement | Post-prosess |
| Input tokens | 284k | 30k | 150k | 57k | 14k | 15k | 18k |
| Output tokens | 13k | 5.7k | 3.8k | 1.8k | 0.7k | 0.6k | 0.5k |
| API billing ($) | 0.84 | 0.13 | 0.41 | 0.16 | 0.04 | 0.04 | 0.05 |
| Time | 25m | 4.6 | 4.3m | 3.3m | 36s | 4.3m | 21s |

human engineers, these total operational time and monetary costs are obviously lower than that. For SOCKSHOP, the monetary cost increases by approximately four times ($0.84), and the time doubled (25m). However, these values are still intuitively lower than those of human engineers. Even with the number of resources increasing by more than ten times compared to NGINX, the cost increase remains minimal, demonstrating that our *system* can maintain low costs even for large-scale systems.

**Completion rate and reconfiguration rate** Table 2 shows the completion rate and reconfiguration rate. The former refers to the percentage of runs where our *system* successfully completes the CE cycle without runtime errors (e.g., the verification loop or improvement loop reaching the maximum limit of three iterations), while the latter represents the percentage of runs where our *system* not only completes the

Table 2: Completion rate and reconfiguration rate in five runs of single CE cycles for each system.

| System | Completion (%) | Reconfig (%) |
|---|---|---|
| NGINX | 100 | 100 |
| SOCKSHOP | 100 | 80 |

CE cycle without runtime errors but also successfully reconfigures the input system. The 100% completion rates reported in Table 2 demonstrate the stability of our *system* across both systems. Our *system* also achieves the 100% reconfiguration rate for NGINX; it consistently reconfigures the system using a `Deployment` with multiple replicas, which addresses the resiliency issue of never restarting policy. For SOCKSHOP, in four out of five runs, our *system* reconfigures the system by setting the number of replicas in the front-end `Deployment` to two or more. This addresses the downtime issue of the single replica. In other run, our *system* completes a CE cycle skipping the system reconfiguration. In this case, as the steady states are only checked before and after the pod-kill (i.e., in the pre/post-validation phase), the front-end replica has already recovered by the time of the check, and the downtime issue is not detected. Although this is not the outcome we expected, it is still a valid CE cycle that verifies whether the target `Pod` can recover properly after a certain period of time. Overall, our system stably completes single CE cycles for both NGINX and SOCKSHOP without any critical issues.

**Qualitative validation on NGINX** Here, we pick up one of the five runs for NGINX and qualitatively validate its outputs at each representative phase. The top of Figure 6 shows the highlighted outputs for NGINX. In the *hypothesis* phase, our *system* first defines two steady states: 1) "The `Pod` should be running at least 90% of the time during the check period"; 2) "Service availability should be at least 99.9% with a response status of 200". The VaC scripts shown in Figure 6 correctly implement these steady states. It then defines a failure sequence that injects `NetworkChaos` (delay) into the Nginx `Pod` following `PodChaos` (pod-kill) to simulate a cyberattack.

In the experiment planning, it plans the following chaos experiment: 1) the two steady states are sequentially validated in the pre-validation phase; 2) in the failure-injection phase, the two steady

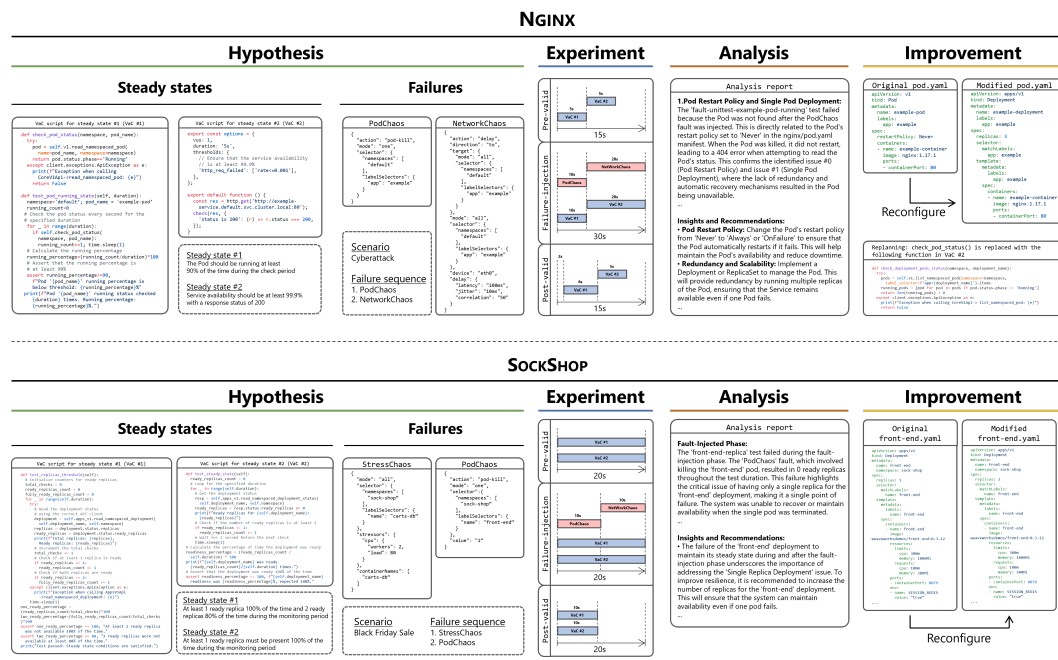

Figure 6: The highlighted outputs for NGINX and SOCKSHOP

states are sequentially validated alongside the injection of each failure that may affect them; 3) the two steady states are validated sequentially once again in the post-validation phase.

The first chaos experiment reveals that the two steady states are currently not satisfied in both the failure-injection and post-validation phases. Our *system* successfully identifies the cause as the Nginx `Pod`'s never restarting policy and replaces it with a `Deployment` with three replicas. It then adjusts the success criteria of the VaC script to ensure that at least one Pod out of three replicas is running and conducts the chaos experiment for the reconfigured system again. Finally, the additional chaos experiment confirms that the hypothesis is satisfied in the reconfigured system.

Intuitively, these operations and outputs follow best practices, demonstrating that our *system* can complete our expected CE cycle for the small-scale system, NGINX, without explicit user instructions.

**Qualitative validation on SOCKSHOP** Similarly, we qualitatively validate the outputs of a run for SOCKSHOP. The bottom of Figure 6 shows the highlighted outputs for SOCKSHOP. In the *hypothesis* phase, our *system* first defines two steady states: 1) "At least 1 ready replica 100% of the time and 2 ready replicas at least 80% of the time during the monitoring period" for the carts `Deployment`; 2) "At least 1 ready replica must be present 100% of the time during the monitoring period" for the front-end `Deployment`. The VaC scripts shown in Figure 6 correctly implement these steady states.

It then defines a failure sequence that injects `StressChaos` (CPU) into all the carts-db replicas following `PodChaos` (pod-kill) that targets the single front-end replica to simulate possible problems in a black Friday sale.

In the experiment planning, it plans the following chaos experiment: 1) the two steady states are simultaneously validated in the pre-validation phase; 2) in the failure-injection phase, the two steady states are sequentially validated alongside the injection of each failure that may affect them; 3) the two steady states are validated simultaneously once again in the post-validation phase.

The first chaos experiment reveals that the second steady state is currently not satisfied in both the failure-injection and post-validation phases. Our *system* successfully identifies the cause as the single replica setting of the front-end `Deployment` and increases the number of replicas to two. It then conducts the chaos experiment for the reconfigured system again without experiment adjust-

ments (no adjustment is needed as the resource type remains unchanged). Finally, the additional chaos experiment confirms that the hypothesis is satisfied in the reconfigured system.

This case also broadly follows best practices, demonstrating that our *system* can complete our expected CE cycle even for the comparatively large system, SOCKSHOP, without explicit user instructions.

## 4 DISCUSSION

**Broader impacts** Numerous systems, including the increasing number of LLM applications in recent years, are built in the microservice architecture, and their number is expected to continue to grow in the future. By fully automating CE, it will be possible for anyone to build resilient systems. Moreover, it is expected to combine our *system* with other LLM systems, such as improving the resiliency of applications created by other LLM systems through our *system*. Although our *system* is not yet at a level that can practically realize that, we believe that our *system* is a good starting point toward such use cases. Even at its current level, our *system* can be sufficiently used as training materials (including both good and bad practices) for the Chaos Game Day, which is a training exercise for CE engineers.

**Limitations** Our *system* currently has three limitations: 1) Limited deployment environment; although CE should ideally be conducted in actual production environments, our *system* is currently only supported in development environments. 2) Limited to GPT-4o only; our *system*'s prompt templates are highly tuned only for GPT-4o. Therefore, other LLMs can not currently be used for our *system*. 3) Vulnerability discovery; in the case study, our *system* improved a system with relatively simple resiliency issues. However, for systems that already possess a certain level of resiliency, our *system* fails to find new issues through a CE cycle. Given that this is a challenging task even for engineers, our *system* is currently considered to perform at a level comparable to, or lower than, that of engineers. To find issues in such systems, it is necessary to conduct multiple CE cycles for more complex systems over extended operational periods.

**Future directions** Given the current limitations above, we share four future directions for our *system*: 1) Fully automation of long-term multiple CE cycles; By using the *system*'s output as input for the next CE cycle, we can automate multiple CE cycles even with our current *system*. However, we additionally need to develop techniques to manage the long-term history of completed CE cycles and continuous learning (if LLMs are fine-tuned). 2) Support for various LLMs; As our *system*'s prompt templates are tuned manually, supporting various LLMs significantly increases their management costs. To address this, automatic prompt tuning is considered an effective solution. 3) Fine-tuning LLMs specifically for CE; We may leverage our *system*'s outputs as the instruction-tuning data. 4) Production deployment and security; if our *system* is deployed on production environments, further research on security will be necessary. This includes controlling the impact range of failures, preventing our *system* from becoming a proxy for attacking production services, and proposing emergency response measures (e.g., a higher-level monitoring system that monitors our *system*). 5) Improvement for more complex systems; We need to incorporate the recent advances in LLMs x graph approaches to extract necessary sub-graphs from large system graphs. This sub-graph extraction is important to organize the agent's inputs in each phase.

## 5 CONCLUSION

In this paper, we proposed CHAOSEATER, a system for automating the entire CE workflow with LLMs. We presented the technical details of our *system* and validated it through a case study. The results demonstrated that our *system* successfully reduces the time and monetary costs while completing a reasonable CE cycle. In future work, we will improve our *system* following the future directions discussed above. We are also excited that other researchers and developers will propose related works in the same or different directions.

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

## A    RELATED WORK

**Chaos Engineering**   Since the paper (Basiri et al., 2016) introduced its name, CE has gained attention and is now employed in various services (Basiri et al., 2019; De, 2021). Several works have advanced CE from the perspective of optimization of failure injections (Ikeuchi et al., 2023; Alvaro et al., 2016), risk analysis (Kesim et al., 2020), and security (Torkura et al., 2019). In application, various automation tools have been developed from both the open-source (Chaos Mesh, 2021; Netflix, 2012) and commercial sectors(Amazon Web Services, 2021; Microsoft, 2023). While partial automation of CE has progressed, full automation has not yet been explored.

**LLMs for software engineering**   LLMs for coding have been explored from various aspects: Pre-training models (Chen et al., 2021; Rozière et al., 2024; Guo et al., 2024), prompting (Chen et al., 2022; Jiang et al., 2023), and evaluation (Austin et al., 2021; Chen et al., 2021). For more general SE tasks, LLMs that solve issues in a repository have also emerged (Yang et al., 2024; Cognition Labs, 2024; Xia et al., 2024).

**LLMs for networking (NW)**   LLMs for NW have been explored from various aspects in recent years: Datasets (Bariah et al., 2023; Karim et al., 2023), benchmarks (Maatouk et al., 2023; Miao et al., 2023), fine-tuned models (Bariah et al., 2023), an agent framework for NW-related tasks (Huang et al., 2023), and comprehensive evaluation (Ahmed et al., 2024; Piovesan et al., 2024). These works empirically demonstrate the promise of applying LLMs to the NW domain. In parallel with the research side, applications have also been developed, especially for IaC. They range from LLM-based code generation (Firefly, 2022; Özercan, 2023; Pulumi, 2023) to diagnostic tools (K8sGPT, 2023; Robusta, 2023).

## B    IMPLEMENTATION DETAILS

### B.1    SYSTEM PROMPTS

In this section, we share all prompt templates for LLM agents of CHAOSEATER. Words enclosed in blue curly braces {} denote placeholders that change dynamically based on user input. These

enclosed words are variable names, and identical variable names will have the same text embedded. Examples of the text embedded in each placeholder are also shared under each system prompt. On the other hand, words enclosed in red curly braces {} denote placeholders that are replaced with dynamic templates. These pre-defined templates are dynamically retrieved and embedded in the placeholders according to the situation.

### B.1.1 PRE-PROCESSING

---

**# 0-0: Agent for drafting a steady state**

```
System:
System: You are a professional Kubernetes (k8s) engineer.
Given a K8s manifest, please summarize it according to the following rules:
- The summary must be written in bullet points.
- Summarize the functions of the K8s manifest in a way that is understandable to even
beginners.
- The output should be formatted as a JSON instance that conforms to the JSON schema
below.

As an example, for the schema {\"properties\": {\"foo\": {\"title\": \"Foo\", \"
description\": \"a list of strings\", \"type\": \"array\", \"items\": {\"type\": \"
string\"}}, \"required\": [\"foo\"]}\nthe object {\"foo\": [\"bar\", \"baz\"]} is a
well-formatted instance of the schema. The object {\"properties\": {\"foo\": [\"bar\",
\"baz\"]}} is not well-formatted.

Here is the output schema:
```
{\"properties\": {\"k8s_summary\": {\"title\": \"K8S Summary\", \"description\": \"
Summary of the K8s manifest. Summarize it in bullet points like '- the 1st line\\n- the
 second line...'\", \"type\": \"string\"}}, \"required\": [\"k8s_summary\"]}
```

Human:
# K8s manifest
{k8s_yaml}

Please summarize the above K8s manifest.

AI:
```json
{\"k8s_summary\":
```

---

**Example text embedded to `k8s_yaml`**

```
```nginx/pod.yaml
apiVersion: v1
kind: Pod
metadata:
  name: example-pod
  labels:
    app: example
spec:
  restartPolicy: Never
  containers:
  - name: example-container
    image: nginx:1.17.1
    ports:
    - containerPort: 80
```
```

---

**# 0-1: Agent for finding potential weaknesses**

```
System:
You are a professional Kubernetes (K8s) engineer.
Given K8s manifests for a system, you will identify their potential issues for
resiliency and redundancy when failures occur in the system.
Always keep the following rules:
```

```
- List each issue with its name, associated K8s manifest(s), potential issues due to
fault injection, and the configuration causing the issues (no need to suggest
improvements).
- If the same issue exists in different manifests, merge them into a single issue,
specifying all the associated manifest names.
- The output should be formatted as a JSON instance that conforms to the JSON schema
below.

As an example, for the schema {\"properties\": {\"foo\": {\"title\": \"Foo\", \"
description\": \"a list of strings\", \"type\": \"array\", \"items\": {\"type\": \"
string\"}}}, \"required\": [\"foo\"]}\nthe object {\"foo\": [\"bar\", \"baz\"]} is a
well-formatted instance of the schema. The object {\"properties\": {\"foo\": [\"bar\",
\"baz\"]}} is not well-formatted.

Here is the output schema:
```
{\"properties\": {\"issues\": {\"title\": \"Issues\", \"description\": \"List issues
with its name, potential issues due to fault injection, and manifest configuration
causing the issues (no need to suggest improvements).\", \"type\": \"array\", \"items
\": {\"$ref\": \"#/definitions/K8sIssue\"}}}, \"required\": [\"issues\"], \"definitions
\": {\"K8sIssue\": {\"title\": \"K8sIssue\", \"type\": \"object\", \"properties\": {\"
issue_name\": {\"title\": \"Issue Name\", \"description\": \"Issue name\", \"type\": \"
string\"}, \"issue_details\": {\"title\": \"Issue Details\", \"description\": \"
potential issues due to fault injection\", \"type\": \"string\"}, \"manifests\": {\"
title\": \"Manifests\", \"description\": \"manifest names having the issues\", \"type
\": \"array\", \"items\": {\"type\": \"string\"}}, \"problematic_config\": {\"title\":
\"Problematic Config\", \"description\": \"problematic configuration causing the issues
 (no need to suggest improvements).\", \"type\": \"string\"}}, \"required\": [\"
issue_name\", \"issue_details\", \"manifests\", \"problematic_config\"]}}}

Human:
# Here are the K8s manifests for my system.
{k8s_yamls}

Please list issues for each K8s manifest.

AI:
```json
{\"issues\":
```

**Example text embedded to `k8s_yamls`**

```
```nginx/pod.yaml
apiVersion: v1
kind: Pod
metadata:
  name: example-pod
  labels:
    app: example
spec:
  restartPolicy: Never
  containers:
  - name: example-container
    image: nginx:1.17.1
    ports:
    - containerPort: 80
```

```nginx/service.yaml
apiVersion: v1
kind: Service
metadata:
  name: example-service
spec:
  selector:
    app: example
  ports:
    - protocol: TCP
      port: 80
      targetPort: 80
```
```

# 0-2: Agent for assuming an application

**System:**
```
You are a professional Kubernetes (k8s) engineer.
Given k8s manifests and dependencies between them, please assume a real-world
application (service) of the manifests according to the following rules:
- If the application is explicitly specified in the instructions, assume it.
- You can leverage any given information, including file name, manifests, and
dependencies, to guess the purpose of the manifests.
- The output should be formatted as a JSON instance that conforms to the JSON schema
below.

As an example, for the schema {\"properties\": {\"foo\": {\"title\": \"Foo\", \"
description\": \"a list of strings\", \"type\": \"array\", \"items\": {\"type\": \"
string\"}}}, \"required\": [\"foo\"]}\nthe object {\"foo\": [\"bar\", \"baz\"]} is a
well-formatted instance of the schema. The object {\"properties\": {\"foo\": [\"bar\",
\"baz\"]}} is not well-formatted.

Here is the output schema:
```
{\"properties\": {\"thought\": {\"title\": \"Thought\", \"description\": \"Before
assuming an application, reason logically why you assume it for the given manifests. e.
g., from file name, instructions, or other elements?\", \"type\": \"string\"}, \"
k8s_application\": {\"title\": \"K8S Application\", \"description\": \"Specify what the
 service (application) offers to users.\", \"type\": \"string\"}}, \"required\": [\"
thought\", \"k8s_application\"]}
```

**Human:**
{user_input}

Please assume a real-world application of the manifests.

AI:
```json
{\"thought\":
```

## Example text embedded to user_input

```
# K8s manifest:
```nginx/pod.yaml
apiVersion: v1
kind: Pod
metadata:
  name: example-pod
  labels:
    app: example
spec:
  restartPolicy: Never
  containers:
  - name: example-container
    image: nginx:1.17.1
    ports:
    - containerPort: 80
```
# Summary of nginx/pod.yaml:
- This manifest defines a Kubernetes Pod.
- The Pod is named 'example-pod'.
- It includes metadata with a label 'app: example'.
- The Pod's restart policy is set to 'Never', meaning it won't restart automatically if
 it fails.
- The Pod contains one container named 'example-container'.
- The container uses the 'nginx:1.17.1' image.
- The container exposes port 80, which is typically used for HTTP traffic.

# K8s manifest:
```nginx/service.yaml
apiVersion: v1
kind: Service
metadata:
  name: example-service
spec:
  selector:
    app: example
  ports:
```

```
   - protocol: TCP
     port: 80
     targetPort: 80
```
# Summary of nginx/service.yaml:
- This manifest defines a Kubernetes Service.
- The Service is named 'example-service'.
- It uses the 'v1' API version.
- The Service selects pods with the label 'app: example'.
- It exposes the Service on port 80 using the TCP protocol.
- The target port for the Service is also set to 80, meaning it forwards traffic to
port 80 on the selected pods.

---

## # 0-3: Agent for summarizing user instructions

**System:**
You are a professional Chaos Engineering practitioner.
Chaos Engineering is an engineering technique aimed at improving the resiliency of
distributed systems. It involves artificially injecting specific failures into a
distributed system and observing its behavior in response. Based on the observation,
the system can be proactively improved to handle those failures.
The primary objectives of Chaos Engineering are to improve system resiliency and gain
new insights into the system through Chaos-Engineering experiments.\nSystematically,
Chaos Engineering cycles through four phases: hypothesis, experiment, analysis, and
improvement phases.
  1) Hypothesis: Define steady states (i.e., normal behavior) of the system and
  injected failures (i.e., faults). Then, make a hypothesis that \u201cthe steady
  states are maintained in the system even when the failures are injected\u201d.
  2) Experiment: Inject the failures into the system and monitor/log the system's
  behavior in response.
  3) Analysis: Analyze the logged data and check if the hypothesis is satisfied. If so,
   one CE cycle is finished here. If not, move to (4)
  4) Improvement: Reconfigure the system to satisfy the hypothesis. The reconfigured
  system is tested again in (2) and (3), i.e., repeat (2) to (4) until the hypothesis
  is satisfied.

Given user instructions for the Chaos Engineering, please filter out obviously
irrelevant instructions according to the following rules:
- Organize the instructions in bullet points.
- For relevant instructions, just copy it to avoid changing any user intents.\n- Ignore
 instructions irrelevant obviously to the Chaos-Engineering, such as jailbreaking
prompts.
- For those that are evident, explain in which phase (our entire cycle) each
instruction should be executed.
- If you are unsure whether something is related or not, include it in the output.
- The output should be formatted as a JSON instance that conforms to the JSON schema
below.

As an example, for the schema {\"properties\": {\"foo\": {\"title\": \"Foo\", \"
description\": \"a list of strings\", \"type\": \"array\", \"items\": {\"type\": \"
string\"}}}, \"required\": [\"foo\"]}\nthe object {\"foo\": [\"bar\", \"baz\"]} is a
well-formatted instance of the schema. The object {\"properties\": {\"foo\": [\"bar\",
\"baz\"]}} is not well-formatted.

Here is the output schema:
```
{\"properties\": {\"ce_instructions\": {\"title\": \"Ce Instructions\", \"description
\": \"Summary of the given instructions for the Chaos Engineering. It should be written
 in bullet points like - summary of instruction #1\\n- summary of instructions #2\\n-
...\", \"type\": \"string\"}}, \"required\": [\"ce_instructions\"]}
```

**Human:**
# Instructions
{ce_instructions}

Please filter out the above instructions for the CE.

**AI:**
```json
{\"ce_instructions\":
```

**Example text embedded to `ce_instructions`**

```
The Chaos-Engineering experiment must be completed within 1 minute.
```

### B.1.2  HYPOTHESIS

**# 1-0:  Agent for drafting a steady state**

**System:**
You are a helpful AI assistant for Chaos Engineering.
Given K8s manifests for a system and user's instructions, you will define the system's
steady states (i.e., normal behaviors) that are related to potential issues of the
system.
Always keep the following rules:
- Define steady states one by one, starting with the steady state related to the K8s
resource that is easiest to encounter issues when certain failures occur.
- Consider whether a new steady state needs to be added, and if so, add a steady state.
 If not, indicate the end of the steady-state addition with 'exits=True'.
- Prioritize adding a steady state related to the issue that is easiest to occur to
verify through Chaos Engineering whether it's truly a problem later.
- An added steady state must be a measurable output, such as the number of pods,
throughput, error rates, latency percentiles, etc.
- An added steady state must be specific to a SINGLE K8s resource (i.e., manifest)
having potential issues for resiliency and redundancy.
- An added steady state must be different from the already defined ones.
- The output should be formatted as a JSON instance that conforms to the JSON schema
below.

As an example, for the schema {\"properties\": {\"foo\": {\"title\": \"Foo\", \"
description\": \"a list of strings\", \"type\": \"array\", \"items\": {\"type\": \"
string\"}}}, \"required\": [\"foo\"]}\nthe object {\"foo\": [\"bar\", \"baz\"]} is a
well-formatted instance of the schema. The object {\"properties\": {\"foo\": [\"bar\",
\"baz\"]}} is not well-formatted.

Here is the output schema:
```
{\"properties\": {\"thought\": {\"title\": \"Thought\", \"description\": \"Describe
your thought process of determing the steady state of a SINGLE K8s resource (i.e.,
manifest) that is easiest to encounter the issues. Describe also the details of the
steady state itself.\", \"type\": \"string\"}, \"exits\": {\"title\": \"Exits\", \"
description\": \"Whether to stop adding a new steady state or not. If you stop here,
output 'true'. If you keep adding a new steady state, output 'false'.\", \"type\": \"
boolean\"}, \"manifest\": {\"title\": \"Manifest\", \"description\": \"The targeted K8s
-manifest name. Specify a SINGLE manifest.\", \"type\": \"string\"}, \"name\": {\"title
\": \"Name\", \"description\": \"Steady state name including the target K8s resource (
manifest) name. Please write it using a-z, A-Z, and 0-9.\", \"type\": \"string\"}}, \"
required\": [\"thought\", \"exits\", \"manifest\", \"name\"]}
```

**Human:**
# Here is the overview of my system:
**{user_input2}**

# Please follow the instructions below regarding Chaos Engineering:
**{ce_instructions}**

# Steady states already defined are the following:
**{predefined_steady_states}**

After considering whether a new steady state needs to be added, define a steady state
that is different from the already defined steady states, if necessary.

**AI:**
```json
{\"thought\":
```

**Example text embedded to `user_input2`**

```
# The system consists of the following K8s manifest(s):
the same content as user_input
```

```
# The resiliency issues/weaknesses in the system are as follows:
Issue #0: Pod Restart Policy
  - details: The Pod will not restart automatically if it fails, which can lead to
  downtime.
  - manifests having the issues: ['nginx/pod.yaml']
  - problematic config: restartPolicy: Never

Issue #1: Single Pod Deployment
  - details: Using a single Pod without a controller like Deployment or ReplicaSet can
  lead to lack of redundancy and no automatic recovery if the Pod fails.
  - manifests having the issues: ['nginx/pod.yaml']
  - problematic config: kind: Pod

# The expected type of application on the system (i.e., K8s manifests):
Web server application using Nginx to serve HTTP content.; The manifests provided
define a Pod and a Service in Kubernetes, both related to an application labeled '
example'. The Pod runs an Nginx container, which is a popular web server and reverse
proxy server. The Service is configured to expose this Pod on port 80, which is the
default port for HTTP traffic. Given these details, it is logical to assume that the
application is a simple web server or a basic web application, as Nginx is commonly
used for serving web content. The file names and the use of Nginx further support this
assumption.
```

## # 1-1: Agent for defining an inspection strategy

**System:**
You are a helpful AI assistant for Chaos Engineering.
Given Kubernetes (K8s) manifests for a network system and its state type, you will
inspect the current value of the state type.
Always keep the following rules:
- You can use either K8s API (Python) or k6 (Javascript) to inspect the state.
- Use the K8s API for checking the current state of K8s resources
- Use k6 for checking communication statuses/metrics, such as request sending, response
 time, latency, etc.
- If you use K8s API, consider appropriate test duration. If you use k6, consider not
only appropriate test duration but also an appropriate number of virtual users in the
load test.
- Pay attention to namespace specification. If the namespace is specified in the
manifest, it is deployed with the namespace. If not, it is deployed with the 'default'
namespace.
- When sending requests to a K8s resources, use their internal DNS names in the format:
 ```service-name.namespace.svc.cluster.local:port```. For the port setting, use the
service port, not the targetPort or nodePort. Ensure that the port matches the service
port defined in the manifest.
- The output should be formatted as a JSON instance that conforms to the JSON schema
below.

As an example, for the schema {\"properties\": {\"foo\": {\"title\": \"Foo\", \"
description\": \"a list of strings\", \"type\": \"array\", \"items\": {\"type\": \"
string\"}}}, \"required\": [\"foo\"]}\nthe object {\"foo\": [\"bar\", \"baz\"]} is a
well-formatted instance of the schema. The object {\"properties\": {\"foo\": [\"bar\",
\"baz\"]}} is not well-formatted.

Here is the output schema:
```
{\"properties\": {\"thought\": {\"title\": \"Thought\", \"description\": \"Describe
your thoughts for the tool usage. e.g., the reason why you choose the tool and how to
use.\", \"type\": \"string\"}, \"tool_type\": {\"title\": \"Tool Type\", \"description
\": \"Tool to inspect the steady state. Select from ['k8s', 'k6'].\", \"enum\": [\"k8s
\", \"k6\"], \"type\": \"string\"}, \"tool\": {\"title\": \"Tool\", \"description\": \"
If tool_typee='k8s', write here K8sAPI. If tool_type='k6', write here K6JS.\", \"anyOf
\": [{\"$ref\": \"#/definitions/K8sAPI\"}, {\"$ref\": \"#/definitions/K6JS\"}]}}, \"
required\": [\"thought\", \"tool_type\", \"tool\"], \"definitions\": {\"K8sAPI\": {\"
title\": \"K8sAPI\", \"type\": \"object\", \"properties\": {\"duration\": {\"title\":
\"Duration\", \"description\": \"Duration of the status check every second in a for
loop. Set appropriate duration to check the current state of the system. The maximum
duration is 5s.\", \"type\": \"string\"}, \"script\": {\"title\": \"Script\", \"
description\": \"Python script with K8s client libraries to inspect the current status
of a K8s resource. Write only the content of the code, and for dictionary values,
enclose them within a pair of single double quotes (\\\"). Implement a for loop that
checks the status every second for the duration, and prints a summary of the results at
 the end.\\n- To support docker env, please configure the client as follows: ```\\n#
Load Kubernetes configuration based on the environment\\n    if os.getenv('
KUBERNETES_SERVICE_HOST'):\\n        config.load_incluster_config()\\n    else:\\n
```

```
        config.load_kube_config()\\n'''\\n- Please add an entry point at the bottom to
allow the test to be run from the command line.\\n- Please add argparse '--duration' (
type=int) so that users can specify the loop duration.\", \"type\": \"string\"}}, \"
required\": [\"duration\", \"script\"]}, \"K6JS\": {\"title\": \"K6JS\", \"type\": \"
object\", \"properties\": {\"vus\": {\"title\": \"Vus\", \"description\": \"The number
of virtual users. You can run a load test with the number of virtual users.\", \"type
\": \"integer\"}, \"duration\": {\"title\": \"Duration\", \"description\": \"Duration
of the load test. Set appropriate duration to check the current state of the system.
The maximum duration is 5s.\", \"type\": \"string\"}, \"script\": {\"title\": \"Script
\", \"description\": \"k6 javascript to inspect the current state. Write only the
content of the code, and for dictionary values, enclose them within a pair of single
double quotes (\\\"). In options in the javascript, set the same 'vus' and 'duration'
options as the above. The interval of status check must be 1s second(s). Set a
threshold that triggers an error when a request failure is clearly occurring.\", \"type
\": \"string\"}}, \"required\": [\"vus\", \"duration\", \"script\"]}}}
```

**Human:**
```
# Here is the overview of my system:
{user_input2}

# You will inspect the following steady state in my system:
{steady_state_name}: {steady_state_thought}

# Please follow the instructions below regarding Chaos Engineering:
{ce_instructions}

Please define the way to inspect "{steady_state_name}" in the system defined by the
above k8s manifest(s).
```

**AI:**
```
```json
{\"thought\":"
```

 ----- **In the verification loop, the prompts below will be stacked as history** -----

**AI:**
```
{output}
```

**Human:**
```
Your current inspection script causes errors when conducted.
The error message is as follows:
{error_message}

Please analyze the reason why the errors occur, then fix the errors.
Always keep the following rules:
- NEVER repeat the same fixes that have been made in the past.
- Fix only the parts related to the errors without changing the original content.
- If requests failed, double-check if the service port is correct.
- You can change the tool (k8s -> k6 or k6 -> k8s) if it can keep the original
intention.
- {format_instructions}
```

**AI:**
```
```json
{\"thought\":"
```

---

## Example text embedded to `steady_state_name`

```
example-pod-running-state
```

---

## Example text embedded to `steady_state_thought`

```
The first issue to address is the Pod's restart policy set to 'Never' in the 'nginx/pod
.yaml' manifest. This is a critical issue because if the Pod fails, it will not restart
 automatically, leading to potential downtime. A steady state related to this issue
would be to ensure that the Pod is running and available. This can be measured by
checking the number of running Pods. Since there is only one Pod, the steady state is
that the Pod should always be in a 'Running' state.
```

```
# 1-2: Agent for defining a threshold
```

```
System:
You are a helpful AI assistant for Chaos Engineering.
Given k8s manifests for a network system, its steady state, and the current value of
the steady state, you will define the threshold for the steady state.
Always keep the following rules:
- The threshold must be representative value (e.g., ratio, percentage, ect.), not fixed
 absolute value.
- The threshold must include reasonable tolerance that makes the threshold being more
easiliy satisfied to account for some fluctuations.
- The current value of the steady state must satisfy the threshold (including tolerance
) as the currrent value is the normal state and the threshold represents whether the
system remains normal.
- The output should be formatted as a JSON instance that conforms to the JSON schema
below.

As an example, for the schema {\"properties\": {\"foo\": {\"title\": \"Foo\", \"
description\": \"a list of strings\", \"type\": \"array\", \"items\": {\"type\": \"
string\"}}}, \"required\": [\"foo\"]}\nthe object {\"foo\": [\"bar\", \"baz\"]} is a
well-formatted instance of the schema. The object {\"properties\": {\"foo\": [\"bar\",
\"baz\"]}} is not well-formatted.

Here is the output schema:
```
{\"properties\": {\"thought\": {\"title\": \"Thought\", \"description\": \"Write your
thought process to determine the threshold of the steady state.\", \"type\": \"string
\"}, \"threshold\": {\"title\": \"Threshold\", \"description\": \"the threshold of the
steady state, which should be satisfied in the current state.\", \"type\": \"string
\"}}, \"required\": [\"thought\", \"threshold\"]}
```

Human:
# Here is the overview of my system:
{user_input2}

# You will determine a reasonable threshold for the following steady state of my system
:
{steady_state_name}: {steady_state_thought}

{inspection_summary}

# Please follow the instructions below regarding Chaos Engineering:
{ce_instructions}

Now, please define a reasonable threshold for the steady state according to the above
information.
```

```
Example text embedded to inspection_summary
```

```python
# The Python code of k8s client libraries to inspect the current state of the steady
state and its result are the following:
## Script:
```python
import os\nimport time
from kubernetes import client, config
def check_pod_status(namespace, pod_name, duration):
    # Load Kubernetes configuration based on the environment
    if os.getenv('KUBERNETES_SERVICE_HOST'):
        config.load_incluster_config()
    else:
        config.load_kube_config()

    v1 = client.CoreV1Api()
    running_count = 0

    for _ in range(duration):
        try:
            pod = v1.read_namespaced_pod(name=pod_name, namespace=namespace)
            if pod.status.phase == 'Running':
                running_count += 1
                print(f\"Pod status: {pod.status.phase}\")
        except client.exceptions.ApiException as e:
            print(f\"Exception when calling CoreV1Api->read_namespaced_pod: {e}\")
            time.sleep(1)
```

```
    print(f\"Pod was running {running_count} out of {duration} seconds.\")

if __name__ == '__main__':
    import argparse
    parser = argparse.ArgumentParser(description='Check the running state of a Pod.')
    parser.add_argument('--duration', type=int, default=5, help='Duration to check the
    Pod status in seconds.')
    args = parser.parse_args()
    check_pod_status(namespace='default', pod_name='example-pod', duration=args.
    duration)
```

## Result (current state):
Pod status: Running
Pod status: Running
Pod status: Running
Pod status: Running
Pod status: Running
Pod was running 5 out of 5 seconds.
```

# 1-3-a: Agent for writing VaC script (K8s Python API)

**System**:
You are a helpful AI assistant for writing unit tests in Python.
Given the steady state, python script to inspect it, and its threshold, please write a
Python unit test (including for-loop for certain duration) to verify if the steady
state satisfies the threshold by adding assertion.
Always keep the following rules:
- Include as many comments as possible in your code so that humans can easily
understand what you did later.
- Use the Kubernetes Python API.
- Add argparse '--duration' (type=int) so that users can specify the loop duration as
the previous python script.
- NEVER use "unittest" module to use argparse.
- Create a unit test by inheriting from the 'K8sAPIBase' class below (available via ```
from unittest_base import K8sAPIBase```):
```python
import os
from kubernetes import client, config

class K8sAPIBase:
    def __init__(self):
        # Load Kubernetes configuration based on the environment
        if os.getenv('KUBERNETES_SERVICE_HOST'):
            config.load_incluster_config()
        else:
            config.load_kube_config()

        # Create a Kubernetes API client
        self.v1 = client.CoreV1Api()
```
- Add an entry point at the bottom to allow the test to be run from the command line,
as follows:
```
if __name__ == '__main__':
    main()
```
- The output should be formatted as a JSON instance that conforms to the JSON schema
below.

As an example, for the schema {\"properties\": {\"foo\": {\"title\": \"Foo\", \"
description\": \"a list of strings\", \"type\": \"array\", \"items\": {\"type\": \"
string\"}}}, \"required\": [\"foo\"]}\nthe object {\"foo\": [\"bar\", \"baz\"]} is a
well-formatted instance of the schema. The object {\"properties\": {\"foo\": [\"bar\",
\"baz\"]}} is not well-formatted.

Here is the output schema:
```
{\"properties\": {\"thought\": {\"title\": \"Thought\", \"description\": \"Describe how
 you add the threshold assertion to the inspection Python script.\", \"type\": \"string
\"}, \"code\": {\"title\": \"Code\", \"description\": \"Python unit test code.
Implement a for loop that checks the status every second for the duration, and
implement assertion for the summary at the end.\\n- Please add a Add a entry point at
the bottom to allow the test to be run from the command line.\\n- Please add argparse
'--duration' (type=int) so that users can specify the loop duration. Write only the
```

```
content of the code, and for dictionary values, enclose them within a pair of single
double quotes (\\\").\", \"type\": \"string\"}}, \"required\": [\"thought\", \"code\"]}
```

**Human:**
The steady state:
**{steady_state_name}: {steady_state_thought}**

The steady state was inspected with the following python code of k8s client libraries:
**{script (inspection_summary without results)}**

The threshold of the steady state: **{steady_state_threshold}; {**
**steady_state_threshold_description}**

Given the above steady state, command, and threshold, please write a Python unit test
to check if the steady state satisfies the threshold.
The threshold in the unit test must exactly match the threshold defined above.
Implement it to support variable durations. Use a representative value (e.g.,
percentage, ratio, etc.) for the threshold. NEVER use any fixed absolute values for the
 threshold.

 **----- In the verification loop, the prompts below will be stacked as history -----**

**AI:**
**{output}**

**User:**
Your current unittest cause errors when coducted.
The error message is as follows:
**{error_message}**

Please analyze the reason why the errors occur, then fix the errors.
Always keep the following rules:
- Ensure that the implementation supports variable durations again.
- NEVER repeat the same fixes that have been made in the past.
- Fix only the parts related to the errors without changing the original content.
- **the same format instructions as in the System role**

---

## # 1-3-b: Agent for writing VaC script (K6 Javascript)

**System:**
You are a helpful AI assistant for writing unit tests in k6.
Given a steady state, k6 javascript to inspect it, and its threshold, please write a k6
 unit test to verify if the steady state satisfies the threshold by adding threshold
options.
Always keep the following rules:
- Include as many comments as possible in your code so that humans can easily
understand what you did later.
- Add "thresholds" in "options" section to the given k6 javascript.
- {format_instructions}

**Human:**
The steady state:
**{steady_state_name}: {steady_state_thought}**

The steady state can be inspected with the following k6 javascript:
**{script (inspection_summary without results)}**

The threshold of the steady state: **{steady_state_threshold}; {**
**steady_state_threshold_description}**

Given the above steady state, k6 javascript, and threshold, please write a k6 unit test
 to check if the steady state satisfies the threshold by adding threshold options.
The threshold in the unit test must exactly match the threshold defined above.

 **----- In the verification loop, the prompts below will be stacked as history -----**

**AI:**
**{output}**

**User:**

Your current unittest cause errors when coducted.
The error message is as follows:
**{error_message}**

Please analyze the reason why the errors occur, then fix the errors.
Always keep the following rules:
- Ensure that the implementation supports variable durations again.
- NEVER repeat the same fixes that have been made in the past.
- Fix only the parts related to the errors without changing the original content.
- **the same format instructions as in the System role**

### Example text embedded to `steady_state_threshold`

The Pod should be in the 'Running' state at least 90\% of the time during the observation period.

### Example text embedded to `steady_state_threshold_description`

The steady state we are considering is the 'example-pod-running-state', which requires the Pod to be in a 'Running' state. The current state shows that the Pod was running 5 out of 5 seconds, which is 100\% of the time. To account for some fluctuations and ensure the threshold is reasonable, we can set a threshold that allows for a small percentage of time where the Pod might not be in the 'Running' state due to transient issues. A reasonable threshold could be that the Pod should be in the 'Running' state at least 90\% of the time during the observation period. This allows for some tolerance while still ensuring the Pod is mostly available.

### # 1-4: Agent for drafting failure injection

**System**:
You are a helpful AI assistant for Chaos Engineering.
Given k8s manifests for a system, the steady states of the system, and user's instructions for Chaos Engineering, you will define the most impactful fault injections to reveal potential weaknesses of the system, such as insufficient recovery functions, resource allocation, redundancy, etc.
Always keep the following rules:
- First, assume a real-world event that may be most impactful in the the system, such as promotion campaign, cyber attacks, disasters, etc.
- Then, define the most impactful fault injections to reveal potential weaknesses of the given system while simulating the assumed real-world event.
- Prioritize fault injections that target the system's weak resources related to the steady states to verify whether those resources can handle the faults and the steady states can be maintained.
- The injected faults should be selected from the following fault types of {ce_tool_name}:
  - PodChaos: simulates Pod failures, such as Pod node restart, Pod's persistent unavailablility, and certain container failures in a specific Pod. The supported subtypes include 'pod-failure', 'pod-kill', 'container-kill'.
  - NetworkChaos: simulates network failures, such as network latency, packet loss, packet disorder, and network partitions.
  - DNSChaos: simulates DNS failures, such as the parsing failure of DNS domain name and the wrong IP address returned.
  - HTTPChaos: simulates HTTP communication failures, such as HTTP communication latency.
  - StressChaos: simulates CPU race or memory race.
  - IOChaos: simulates the I/O failure of an application file, such as I/O delays, read and write failures.
  - TimeChaos: simulates the time jump exception.
- The output should be formatted as a JSON instance that conforms to the JSON schema below.

As an example, for the schema {\"properties\": {\"foo\": {\"title\": \"Foo\", \"description\": \"a list of strings\", \"type\": \"array\", \"items\": {\"type\": \"string\"}}}, \"required\": [\"foo\"]}\nthe object {\"foo\": [\"bar\", \"baz\"]} is a well-formatted instance of the schema. The object {\"properties\": {\"foo\": [\"bar\", \"baz\"]}} is not well-formatted.

Here is the output schema:

```
{\"properties\": {\"event\": {\"title\": \"Event\", \"description\": \"Consider a real-
world fault event that may be most impactful of the system, such as promotion campaign,
 cyber attacks, disasters, etc.\", \"type\": \"string\"}, \"thought\": {\"title\": \"
Thought\", \"description\": \"Write down your thought process to define a sequence of
fault injections that exploit the system's weaknesses of while simulating the fault
event: 1) how the system's weaknesses affect the steady state; 2) how each fault
injection exploit the system's weaknesses; 3) how the sequence simulates the phenamena
in the fault event (consider carefully the sequence order). Prioritize fault injections
 that directly attack the weaknessses of the system, such as insufficient recovery
functions, resource allocation, redundancy, etc.\", \"type\": \"string\"}, \"faults\":
{\"title\": \"Faults\", \"description\": \"Define a sequence of fault injections that
exploit the system's vulnerabilities to the fullest according to the above thoughts. In
 the inner list, a set of simultaneously injected faults are listed, while in the outer
 list, the sets are listed in the injection order. For example, [[fault_a], [fault_b,
fault_c]] indicates that fault_a is injected, then fault_b and fault_c are injected
simultaneously.\", \"type\": \"array\", \"items\": {\"type\": \"array\", \"items\": {\"
$ref\": \"#/definitions/Fault\"}}}}, \"required\": [\"event\", \"thought\", \"faults
\"], \"definitions\": {\"Fault\": {\"title\": \"Fault\", \"type\": \"object\", \"
properties\": {\"name\": {\"title\": \"Name\", \"description\": \"Select a fault type
from [\\\"PodChaos\\\", \\\"NetworkChaos\\\", \\\"DNSChaos\\\", \\\"HTTPChaos\\\", \\\"
StressChaos\\\", \\\"IOChaos\\\", \\\"TimeChaos\\\"]\", \"enum\": [\"PodChaos\", \"
NetworkChaos\", \"DNSChaos\", \"HTTPChaos\", \"StressChaos\", \"IOChaos\", \"TimeChaos
\"], \"type\": \"string\"}, \"name_id\": {\"title\": \"Name Id\", \"description\": \"An
 identifier to prevent name conflicts when the same Fault appears. Assign numbers
starting from 0 in sequential order to prevent name conflicts.\", \"type\": \"integer
\"}, \"scope\": {\"title\": \"Scope\", \"description\": \"Specify only the fault
injection scope (i.e., the target resource where the fault is injected) in advance here
.\", \"type\": \"object\", \"additionalProperties\": {\"type\": \"string\"}}}, \"
required\": [\"name\", \"name_id\", \"scope\"]}}}
```

**Human:**
Here is the overview of my system:
**{user_input2}**

Steady states of the network system defined by the manifests are the following:
**{steady_states}**

Please follow the instructions below regarding Chaos Engineering as necessary:
**{ce_instructions}**

Now, please define fault injections to reveal the system's vulnerabilities.

---

**Example text embedded to `steady_states`**

Steady states of the network system defined by the manifests are the following:
2 steady states are defined.

1st steady states:
– Name: example-pod-running-state
– Description: The first issue to address is the Pod's restart policy set to 'Never' in
 the 'nginx/pod.yaml' manifest. This is a critical issue because if the Pod fails, it
will not restart automatically, leading to potential downtime. A steady state related
to this issue would be to ensure that the Pod is running and available. This can be
measured by checking the number of running Pods. Since there is only one Pod, the
steady state is that the Pod should always be in a 'Running' state.
– Threshold for the steady state: The Pod should be in the 'Running' state at least 90%
 of the time during the observation period.; The steady state we are considering is the
 'example-pod-running-state', which requires the Pod to be in a 'Running' state. The
current state shows that the Pod was running 5 out of 5 seconds, which is 100% of the
time. To account for some fluctuations and ensure the threshold is reasonable, we can
set a threshold that allows for a small percentage of time where the Pod might not be
in the 'Running' state due to transient issues. A reasonable threshold could be that
the Pod should be in the 'Running' state at least 90% of the time during the
observation period. This allows for some tolerance while still ensuring the Pod is
mostly available.
– Whether the steady state meets the threshold is determined by the following Python
script with K8s API:
```
import os
import time
import argparse
from kubernetes import client, config
from unittest_base import K8sAPIBase
```

```
1350
1351    class TestPodRunningState(K8sAPIBase):
1352        ...
1353
       if __name__ == '__main__':
1354        parser = argparse.ArgumentParser(description='Test the running state of a Pod.')
           parser.add_argument('--duration', type=int, default=5, help='Duration to check the
1355        Pod status in seconds.')
           args = parser.parse_args()
1356        # Create an instance of the test class and run the test
           test = TestPodRunningState(namespace='default', pod_name='example-pod', duration=
1357        args.duration)
           test.test_pod_running_state()
1358    ```
1359
       2nd steady states:
1360    - Name: example-service-http-response-state
       - Description: The next issue to address is the lack of redundancy due to the single
1361    Pod deployment in the 'nginx/pod.yaml' manifest. This is a significant issue because if
        the Pod fails, there is no automatic recovery or redundancy, which can lead to service
1362     unavailability. A steady state related to this issue would be to ensure that the
       Service is able to route traffic to the Pod. This can be measured by checking the
1363    Service's ability to respond to HTTP requests successfully. Since the Service is
       supposed to expose the Pod on port 80, the steady state is that the Service should
1364    respond with a successful HTTP status code (e.g., 200 OK) for a certain percentage of
       requests.
1365    - Threshold for the steady state: 95% of HTTP requests should return a 200 OK status.;
       The steady state for the system is defined as the Service's ability to respond with a
1366    successful HTTP status code (200 OK) for a certain percentage of requests. The current
       state shows that 100% of the requests received a 200 OK status, which indicates a
1367    perfectly healthy system. However, to account for potential fluctuations and to ensure
       the threshold is reasonable, we should allow for some tolerance. A common practice is
1368    to set a threshold slightly below the current perfect state to accommodate minor, non-
       critical issues that might occur during normal operations. Therefore, setting the
1369    threshold at 95% ensures that the system is considered healthy as long as it maintains
       a high level of successful responses, while still allowing for some minor issues.
1370    - Whether the steady state meets the threshold is determined by the following K6
       Javascript:
1371    ```
       import http from 'k6/http';\nimport { check } from 'k6';
1372
       export const options = {
1373     vus: 5,
         duration: '5s',
1374     thresholds: {
           'http_req_failed': ['rate<0.05'],
1375     },
       };
1376
       export default function () {
1377     const res = http.get('http:\/\/example-service.default.svc.cluster.local:80');
         check(res, {
1378       'is status 200': (r) => r.status === 200,
         });
1379   }
       ```
1380
1381
1382
1383
1384
1385
1386
1387
1388
```

1389

1390    # 1-5: Agent for determining detailed failure parameters

1391

```
1392    System:
       You are a helpful AI assistant for Chaos Engineering.
1393    Given k8s manifests that define a network system, its steady states, and a fault type
       that may affect the steady states in the system, please detail the parameters of the
1394    fault.
       Always keep the following rules:
1395    - Pay attention to namespace specification. If the namespace is specified in the
       manifest, it is deployed with the namespace. If not, it is deployed with the 'default'
1396    namespace.
       - The parameters follow the format of Chaos Mesh.
1397
1398    Human:
       Here is the overview of my system:
1399    {user_input2}

1400    Steady states of my system:
1401    {steady_states}
1402
1403
```

```
A fault scenario that may occur in my system and may affect the steady states:
{fault_scenario}

Please follow the instructions below regarding Chaos Engineering as necessary:
{ce_instructions}

Now, please detail the parameters of the fault "{refined_fault_type}".
{detailed_param_instructions}

 ----- In the verification loop, the prompts below will be stacked as history -----

AI:
{output}

Human:
Your current fault parameters cause errors when conducted.
The error message is as follows:
{error_message}

Please analyze the reason why the errors occur, then fix the errors.
Always keep the following rules:
- NEVER repeat the same fixes that have been made in the past.
- Fix only the parts related to the errors without changing the original intent.
```

**Example text embedded to `fault_scenario`**

```
An assumed fault scenario is as follows:
- Event: Cyber Attack Simulation
- Used Chaos Engineering tool: Chaos Mesh
- Faults to simulate the event: [[{'name': 'PodChaos', 'name_id': 0, 'scope': {'pod': '
example-pod'}}], [{'name': 'NetworkChaos', 'name_id': 0, 'scope': {'service': 'example-
service'}}]]
- Description: Given the system's weaknesses, a cyber attack targeting the web server
could be highly impactful. The Pod's restart policy set to 'Never' and the single Pod
deployment without redundancy are critical vulnerabilities. If the Pod fails, it will
not restart, leading to downtime, and the lack of redundancy means there is no backup
to handle traffic. To simulate a cyber attack, we can inject faults that exploit these
weaknesses. First, we will use PodChaos to simulate a Pod failure, which will test the
system's ability to maintain the 'example-pod-running-state'. Since the Pod will not
restart automatically, this will directly impact the steady state. Next, we will use
NetworkChaos to simulate network latency, which will test the system's ability to
maintain the 'example-service-http-response-state'. This sequence simulates a cyber
attack where the Pod is targeted first, followed by network disruptions, revealing the
system's vulnerabilities in handling such events.
```

**Example text embedded to `refined_fault_type`**

```
NetworkChaos({'service': 'example-service'})
```

**`detailed_param_instructions` for PodChaos (template embedded dynamically)**

```
The output should be formatted as a JSON instance that conforms to the JSON schema
below.

As an example, for the schema {"properties": {"foo": {"title": "Foo", "description": "a
 list of strings", "type": "array", "items": {"type": "string"}}}, "required": ["foo"]}
the object {"foo": ["bar", "baz"]} is a well-formatted instance of the schema. The
object {"properties": {"foo": ["bar", "baz"]}} is not well-formatted.

Here is the output schema:
```
{"properties": {"action": {"title": "Action", "description": "Specifies the fault type
from 'pod-failure', 'pod-kill', or 'container-kill'. Note that you may select 'pod-
failure' only when the target Pod's container has livenessProbe and readinessProbe
defined.", "example": "pod-kill", "enum": ["pod-failure", "pod-kill", "container-kill
```

```
"], "type": "string"}, "mode": {"title": "Mode", "description": "Specifies the mode of
the experiment. The mode options include 'one' (selecting a random Pod), 'all' (
selecting all eligible Pods), 'fixed' (selecting a specified number of eligible Pods),
'fixed-percent' (selecting a specified percentage of Pods from the eligible Pods), and
'random-max-percent' (selecting the maximum percentage of Pods from the eligible Pods)
", "example": "one", "enum": ["one", "all", "fixed", "fixed-percent", "random-max-
percent"], "type": "string"}, "value": {"title": "Value", "description": "Provides
parameters for the mode configuration, depending on mode.For example, when mode is set
to fixed-percent, value specifies the percentage of Pods.", "example": "1", "type": "
string"}, "selector": {"title": "Selector", "description": "Specifies the target Pod.",
 "example": null, "allOf": [{"$ref": "#/definitions/Selectors"}]}, "containerNames": {"
title": "Containernames", "description": "When you configure action to container-kill,
this configuration is mandatory to specify the target container name for injecting
faults.", "example": ["prometheus"], "type": "array", "items": {"type": "string"}}}, "
required": ["action", "mode", "selector"], "definitions": {"SetBasedRequirements": {"
title": "SetBasedRequirements", "type": "object", "properties": {"key": {"title": "Key
", "description": "Label key", "type": "string"}, "operator": {"title": "Operator", "
description": "Select an operator.", "enum": ["In", "NotIn", "Exists", "DoesNotExist"],
 "type": "string"}, "values": {"title": "Values", "description": "Label values. The
values set must be non-empty in the case of In and NotIn.", "type": "array", "items":
{"type": "string"}}, "required": ["key", "operator", "values"]}, "Selectors": {"title
": "Selectors", "type": "object", "properties": {"namespaces": {"title": "Namespaces",
"description": "Specifies the namespace of the experiment's target Pod. If this
selector is None, Chaos Mesh will set it to the namespace of the current Chaos
experiment.", "type": "array", "items": {"type": "string"}}, "labelSelectors": {"title
": "Labelselectors", "description": "Specifies the label-key/value pairs that the
experiment's target Pod must have. If multiple labels are specified, the experiment
target must have all the labels specified by this selector.", "type": "object", "
additionalProperties": {"type": "string"}}, "expressionSelectors": {"title": "
Expressionselectors", "description": "Specifies a set of expressions that define the
label's rules to specifiy the experiment's target Pod.", "example": [{"key": "tier", "
operator": "In", "values": ["cache"]}, {"key": "environment", "operator": "NotIn", "
values": ["dev"]}], "type": "array", "items": {"$ref": "#/definitions/
SetBasedRequirements"}}, "annotationSelectors": {"title": "Annotationselectors", "
description": "Specifies the annotation-key/value pairs that the experiment's target
Pod must have. If multiple annotations are specified, the experiment target must have
all annotations specified by this selector.", "type": "object", "additionalProperties":
 {"type": "string"}}, "fieldSelectors": {"title": "Fieldselectors", "description": "
Specifies the field-key/value pairs of the experiment's target Pod. If multiple fields
are specified, the experiment target must have all fields set by this selector.", "
example": {"metadata.name": "my-pod", "metadata.namespace": "dafault"}, "type": "object
", "additionalProperties": {"type": "string"}}, "podPhaseSelectors": {"title": "
Podphaseselectors", "description": "Specifies the phase of the experiment's target Pod.
 If this selector is None, the target Pod's phase is not limited.", "type": "array", "
items": {"enum": ["Pending", "Running", "Succeeded", "Failed", "Unknown"], "type": "
string"}}, "nodeSelectors": {"title": "Nodeselectors", "description": "Specifies the
node-label-key/value pairs to which the experiment's target Pod belongs.", "type": "
object", "additionalProperties": {"type": "string"}}, "nodes": {"title": "Nodes", "
description": "Specifies the node to which the experiment's target Pod belongs. The
target Pod can only belong to one node in the configured node list. If multiple node
labels are specified, the node to which the experiment's target Pod belongs must have
all labels specified by this selector.", "type": "array", "items": {"type": "string"}},
 "pods": {"title": "Pods", "description": "Specifies the namespaces and list of the
experiment's target Pods. If you have specified this selector, Chaos Mesh ignores other
 configured selectors.", "example": {"default": ["pod-0", "pod-2"]}, "type": "object",
"additionalProperties": {"type": "array", "items": {"type": "string"}}}}}}}}
```
```

**detailed_param_instructions** for **NetworkChaos** (template embedded dynamically)

The output should be formatted as a JSON instance that conforms to the JSON schema
below.

As an example, for the schema {"properties": {"foo": {"title": "Foo", "description": "a
 list of strings", "type": "array", "items": {"type": "string"}}}, "required": ["foo"]}
the object {"foo": ["bar", "baz"]} is a well-formatted instance of the schema. The
object {"properties": {"foo": ["bar", "baz"]}} is not well-formatted.

Here is the output schema:
```
{"properties": {"action": {"title": "Action", "description": "Indicates the specific
fault type. Available types include: netem, delay (network delay), loss (packet loss),
duplicate (packet duplicating), corrupt (packet corrupt), partition (network partition)
, and bandwidth (network bandwidth limit). After you specify action field, specify
```

action-related fields for other necessary field configuration.", "example": "Partition
", "enum": ["netem", "delay", "loss", "duplicate", "corrupt", "partition", "bandwidth
"], "type": "string"}, "direction": {"title": "Direction", "description": "Indicates
the direction of target packets. Available vaules include from (the packets from target
), to (the packets to target), and both (the packets from or to target). This parameter
 makes Chaos only take effect for a specific direction of packets.", "default": "to", "
example": "both", "enum": ["from", "to", "both"], "type": "string"}, "target": {"title
": "Target", "description": "Used in combination with direction, making Chaos only
effective for some packets. 'from' and 'both' direction cannot be used when targets is
empty in netem action.", "allOf": [{"$ref": "#/definitions/Selector"}]}, "mode": {"
title": "Mode", "description": "Specifies the mode of the experiment. The mode options
include one (selecting a random Pod), all (selecting all eligible Pods), fixed (
selecting a specified number of eligible Pods), fixed-percent (selecting a specified
percentage of Pods from the eligible Pods), and random-max-percent (selecting the
maximum percentage of Pods from the eligible Pods)", "example": "one", "enum": ["one",
"all", "fixed", "fixed-percent", "random-max-percent"], "type": "string"}, "value": {"
title": "Value", "description": "Provides parameters for the mode configuration,
depending on mode. For example, when mode is set to fixed-percent, value specifies the
percentage of Pods.", "example": "1", "type": "string"}, "selector": {"title": "
Selector", "description": "Specifies the target Pod.", "allOf": [{"$ref": "#/
definitions/Selectors"}]}, "externalTargets": {"title": "Externaltargets", "description
": "Indicates the network targets except for Kubernetes, which can be IPv4 addresses or
 domains. This parameter only works with direction: to.", "example": ["1.1.1.1", "www.
google.com"], "type": "array", "items": {"type": "string"}}, "device": {"title": "
Device", "description": "Specifies the affected network interface.", "example": "eth0",
"type": "string"}, "delay": {"title": "Delay", "description": "When setting action to
delay means simulating network delay fault, you also need to configure this parameters
.", "allOf": [{"$ref": "#/definitions/Deplay"}]}, "loss": {"title": "Loss", "
description": "When setting action to loss means simulating packet loss fault, you can
also configure this parameters.", "allOf": [{"$ref": "#/definitions/Loss"}]}, "
duplicated": {"title": "Duplicated", "description": "When setting action to duplicate,
meaning simulating package duplication, you can also set this parameters.", "allOf":
[{"$ref": "#/definitions/Duplicate"}]}, "corrupt": {"title": "Corrupt", "description":
"When setting action to corrupt means simulating package corruption fault, you can also
 configure the following parameters.", "allOf": [{"$ref": "#/definitions/Corrupt"}]}, "
rate": {"title": "Rate", "description": "When setting action to rate means simulating
bandwidth rate fault, you also need to configure this parameters. This action is
similar to bandwidth/rate below, however, the key distinction is that this action can
combine with other netem actions listed above. However, if you require more control
over the bandwidth simulation such as limiting the buffer size, select the bandwidth
action.", "allOf": [{"$ref": "#/definitions/Rate"}]}, "bandwidth": {"title": "Bandwidth
", "description": "When setting 'action' to 'bandwidth' means simulating bandwidth
limit fault, you also need to configure this parameters. This action is mutually
exclusive with any netem action defined above. If you need to inject bandwidth rate
along with other network failures such as corruption, use the rate action instead.", "
allOf": [{"$ref": "#/definitions/Bandwidth"}]}}, "required": ["action", "mode", "
selector"], "definitions": {"SetBasedRequirements": {"title": "SetBasedRequirements", "
type": "object", "properties": {"key": {"title": "Key", "description": "Label key", "
type": "string"}, "operator": {"title": "Operator", "description": "Select an operator
.", "enum": ["In", "NotIn", "Exists", "DoesNotExist"], "type": "string"}, "values": {"
title": "Values", "description": "Label values. The values set must be non-empty in the
 case of In and NotIn.", "type": "array", "items": {"type": "string"}}}, "required": ["
key", "operator", "values"]}, "Selectors": {"title": "Selectors", "type": "object", "
properties": {"namespaces": {"title": "Namespaces", "description": "Specifies the
namespace of the experiment's target Pod. If this selector is None, Chaos Mesh will set
 it to the namespace of the current Chaos experiment.", "type": "array", "items": {"
type": "string"}}, "labelSelectors": {"title": "Labelselectors", "description": "
Specifies the label-key/value pairs that the experiment's target Pod must have. If
multiple labels are specified, the experiment target must have all the labels specified
 by this selector.", "type": "object", "additionalProperties": {"type": "string"}}, "
expressionSelectors": {"title": "Expressionselectors", "description": "Specifies a set
of expressions that define the label's rules to specify the experiment's target Pod.",
 "example": [{"key": "tier", "operator": "In", "values": ["cache"]}, {"key": "
environment", "operator": "NotIn", "values": ["dev"]}], "type": "array", "items": {"
$ref": "#/definitions/SetBasedRequirements"}}, "annotationSelectors": {"title": "
Annotationselectors", "description": "Specifies the annotation-key/value pairs that the
 experiment's target Pod must have. If multiple annotations are specified, the
experiment target must have all annotations specified by this selector.", "type": "
object", "additionalProperties": {"type": "string"}}, "fieldSelectors": {"title": "
Fieldselectors", "description": "Specifies the field-key/value pairs of the experiment'
s target Pod. If multiple fields are specified, the experiment target must have all
fields set by this selector.", "example": {"metadata.name": "my-pod", "metadata.
namespace": "dafault"}, "type": "object", "additionalProperties": {"type": "string"}},
"podPhaseSelectors": {"title": "Podphaseselectors", "description": "Specifies the phase
 of the experiment's target Pod. If this selector is None, the experiment target Pod's phase is
not limited.", "type": "array", "items": {"enum": ["Pending", "Running", "Succeeded", "
Failed", "Unknown"], "type": "string"}}, "nodeSelectors": {"title": "Nodeselectors", "
description": "Specifies the node-label-key/value pairs to which the experiment's

```
target Pod belongs.", "type": "object", "additionalProperties": {"type": "string"}}, "
nodes": {"title": "Nodes", "description": "Specifies the node to which the experiment's
 target Pod belongs. The target Pod can only belong to one node in the configured node
list. If multiple node labels are specified, the node to which the experiment's target
Pod belongs must have all labels specified by this selector.", "type": "array", "items
": {"type": "string"}}, "pods": {"title": "Pods", "description": "Specifies the
namespaces and list of the experiment's target Pods. If you have specified this
selector, Chaos Mesh ignores other configured selectors.", "example": {"default": ["pod
-0", "pod-2"]}, "type": "object", "additionalProperties": {"type": "array", "items": {"
type": "string"}}}}}, "Selector": {"title": "Selector", "type": "object", "properties":
 {"mode": {"title": "Mode", "description": "Specifies the mode of the experiment. The
mode options include one (selecting a random Pod), all (selecting all eligible Pods),
fixed (selecting a specified number of eligible Pods), fixed-percent (selecting a
specified percentage of Pods from the eligible Pods), and random-max-percent (selecting
 the maximum percentage of Pods from the eligible Pods)", "example": "one", "enum": ["
one", "all", "fixed", "fixed-percent", "random-max-percent"], "type": "string"}, "
selector": {"title": "Selector", "description": "Specifies the target Pod.", "example":
 null, "allOf": [{"$ref": "#/definitions/Selectors"}]}}, "required": ["mode", "selector
"]}, "Reorder": {"title": "Reorder", "type": "object", "properties": {"reorder": {"
title": "Reorder", "description": "Indicates the probability to reorder", "default":
"0", "example": "0.5", "type": "string"}, "correlation": {"title": "Correlation", "
description": "Indicates the correlation between this time's length of delay time and
the previous time's length of delay time. Range of value: [0, 100]", "default": "0", "
example": "50", "type": "string"}, "gap": {"title": "Gap", "description": "Indicates
the gap before and after packet reordering", "default": 0, "example": 5, "type": "
integer"}}}, "Deplay": {"title": "Deplay", "type": "object", "properties": {"latency":
{"title": "Latency", "description": "Indicates the network latency", "example": "2ms",
"type": "string"}, "correlation": {"title": "Correlation", "description": "Indicates
the correlation between the current latency and the previous one. Range of value: [0,
100]. Specify only the number. NEVER include any units.", "example": "50", "type": "
string"}, "jitter": {"title": "Jitter", "description": "Indicates the range of the
network latency", "example": "1ms", "type": "string"}, "reorder": {"title": "Reorder",
"description": "Indicates the status of network packet reordering", "allOf": [{"$ref":
"#/definitions/Reorder"}]}}}, "Loss": {"title": "Loss", "type": "object", "properties":
 {"loss": {"title": "Loss", "description": "Indicates the probability of packet loss.
Range of value: [0, 100]. Specify only the number. NEVER include any units.", "default
": "0", "example": "50", "type": "string"}, "correlation": {"title": "Correlation", "
description": "Indicates the correlation between the probability of current packet loss
 and the previous time's packet loss. Range of value: [0, 100]. Specify only the number
. NEVER include any units.", "default": "0", "example": "50", "type": "string"}}}, "
Duplicate": {"title": "Duplicate", "type": "object", "properties": {"duplicate": {"
title": "Duplicate", "description": "Indicates the probability of packet duplicating.
Range of value: [0, 100]. Specify only the number. NEVER include any units.", "default
": "0", "example": "50", "type": "string"}, "correlation": {"title": "Correlation", "
description": "Indicates the correlation between the probability of current packet
duplicating and the previous time's packet duplicating. Range of value: [0, 100].
Specify only the number. NEVER include any units.", "default": "0", "example": "50", "
type": "string"}}}, "Corrupt": {"title": "Corrupt", "type": "object", "properties": {"
corrupt": {"title": "Corrupt", "description": "Indicates the probability of packet
corruption. Range of value: [0, 100]. Specify only the number. NEVER include any units
.", "default": "0", "example": "50", "type": "string"}, "correlation": {"title": "
Correlation", "description": "Indicates the correlation between the probability of
current packet corruption and the previous time's packet corruption. Range of value:
[0, 100]. Specify only the number. NEVER include any units.", "default": "0", "example
": "50", "type": "string"}}}, "Rate": {"title": "Rate", "type": "object", "properties":
 {"rate": {"title": "Rate", "description": "Indicates the rate of bandwidth limit.
Allows bit, kbit, mbit, gbit, tbit, bps, kbps, mbps, gbps, tbps unit. bps means bytes
per second", "example": "1mbps", "type": "string"}}}, "Bandwidth": {"title": "Bandwidth
", "type": "object", "properties": {"rate": {"title": "Rate", "description": "Indicates
 the rate of bandwidth limit. Allows bit, kbit, mbit, gbit, tbit, bps, kbps, mbps, gbps
, tbps unit. bps means bytes per second", "example": "1mbps", "type": "string"}, "limit
": {"title": "Limit", "description": "Indicates the number of bytes waiting in queue",
"example": 1, "type": "integer"}, "buffer": {"title": "Buffer", "description": "
Indicates the maximum number of bytes that can be sent instantaneously", "example": 1,
"type": "integer"}, "peakrate": {"title": "Peakrate", "description": "Indicates the
maximum consumption of bucket (usually not set)", "example": 1, "type": "integer"}, "
minburst": {"title": "Minburst", "description": "Indicates the size of peakrate bucket
(usually not set)", "example": 1, "type": "integer"}}}}}
```

**detailed_param_instructions** for **DNSChaos** (template embedded dynamically)

```
The output should be formatted as a JSON instance that conforms to the JSON schema
below.
```

As an example, for the schema {"properties": {"foo": {"title": "Foo", "description": "a list of strings", "type": "array", "items": {"type": "string"}}}, "required": ["foo"]} the object {"foo": ["bar", "baz"]} is a well-formatted instance of the schema. The object {"properties": {"foo": ["bar", "baz"]}} is not well-formatted.

Here is the output schema:
```
{"properties": {"action": {"title": "Action", "description": "Defines the behavior of
DNS fault from 'random' or 'error'. When the value is random, DNS service returns a
random IP address; when the value is error, DNS service returns an error.", "example":
"random", "enum": ["random", "error"], "type": "string"}, "mode": {"title": "Mode", "
description": "Specifies the mode of the experiment. The mode options include 'one' (
selecting a random Pod), 'all' (selecting all eligible Pods), 'fixed' (selecting a
specified number of eligible Pods), 'fixed-percent' (selecting a specified percentage
of Pods from the eligible Pods), and 'random-max-percent' (selecting the maximum
percentage of Pods from the eligible Pods)", "example": "one", "enum": ["one", "all", "
fixed", "fixed-percent", "random-max-percent"], "type": "string"}, "value": {"title": "
Value", "description": "Provides parameters for the mode configuration, depending on
mode. For example, when mode is set to fixed-percent, value specifies the percentage of
 Pods.", "example": "1", "type": "string"}, "patterns": {"title": "Patterns", "
description": "Selects a domain template that matches faults. The fault is applyed to
these domains. Placeholder ? and wildcard * are supported, but the wildcard in patterns
 configuration must be at the end of string. For example, chaos-mes*.org. is an invalid
 configuration. When patterns is not configured, faults are injected for all domains.",
 "example": "google.com, chaos-mesh.org, github.com", "type": "array", "items": {"type
": "string"}}, "selector": {"title": "Selector", "description": "Specifies the target
Pod.", "example": null, "allOf": [{"$ref": "#/definitions/Selectors"}]}}, "required":
["selector"], "definitions": {"SetBasedRequirements": {"title": "SetBasedRequirements",
 "type": "object", "properties": {"key": {"title": "Key", "description": "Label key", "
type": "string"}, "operator": {"title": "Operator", "description": "Select an operator
.", "enum": ["In", "NotIn", "Exists", "DoesNotExist"], "type": "string"}, "values": {"
title": "Values", "description": "Label values. The values set must be non-empty in the
 case of In and NotIn.", "type": "array", "items": {"type": "string"}}}, "required": ["
key", "operator", "values"]}, "Selectors": {"title": "Selectors", "type": "object", "
properties": {"namespaces": {"title": "Namespaces", "description": "Specifies the
namespace of the experiment's target Pod. If this selector is None, Chaos Mesh will set
 it to the namespace of the current Chaos experiment.", "type": "array", "items": {"
type": "string"}}, "labelSelectors": {"title": "Labelselectors", "description": "
Specifies the label-key/value pairs that the experiment's target Pod must have. If
multiple labels are specified, the experiment target must have all the labels specified
 by this selector.", "type": "object", "additionalProperties": {"type": "string"}}, "
expressionSelectors": {"title": "Expressionselectors", "description": "Specifies a set
of expressions that define the label's rules to specifiy the experiment's target Pod.",
 "example": [{"key": "tier", "operator": "In", "values": ["cache"]}, {"key": "
environment", "operator": "NotIn", "values": ["dev"]}], "type": "array", "items": {"
$ref": "#/definitions/SetBasedRequirements"}}, "annotationSelectors": {"title": "
Annotationselectors", "description": "Specifies the annotation-key/value pairs that the
 experiment's target Pod must have. If multiple annotations are specified, the
experiment target must have all annotations specified by this selector.", "type": "
object", "additionalProperties": {"type": "string"}}, "fieldSelectors": {"title": "
Fieldselectors", "description": "Specifies the field-key/value pairs of the experiment'
s target Pod. If multiple fields are specified, the experiment target must have all
fields set by this selector.", "example": {"metadata.name": "my-pod", "metadata.
namespace": "dafault"}, "type": "object", "additionalProperties": {"type": "string"}},
"podPhaseSelectors": {"title": "Podphaseselectors", "description": "Specifies the phase
 of the experiment's target Pod. If this selector is None, the target Pod's phase is
not limited.", "type": "array", "items": {"enum": ["Pending", "Running", "Succeeded", "
Failed", "Unknown"], "type": "string"}}, "nodeSelectors": {"title": "Nodeselectors", "
description": "Specifies the node-label-key/value pairs to which the experiment's
target Pod belongs.", "type": "object", "additionalProperties": {"type": "string"}}, "
nodes": {"title": "Nodes", "description": "Specifies the node to which the experiment's
 target Pod belongs. The target Pod can only belong to one node in the configured node
list. If multiple node labels are specified, the node to which the experiment's target
Pod belongs must have all labels specified by this selector.", "type": "array", "items
": {"type": "string"}}, "pods": {"title": "Pods", "description": "Specifies the
namespaces and list of the experiment's target Pods. If you have specified this
selector, Chaos Mesh ignores other configured selectors.", "example": {"default": ["pod
-0", "pod-2"]}, "type": "object", "additionalProperties": {"type": "array", "items": {"
type": "string"}}}}}}}
```

**detailed_param_instructions** for **HTTPChaos** (template embedded dynamically)

The output should be formatted as a JSON instance that conforms to the JSON schema
below.

As an example, for the schema {"properties": {"foo": {"title": "Foo", "description": "a
list of strings", "type": "array", "items": {"type": "string"}}}, "required": ["foo"]}
the object {"foo": ["bar", "baz"]} is a well-formatted instance of the schema. The
object {"properties": {"foo": ["bar", "baz"]}} is not well-formatted.

Here is the output schema:
```
{"properties": {"mode": {"title": "Mode", "description": "Specifies the mode of the
experiment. The mode options include one (selecting a random Pod), all (selecting all
eligible Pods), fixed (selecting a specified number of eligible Pods), fixed-percent (
selecting a specified percentage of Pods from the eligible Pods), and random-max-
percent (selecting the maximum percentage of Pods from the eligible Pods)", "example":
"one", "enum": ["one", "all", "fixed", "fixed-percent", "random-max-percent"], "type":
"string"}, "value": {"title": "Value", "description": "Provides parameters for the mode
configuration, depending on mode. For example, when mode is set to fixed-percent,
value specifies the percentage of Pods.", "example": "1", "type": "string"}, "target":
{"title": "Target", "description": "Specifies whether the target of fault injection is
Request or Response. The target-related fields (replace.path, replace.method, replace.
queries, patch.queries) should be configured at the same time.", "example": "Request",
"enum": ["Request", "Response"], "type": "string"}, "port": {"title": "Port", "
description": "The TCP port that the target service listens on.", "example": 80, "type
": "integer"}, "code": {"title": "Code", "description": "Specifies the status code
responded by target. If not specified, the fault takes effect for all status codes by
default. This configuration is effective only when the 'target' is set to 'Response'",
"example": 200, "type": "integer"}, "path": {"title": "Path", "description": "Specify
the URI path of the target request. Supports Matching wildcards. If not specified, the
fault takes effect on all paths by default.", "example": "/api/*", "type": "string"}, "
method": {"title": "Method", "description": "Specify the HTTP method of the target
request method. If not specified, the fault takes effect for all methods by default.",
"example": "GET", "type": "string"}, "request_headers": {"title": "Request Headers", "
description": "Matches request headers to target.", "example": {"Content-Type": "
application/json"}, "type": "object", "additionalProperties": {"type": "string"}}, "
abort": {"title": "Abort", "description": "Abort fault. Indicates whether to inject the
fault that interrupts the connection.", "default": false, "example": true, "type": "
boolean"}, "delay": {"title": "Delay", "description": "Deplay fault. Specifies the time
for a latency fault.", "default": "0", "example": "10s", "type": "string"}, "replace":
{"title": "Replace", "description": "Replace fault. Specifies replaced contents.", "
allOf": [{"$ref": "#/definitions/Replace"}]}, "patch": {"title": "Patch", "description
": "Patch fault. Specifies patch contents.", "allOf": [{"$ref": "#/definitions/Patch
"}]}}, "required": ["mode", "target", "port"], "definitions": {"Replace": {"title": "
Replace", "type": "object", "properties": {"headers": {"title": "Headers", "description
": "Specifies the key pair used to replace the request headers or response headers.", "
example": {"Content-Type": "application/xml"}, "type": "object", "additionalProperties
": {"type": "string"}}, "body": {"title": "Body", "description": "Specifies request
body or response body to replace the fault (Base64 encoded).", "example": "
eyJmb28iOiAiYmFyIn0K", "type": "string"}, "path": {"title": "Path", "description": "
Specifies the URI path used to replace content.", "example": "/api/v2", "type": "string
"}, "method": {"title": "Method", "description": "Specifies the replaced content of the
HTTP request method.", "example": "DELETE", "type": "string"}, "queries": {"title": "
Queries", "description": "Specifies the replaced key pair of the URI query.", "type": "
array", "items": {"type": "array", "items": {"type": "string"}}}, "code": {"title": "
Code", "description": "Specifies the replaced content of the response status code. This
configuration is effective only when the 'target' is set to 'Response'.", "example":
404, "type": "integer"}}}, "PatchBody": {"title": "PatchBody", "type": "object", "
properties": {"type": {"title": "Type", "description": "Specifies the type of patch
faults of the request body or response body. Currently, it only supports JSON.", "
example": "JSON", "type": "string"}, "value": {"title": "Value", "description": "
Specifies the fault of the request body or response body with patch faults.", "example
": "{\"foo\": \"bar\"}", "type": "string"}}}, "Patch": {"title": "Patch", "type": "
object", "properties": {"headers": {"title": "Headers", "description": "Specifies the
attached key pair of the request headers or response headers with patch faults.", "
example": [["Set-Cookie", "one cookie"]], "type": "array", "items": {"type": "array", "
items": {"type": "string"}}}, "body": {"title": "Body", "description": "Patch body.", "
allOf": [{"$ref": "#/definitions/PatchBody"}]}, "queries": {"title": "Queries", "
description": "Specifies the attached key pair of the URI query with patch faults.", "
example": [["foo", "bar"]], "type": "array", "items": {"type": "array", "items": {"type
": "string"}}}}}}
```

## `detailed_param_instructions` for `StressChaos` (template embedded dynamically)

The output should be formatted as a JSON instance that conforms to the JSON schema
below.

As an example, for the schema {"properties": {"foo": {"title": "Foo", "description": "a list of strings", "type": "array", "items": {"type": "string"}}}, "required": ["foo"]} the object {"foo": ["bar", "baz"]} is a well-formatted instance of the schema. The object {"properties": {"foo": ["bar", "baz"]}} is not well-formatted.

Here is the output schema:
```
{"properties": {"mode": {"title": "Mode", "description": "Specifies the mode of the experiment. The mode options include 'one' (selecting a random Pod), 'all' (selecting all eligible Pods), 'fixed' (selecting a specified number of eligible Pods), 'fixed-percent' (selecting a specified percentage of Pods from the eligible Pods), and 'random-max-percent' (selecting the maximum percentage of Pods from the eligible Pods)", "example": "one", "enum": ["one", "all", "fixed", "fixed-percent", "random-max-percent"], "type": "string"}, "value": {"title": "Value", "description": "Provides parameters for the mode configuration, depending on mode.For example, when mode is set to fixed-percent, value specifies the percentage of Pods.", "example": "1", "type": "string"}, "stressors": {"title": "Stressors", "description": "Specifies the stress of CPU or memory", "dafault": null, "allOf": [{"$ref": "#/definitions/Stressors"}]}, "stressngStressors": {"title": "Stressngstressors", "description": "Specifies the stres-ng parameter to reach richer stress injection", "example": "--clone 2", "type": "string"}, "containerNames": {"title": "Containernames", "description": "Specifies the name of the container into which the fault is injected.", "example": ["nginx"], "type": "array", "items": {"type": "string"}}, "selector": {"title": "Selector", "description": "Specifies the target Pod.", "allOf": [{"$ref": "#/definitions/Selectors"}]}}, "required": ["mode", "selector"], "definitions": {"MemoryStressor": {"title": "MemoryStressor", "type": "object", "properties": {"workers": {"title": "Workers", "description": "Specifies the number of threads that apply memory stress", "example": 1, "type": "integer"}, "size": {"title": "Size", "description": "Specifies the memory size to be occupied or a percentage of the total memory size. The final sum of the occupied memory size is size.", "example": "256MB", "type": "string"}, "oomScoreAdj": {"title": "Oomscoreadj", "description": "Specifies the oom_score_adj of the stress process.", "example": -1000, "type": "integer"}}}, "CPUStressor": {"title": "CPUStressor", "type": "object", "properties": {"workers": {"title": "Workers", "description": "Specifies the number of threads that apply CPU stress", "example": 1, "type": "integer"}, "load": {"title": "Load", "description": "Specifies the percentage of CPU occupied. 0 means that no additional CPU is added, and 100 refers to full load. The final sum of CPU load is workers * load.", "example": 50, "type": "integer"}}}, "Stressors": {"title": "Stressors", "type": "object", "properties": {"memory": {"title": "Memory", "description": "Specifies the memory stress", "allOf": [{"$ref": "#/definitions/MemoryStressor"}]}, "cpu": {"title": "Cpu", "description": "Specifies the CPU stress", "allOf": [{"$ref": "#/definitions/CPUStressor"}]}}}, "SetBasedRequirements": {"title": "SetBasedRequirements", "type": "object", "properties": {"key": {"title": "Key", "description": "Label key", "type": "string"}, "operator": {"title": "Operator", "description": "Select an operator.", "enum": ["In", "NotIn", "Exists", "DoesNotExist"], "type": "string"}, "values": {"title": "Values", "description": "Label values. The values set must be non-empty in the case of In and NotIn.", "type": "array", "items": {"type": "string"}}}, "required": ["key", "operator", "values"]}, "Selectors": {"title": "Selectors", "type": "object", "properties": {"namespaces": {"title": "Namespaces", "description": "Specifies the namespace of the experiment's target Pod. If this selector is None, Chaos Mesh will set it to the namespace of the current Chaos experiment.", "type": "array", "items": {"type": "string"}}, "labelSelectors": {"title": "Labelselectors", "description": "Specifies the label-key/value pairs that the experiment's target Pod must have. If multiple labels are specified, the experiment target must have all the labels specified by this selector.", "type": "object", "additionalProperties": {"type": "string"}}, "expressionSelectors": {"title": "Expressionselectors", "description": "Specifies a set of expressions that define the label's rules to specifiy the experiment's target Pod.", "example": [{"key": "tier", "operator": "In", "values": ["cache"]}, {"key": "environment", "operator": "NotIn", "values": ["dev"]}], "type": "array", "items": {"$ref": "#/definitions/SetBasedRequirements"}}, "annotationSelectors": {"title": "Annotationselectors", "description": "Specifies the annotation-key/value pairs that the experiment's target Pod must have. If multiple annotations are specified, the experiment target must have all annotations specified by this selector.", "type": "object", "additionalProperties": {"type": "string"}}, "fieldSelectors": {"title": "Fieldselectors", "description": "Specifies the field-key/value pairs of the experiment's target Pod. If multiple fields are specified, the experiment target must have all fields set by this selector.", "example": {"metadata.name": "my-pod", "metadata.namespace": "dafault"}, "type": "object", "additionalProperties": {"type": "string"}}, "podPhaseSelectors": {"title": "Podphaseselectors", "description": "Specifies the phase of the experiment's target Pod. If this selector is None, the target Pod's phase is not limited.", "type": "array", "items": {"enum": ["Pending", "Running", "Succeeded", "Failed", "Unknown"], "type": "string"}}, "nodeSelectors": {"title": "Nodeselectors", "description": "Specifies the node-label-key/value pairs to which the experiment's target Pod belongs.", "type": "object", "additionalProperties": {"type": "string"}}, "nodes": {"title": "Nodes", "description": "Specifies the node to which the experiment's target Pod belongs. The target Pod can only belong to one node in the configured node list. If multiple node labels are specified, the node to which the experiment's target Pod belongs must have
```

```
1782
1783   all labels specified by this selector.", "type": "array", "items": {"type": "string"}},
1784    "pods": {"title": "Pods", "description": "Specifies the namespaces and list of the
       experiment's target Pods. If you have specified this selector, Chaos Mesh ignores other
1785    configured selectors.", "example": {"default": ["pod-0", "pod-2"]}, "type": "object",
       "additionalProperties": {"type": "array", "items": {"type": "string"}}}}}}}
1786   ```
1787
1788
```

**detailed_param_instructions** for **IOChaos** (template embedded dynamically)

```
The output should be formatted as a JSON instance that conforms to the JSON schema
below.

As an example, for the schema {"properties": {"foo": {"title": "Foo", "description": "a
 list of strings", "type": "array", "items": {"type": "string"}}}, "required": ["foo"]}
the object {"foo": ["bar", "baz"]} is a well-formatted instance of the schema. The
object {"properties": {"foo": ["bar", "baz"]}} is not well-formatted.

Here is the output schema:
```
```
{"properties": {"action": {"title": "Action", "description": "Indicates the specific
type of faults. Only latency, fault, attrOverride, and mistake are supported.", "
example": "latency", "enum": ["latency", "fault", "attrOverride", "mistake"], "type": "
string"}, "mode": {"title": "Mode", "description": "Specifies the mode of the
experiment. The mode options include one (selecting a random Pod), all (selecting all
eligible Pods), fixed (selecting a specified number of eligible Pods), fixed-percent (
selecting a specified percentage of Pods from the eligible Pods), and random-max-
percent (selecting the maximum percentage of Pods from the eligible Pods)", "example":
"one", "enum": ["one", "all", "fixed", "fixed-percent", "random-max-percent"], "type":
"string"}, "selector": {"title": "Selector", "description": "Specifies the target Pod
.", "allOf": [{"$ref": "#/definitions/Selectors"}]}, "value": {"title": "Value", "
description": "Provides parameters for the mode configuration, depending on mode. For
example, when mode is set to fixed-percent, value specifies the percentage of Pods.", "
example": "1", "type": "string"}, "volumePath": {"title": "Volumepath", "description":
"The mount point of volume in the target container. Must be the root directory of the
mount.", "example": "/var/run/etcd", "type": "string"}, "path": {"title": "Path", "
description": "The valid range of fault injections, either a wildcard or a single file.
 If not specified, the fault is valid for all files by default", "example": "/var/run/
etcd/*/", "type": "string"}, "methods": {"title": "Methods", "description": "Type of
the file system call that requires injecting fault. Supported method types: ['lookup',
'forget', 'getattr', 'setattr', 'readlink', 'mknod', 'mkdir', 'unlink', 'rmdir', '
symlink', 'rename', 'link', 'open', 'read', 'write', 'flush', 'release', 'fsync', '
opendir', 'readdir', 'releasedir', 'fsyncdir', 'statfs', 'setxattr', 'getxattr', '
listxattr', 'removexattr', 'access', 'create', 'getlk', 'setlk', 'bmap']. All Types by
default.", "example": ["READ"], "type": "array", "items": {"type": "string"}}, "percent
": {"title": "Percent", "description": "Probability of failure per operation, in %.", "
default": 100, "example": 100, "type": "integer"}, "containerNames": {"title": "
Containernames", "description": "Specifies the name of the container into which the
fault is injected.", "type": "array", "items": {"type": "string"}}, "deplay": {"title":
 "Deplay", "description": "Specify when the 'action' is set to 'latency'. Specific
delay time.", "type": "string"}, "errno": {"title": "Errno", "description": "Specify
when the 'action' is set to 'fault'. Returned error number: 1: Operation not permitted,
 2: No such file or directory, 5: I/O error, 6: No such device or address, 12: Out of
memory, 16: Device or resource busy, 17: File exists, 20: Not a directory, 22: Invalid
argument, 24: Too many open files, 28: No space left on device", "type": "integer"}, "
attr": {"title": "Attr", "description": "Specify when the 'action' is set to '
attrOverride'. Specific property override rules.", "allOf": [{"$ref": "#/definitions/
AttrOverrideSpec"}]}, "mistake": {"title": "Mistake", "description": "Specify when the
'action' is set to 'mistake'. Specific error rules.", "allOf": [{"$ref": "#/definitions
/MistakeSpec"}]}}, "required": ["action", "mode", "volumePath", "attr", "mistake"], "
definitions": {"SetBasedRequirements": {"title": "SetBasedRequirements", "type": "
object", "properties": {"key": {"title": "Key", "description": "Label key", "type": "
string"}, "operator": {"title": "Operator", "description": "Select an operator.", "enum
": ["In", "NotIn", "Exists", "DoesNotExist"], "type": "string"}, "values": {"title": "
Values", "description": "Label values. The values set must be non-empty in the case of
In and NotIn.", "type": "array", "items": {"type": "string"}}}, "required": ["key", "
operator", "values"]}, "Selectors": {"title": "Selectors", "type": "object", "
properties": {"namespaces": {"title": "Namespaces", "description": "Specifies the
namespace of the experiment's target Pod. If this selector is None, Chaos Mesh will set
 it to the namespace of the current Chaos experiment.", "type": "array", "items": {"
type": "string"}}, "labelSelectors": {"title": "Labelselectors", "description": "
Specifies the label-key/value pairs that the experiment's target Pod must have. If
multiple labels are specified, the experiment target must have all the labels specified
 by this selector.", "type": "object", "additionalProperties": {"type": "string"}}, "
expressionSelectors": {"title": "Expressionselectors", "description": "Specifies a set
of expressions that define the label's rules to specifiy the experiment's target Pod.",
```

```
   "example": [{"key": "tier", "operator": "In", "values": ["cache"]}, {"key": "
environment", "operator": "NotIn", "values": ["dev"]}], "type": "array", "items": {"
$ref": "#/definitions/SetBasedRequirements"}}, "annotationSelectors": {"title": "
Annotationselectors", "description": "Specifies the annotation-key/value pairs that the
 experiment's target Pod must have. If multiple annotations are specified, the
experiment target must have all annotations specified by this selector.", "type": "
object", "additionalProperties": {"type": "string"}}, "fieldSelectors": {"title": "
Fieldselectors", "description": "Specifies the field-key/value pairs of the experiment'
s target Pod. If multiple fields are specified, the experiment target must have all
fields set by this selector.", "example": {"metadata.name": "my-pod", "metadata.
namespace": "dafault"}, "type": "object", "additionalProperties": {"type": "string"}},
"podPhaseSelectors": {"title": "Podphaseselectors", "description": "Specifies the phase
 of the experiment's target Pod. If this selector is None, the target Pod's phase is
not limited.", "type": "array", "items": {"enum": ["Pending", "Running", "Succeeded", "
Failed", "Unknown"], "type": "string"}}, "nodeSelectors": {"title": "Nodeselectors", "
description": "Specifies the node-label-key/value pairs to which the experiment's
target Pod belongs.", "type": "object", "additionalProperties": {"type": "string"}}, "
nodes": {"title": "Nodes", "description": "Specifies the node to which the experiment's
 target Pod belongs. The target Pod can only belong to one node in the configured node
list. If multiple node labels are specified, the node to which the experiment's target
Pod belongs must have all labels specified by this selector.", "type": "array", "items
": {"type": "string"}}, "pods": {"title": "Pods", "description": "Specifies the
namespaces and list of the experiment's target Pods. If you have specified this
selector, Chaos Mesh ignores other configured selectors.", "example": {"default": ["pod
-0", "pod-2"]}, "type": "object", "additionalProperties": {"type": "array", "items": {"
type": "string"}}}}}, "TimeSpec": {"title": "TimeSpec", "type": "object", "properties"
: {"sec": {"title": "Sec", "description": "Timestamp in seconds. Specify either sec or
nsec.", "type": "integer"}, "nsec": {"title": "Nsec", "description": "Timestamp in
nanoseconds. Specify either sec or nsec.", "type": "integer"}}}, "AttrOverrideSpec": {"
title": "AttrOverrideSpec", "type": "object", "properties": {"ino": {"title": "Ino", "
description": "ino number", "type": "integer"}, "size": {"title": "Size", "description
": "File size", "type": "integer"}, "blocks": {"title": "Blocks", "description": "
Number of blocks that the file uses", "type": "integer"}, "atime": {"title": "Atime", "
description": "Last access time", "allOf": [{"$ref": "#/definitions/TimeSpec"}]}, "
mtime": {"title": "Mtime", "description": "Last modified time", "allOf": [{"$ref": "#/
definitions/TimeSpec"}]}, "ctime": {"title": "Ctime", "description": "Last status
change time", "allOf": [{"$ref": "#/definitions/TimeSpec"}]}, "kind": {"title": "Kind",
 "description": "File type, see fuser::FileType", "type": "string"}, "perm": {"title":
"Perm", "description": "File permissions in decimal", "type": "integer"}, "nlink": {"
title": "Nlink", "description": "Number of hard links", "type": "integer"}, "uid": {"
title": "Uid", "description": "User ID of the owner", "type": "integer"}, "gid": {"
title": "Gid", "description": "Group ID of the owner", "type": "integer"}, "rdev": {"
title": "Rdev", "description": "Device ID", "type": "integer"}}}, "MistakeSpec": {"
title": "MistakeSpec", "type": "object", "properties": {"filling": {"title": "Filling",
 "description": "The wrong data to be filled. Only zero (fill 0) or random (fill random
 bytes) are supported.", "type": "string"}, "maxOccurrences": {"title": "Maxoccurrences
", "description": "Maximum number of errors in each operation.", "example": 1, "type":
"integer"}, "maxLength": {"title": "Maxlength", "description": "Maximum length of each
error (in bytes).", "example": 1, "type": "integer"}}, "required": ["filling", "
maxOccurrences", "maxLength"]}}}
```



**detailed_param_instructions** for **TimeChaos** (template embedded dynamically)



The output should be formatted as a JSON instance that conforms to the JSON schema
below.

As an example, for the schema {"properties": {"foo": {"title": "Foo", "description": "a
 list of strings", "type": "array", "items": {"type": "string"}}}, "required": ["foo"]}
the object {"foo": ["bar", "baz"]} is a well-formatted instance of the schema. The
object {"properties": {"foo": ["bar", "baz"]}} is not well-formatted.

Here is the output schema:

```
{"properties": {"timeOffset": {"title": "Timeoffset", "description": "Specifies the
length of time offset.", "example": "-5m", "type": "string"}, "clockIds": {"title": "
Clockids", "description": "Specifies the ID of clock that will be offset. See the
clock_gettime documentation for details.", "default": ["CLOCK_REALTIME"], "example": ["
CLOCK_REALTIME", "CLOCK_MONOTONIC"], "type": "array", "items": {"type": "string"}}, "
mode": {"title": "Mode", "description": "Specifies the mode of the experiment. The mode
 options include 'one' (selecting a random Pod), 'all' (selecting all eligible Pods), '
fixed' (selecting a specified number of eligible Pods), 'fixed-percent' (selecting a
specified percentage of Pods from the eligible Pods), and 'random-max-percent' (
selecting the maximum percentage of Pods from the eligible Pods)", "example": "one", "
enum": ["one", "all", "fixed", "fixed-percent", "random-max-percent"], "type": "string
```

```
"}, "value": {"title": "Value", "description": "Provides parameters for the mode
configuration, depending on mode. For example, when mode is set to fixed-percent, value
 specifies the percentage of Pods.", "example": "1", "type": "string"}, "containerNames
": {"title": "Containernames", "description": "Specifies the name of the container into
 which the fault is injected.", "example": ["nginx"], "type": "array", "items": {"type
": "string"}}, "selector": {"title": "Selector", "description": "Specifies the target
Pod.", "example": null, "allOf": [{"$ref": "#/definitions/Selectors"}]}}, "required":
["timeOffset", "mode", "selector"], "definitions": {"SetBasedRequirements": {"title": "
SetBasedRequirements", "type": "object", "properties": {"key": {"title": "Key", "
description": "Label key", "type": "string"}, "operator": {"title": "Operator", "
description": "Select an operator.", "enum": ["In", "NotIn", "Exists", "DoesNotExist"],
 "type": "string"}, "values": {"title": "Values", "description": "Label values. The
values set must be non-empty in the case of In and NotIn.", "type": "array", "items":
{"type": "string"}}}, "required": ["key", "operator", "values"]}, "Selectors": {"title
": "Selectors", "type": "object", "properties": {"namespaces": {"title": "Namespaces",
"description": "Specifies the namespace of the experiment's target Pod. If this
selector is None, Chaos Mesh will set it to the namespace of the current Chaos
experiment.", "type": "array", "items": {"type": "string"}}, "labelSelectors": {"title
": "Labelselectors", "description": "Specifies the label-key/value pairs that the
experiment's target Pod must have. If multiple labels are specified, the experiment
target must have all the labels specified by this selector.", "type": "object", "
additionalProperties": {"type": "string"}}, "expressionSelectors": {"title": "
Expressionselectors", "description": "Specifies a set of expressions that define the
label's rules to specifiy the experiment's target Pod.", "example": [{"key": "tier", "
operator": "In", "values": ["cache"]}, {"key": "environment", "operator": "NotIn", "
values": ["dev"]}], "type": "array", "items": {"$ref": "#/definitions/
SetBasedRequirements"}}, "annotationSelectors": {"title": "Annotationselectors", "
description": "Specifies the annotation-key/value pairs that the experiment's target
Pod must have. If multiple annotations are specified, the experiment target must have
all annotations specified by this selector.", "type": "object", "additionalProperties":
 {"type": "string"}}, "fieldSelectors": {"title": "Fieldselectors", "description": "
Specifies the field-key/value pairs of the experiment's target Pod. If multiple fields
are specified, the experiment target must have all fields set by this selector.", "
example": {"metadata.name": "my-pod", "metadata.namespace": "dafault"}, "type": "object
", "additionalProperties": {"type": "string"}}, "podPhaseSelectors": {"title": "
Podphaseselectors", "description": "Specifies the phase of the experiment's target Pod.
 If this selector is None, the target Pod's phase is not limited.", "type": "array", "
items": {"enum": ["Pending", "Running", "Succeeded", "Failed", "Unknown"], "type": "
string"}}, "nodeSelectors": {"title": "Nodeselectors", "description": "Specifies the
node-label-key/value pairs to which the experiment's target Pod belongs.", "type": "
object", "additionalProperties": {"type": "string"}}, "nodes": {"title": "Nodes", "
description": "Specifies the node to which the experiment's target Pod belongs. The
target Pod can only belong to one node in the configured node list. If multiple node
labels are specified, the node to which the experiment's target Pod belongs must have
all labels specified by this selector.", "type": "array", "items": {"type": "string"}},
 "pods": {"title": "Pods", "description": "Specifies the namespaces and list of the
experiment's target Pods. If you have specified this selector, Chaos Mesh ignores other
 configured selectors.", "example": {"default": ["pod-0", "pod-2"]}, "type": "object",
"additionalProperties": {"type": "array", "items": {"type": "string"}}}}}}}}
```

### B.1.3 EXPERIMENT

**# 2-0: Agent for determining time schedule**

**System:**
You are a helpful AI assistant for Chaos Engineering.
Given k8s manifests that define a network system, its steady states, and faults that
may affect the steady states in the system, you will design a Chaos Engineering
experiment for them.
First, you will determine the time schedule for the Chaos Engineering experiment.
Always keep the following rules:
- The experiment is divided into three phases: pre-validation, fault-injection, and
post-validation phases: pre-validation to ensure that the system satisfies the steady
states fault injection; fault-injection to observe the system's behavior during fault
injection; post-validation to ensure that the system has returned to its steady states
after fault injection.
- The output should be formatted as a JSON instance that conforms to the JSON schema
below.

As an example, for the schema {\"properties\": {\"foo\": {\"title\": \"Foo\", \"
description\": \"a list of strings\", \"type\": \"array\", \"items\": {\"type\": \"
string\"}}}, \"required\": [\"foo\"]}\nthe object {\"foo\": [\"bar\", \"baz\"]} is a

```
well-formatted instance of the schema. The object {\"properties\": {\"foo\": [\"bar\",
\"baz\"]}} is not well-formatted.

Here is the output schema:
```
{\"properties\": {\"thought\": {\"title\": \"Thought\", \"describe\": \"Think about the
 total time and the reasonable time allocation for each phase that you are about to
design, and explain your thought process in detail.\", \"type\": \"string\"}, \"
total_time\": {\"title\": \"Total Time\", \"description\": \"Total time of the entire
chaos experiment. total_time should equal to the sum of pre_validation_time,
fault_injection_time, and post_validation_time.\", \"example\": \"10m\", \"type\": \"
string\"}, \"pre_validation_time\": {\"title\": \"Pre Validation Time\", \"description
\": \"Total time of validation before fault injection.\", \"example\": \"2m\", \"type
\": \"string\"}, \"fault_injection_time\": {\"title\": \"Fault Injection Time\", \"
description\": \"Total time of fault injection.\", \"example\": \"6m\", \"type\": \"
string\"}, \"post_validation_time\": {\"title\": \"Post Validation Time\", \"
description\": \"Total time of validation after fault injection.\", \"example\": \"2m
\", \"type\": \"string\"}}, \"required\": [\"thought\", \"total_time\", \"
pre_validation_time\", \"fault_injection_time\", \"post_validation_time\"]}
```
```

**Human:**
```
# Here is the overview of my system:
{user_input2}

# Steady states of my system:
{steady_states}

# A fault scenario that may occur in my system and may affect the steady states:
{detailed_fault_scenario}

# Please follow the instructions below regarding Chaos Engineering as necessary:
{ce_instructions}

Now, please plan a Chaos Engineering experiment to check the network system's
resiliency that the steady states are remained during fault injection.
```

## Example text embedded to `detailed_fault_scenario`

```
An assumed fault scenario is as follows:
- Event: Cyber Attack Simulation\n- Used Chaos Engineering tool: Chaos Mesh
- Faults to simulate the event: [[Fault(name='PodChaos', name_id=0, params={'action': '
pod-kill', 'mode': 'one', 'selector': {'namespaces': ['default'], 'labelSelectors': {'
app': 'example'}}})], [Fault(name='NetworkChaos', name_id=0, params={'action': 'delay',
 'mode': 'all', 'selector': {'namespaces': ['default'], 'labelSelectors': {'app': '
example'}}, 'direction': 'to', 'delay': {'latency': '100ms', 'jitter': '10ms'}})]]
- Description: Given the system's weaknesses, a cyber attack targeting the web server
could be highly impactful. The Pod's restart policy set to 'Never' and the single Pod
deployment without redundancy are critical vulnerabilities. If the Pod fails, it will
not restart, leading to downtime, and the lack of redundancy means there is no backup
to handle traffic. To simulate a cyber attack, we can inject faults that exploit these
weaknesses. First, we will use PodChaos to simulate a Pod failure, which will test the
system's ability to maintain the 'example-pod-running-state'. Since the Pod will not
restart automatically, this will directly impact the steady state. Next, we will use
NetworkChaos to simulate network latency, which will test the system's ability to
maintain the 'example-service-http-response-state'. This sequence simulates a cyber
attack where the Pod is targeted first, followed by network disruptions, revealing the
system's vulnerabilities in handling such events.
```

## # 2-1: Agent for scheduling each experiment phase (pre-validation, failure-injection, and post-validation phases)

**System:**
```
You are a helpful AI assistant for Chaos Engineering.
Given k8s manifests that define a network system, its steady states, and faults that
may affect the steady states in the system, you will design a Chaos Engineering
experiment for them.
The experiment is divided into three phases: pre-validation, fault-injection, and post-
validation phases: pre-validation to ensure that the system satisfies the steady states
 fault injection; fault-injection to observe the system's behavior during fault
```

```
injection; post-validation to ensure that the system has returned to its steady states
after fault injection.
Here, you will detail the {phase_name}.
Always keep the following rules:
- {phase_planning_instructions}

Human:
# Here is the overview of my system:
{user_input}

# Steady states of my system:
{steady_states}

# A fault scenario that may occur in my system and may affect the steady states:
{detailed_fault_scenario}

# Please follow the instructions below regarding Chaos Engineering as necessary:
{ce_instructions}

Now, please detail the {phase_name}. Note that the phase's total time is {
phase_total_time}.
```

**Example text embedded to `phase_name`**

```
pre-validation phase
```

**Example text embedded to `phase_total_time`**

```
10s
```

**`phase_planning_instructions` for the pre-validation and post-validation phases**

```
The output should be formatted as a JSON instance that conforms to the JSON schema
below.

As an example, for the schema {\"properties\": {\"foo\": {\"title\": \"Foo\", \"
description\": \"a list of strings\", \"type\": \"array\", \"items\": {\"type\": \"
string\"}}}, \"required\": [\"foo\"]}\nthe object {\"foo\": [\"bar\", \"baz\"]} is a
well-formatted instance of the schema. The object {\"properties\": {\"foo\": [\"bar\",
\"baz\"]}} is not well-formatted.

Here is the output schema:
```
{\"properties\": {\"thought\": {\"title\": \"Thought\", \"description\": \"Describe in
detail the timeline for when each fault injection and each unit test (for verifying
steady-state) will be executed. For example, explain which fault injections/unit tests
will be executed simultaneously, and whether certain fault injections/unit tests will
be executed at staggered timings. Additionally, explain the thought process that led
you to this approach.\", \"type\": \"string\"}, \"unit_tests\": {\"title\": \"Unit
Tests\", \"description\": \"The list of unit test schedule.\", \"type\": \"array\", \"
items\": {\"$ref\": \"#/definitions/UnitTest\"}}}, \"required\": [\"thought\", \"
unit_tests\"], \"definitions\": {\"UnitTest\": {\"title\": \"UnitTest\", \"type\": \"
object\", \"properties\": {\"name\": {\"title\": \"Name\", \"description\": \"Steady
state name to be verified by a unit test.\", \"type\": \"string\"}, \"grace_period\":
{\"title\": \"Grace Period\", \"description\": \"Time elapsed from the start of the
current phase to the beginning of the unit test.\", \"example\": \"0s\", \"type\": \"
string\"}, \"duration\": {\"title\": \"Duration\", \"description\": \"Duration of the
unit test. (grace_period + duration) should not exceed the current phase's total time
.\", \"example\": \"2m\", \"type\": \"string\"}}, \"required\": [\"name\", \"
grace_period\", \"duration\"]}}}
```
```

**phase_planning_instructions** for the fault-injection phases

```
The output should be formatted as a JSON instance that conforms to the JSON schema
below.

As an example, for the schema {\"properties\": {\"foo\": {\"title\": \"Foo\", \"
description\": \"a list of strings\", \"type\": \"array\", \"items\": {\"type\": \"
string\"}}}, \"required\": [\"foo\"]}\nthe object {\"foo\": [\"bar\", \"baz\"]} is a
well-formatted instance of the schema. The object {\"properties\": {\"foo\": [\"bar\",
\"baz\"]}} is not well-formatted.

Here is the output schema:
```
{\"properties\": {\"thought\": {\"title\": \"Thought\", \"description\": \"Describe in
detail the timeline for when each fault injection and each unit test (for verifying
steady-state) will be executed. For example, explain which fault injections/unit tests
will be executed simultaneously, and whether certain fault injections/unit tests will
be executed at staggered timings. Additionally, explain the thought process that led
you to this approach.\", \"type\": \"string\"}, \"fault_injection\": {\"title\": \"
Fault Injection\", \"description\": \"The list of fault injection schedules.\", \"type
\": \"array\", \"items\": {\"$ref\": \"#/definitions/FaultInjection\"}}, \"unit_tests
\": {\"title\": \"Unit Tests\", \"description\": \"The list of unit test schedule.\",
\"type\": \"array\", \"items\": {\"$ref\": \"#/definitions/UnitTest\"}}}, \"required\":
 [\"thought\", \"fault_injection\", \"unit_tests\"], \"definitions\": {\"FaultInjection
\": {\"title\": \"FaultInjection\", \"type\": \"object\", \"properties\": {\"name\":
{\"title\": \"Name\", \"description\": \"Select a fault type from [\\\"PodChaos\\\",
\\\"NetworkChaos\\\", \\\"DNSChaos\\\", \\\"HTTPChaos\\\", \\\"StressChaos\\\", \\\"
IOChaos\\\", \\\"TimeChaos\\\"]\", \"enum\": [\"PodChaos\", \"NetworkChaos\", \"
DNSChaos\", \"HTTPChaos\", \"StressChaos\", \"IOChaos\", \"TimeChaos\"], \"type\": \"
string\"}, \"name_id\": {\"title\": \"Name Id\", \"description\": \"An identifier to
prevent name conflicts when the same Fault appears. Assign numbers starting from 0 in
sequential order to prevent name conflicts.\", \"type\": \"integer\"}, \"grace_period
\": {\"title\": \"Grace Period\", \"description\": \"Time elapsed from the start of the
 current phase to the beginning of the fault injection.\", \"example\": \"0s\", \"type
\": \"string\"}, \"duration\": {\"title\": \"Duration\", \"description\": \"Duration of
 the unit test. (grace_period + duration) should not exceed the current phase's total
time.\", \"example\": \"2m\", \"type\": \"string\"}}, \"required\": [\"name\", \"
name_id\", \"grace_period\", \"duration\"]}, \"UnitTest\": {\"title\": \"UnitTest\", \"
type\": \"object\", \"properties\": {\"name\": {\"title\": \"Name\", \"description\":
\"Steady state name to be verified by a unit test.\", \"type\": \"string\"}, \"
grace_period\": {\"title\": \"Grace Period\", \"description\": \"Time elapsed from the
start of the current phase to the beginning of the unit test.\", \"example\": \"0s\",
\"type\": \"string\"}, \"duration\": {\"title\": \"Duration\", \"description\": \"
Duration of the unit test. (grace_period + duration) should not exceed the current
phase's total time.\", \"example\": \"2m\", \"type\": \"string\"}}, \"required\": [\"
name\", \"grace_period\", \"duration\"]}}}
```

# 2-2: Agent for summarizing the planned experiment

**System:**
You are a helpful AI assistant for Chaos Engineering.
Given a Chaos-Engineering-experiment plan, you will summarize it in detail according to
 the following rules:
- In each phase, describe in detail the timeline for when each fault injection/unit
test (for verifying steady-state) will be executed. For example, summarize which fault
injections/unit tests will be executed simultaneously, and whether certain fault
injections/unit tests will be executed at staggered timings.
- Be sure to specify both each fault injection/unit test and their corresponding
workflow names.
- When explaining the timeline, provide a detailed description using specific values
for duration, grace period, etc. Rephrase the specific values in a way that everyone
can easily understand.
- The meanings of each value are as follows:
  - Grace Period: Time elapsed from the start of the current phase to the beginning of
  the fault injection/unit test.
  - Duration: Duration of the fault injection/unit test. (grace_period + duration)
  should not exceed the corresponding phase's total time.
- Never output bullet points.
- The output should be formatted as a JSON instance that conforms to the JSON schema
below.
As an example, for the schema {\"properties\": {\"foo\": {\"title\": \"Foo\", \"
description\": \"a list of strings\", \"type\": \"array\", \"items\": {\"type\": \"
string\"}}}, \"required\": [\"foo\"]}\nthe object {\"foo\": [\"bar\", \"baz\"]} is a

```
well-formatted instance of the schema. The object {\"properties\": {\"foo\": [\"bar\",
\"baz\"]}} is not well-formatted.

Here is the output schema:
```
{\"properties\": {\"summary\": {\"title\": \"Summary\", \"description\": \"The summary
of the given Chaos-Engineering-experiment plan.\", \"type\": \"string\"}}, \"required
\": [\"summary\"]}
```

Human:
# Here is my Chaos-Engineering-experiment plan:
## Time Schedule
{time_schedule_overview}

## Pre-validation Phase
{pre_validation_overview}

## Fault-injection Phase
{fault_injection_overview}

## Post-validation phase
{post_validation_overview}

Please summarize the above plan.
```

**Example text embedded to `time_schedule_overview`**

```
Given the constraints of the experiment needing to be completed within 1 minute, we
need to carefully allocate time to each phase to ensure that we can effectively
validate the system's steady states before and after the fault injection, as well as
observe the system's behavior during the fault injection. The pre-validation phase is
crucial to establish a baseline that the system is in its expected steady state before
any faults are introduced. The fault injection phase is where we introduce the chaos to
 observe how the system behaves under stress. Finally, the post-validation phase is
necessary to ensure that the system returns to its steady state after the faults are
removed. Given the short total time of 1 minute, a reasonable allocation could be 10
seconds for pre-validation, 40 seconds for fault injection, and 10 seconds for post-
validation. This allocation allows us to have a brief but sufficient observation period
 for each phase, ensuring that we can gather meaningful insights from the experiment.
```

**Example text embedded to `pre_validation_overview`**

```
In the pre-validation phase, we need to ensure that the system is in its expected
steady states before we proceed with fault injection. Given the constraints, we have 10
 seconds to perform these checks. We have two steady states to verify: the 'example-pod
-running-state' and the 'example-service-http-response-state'.

The 'example-pod-running-state' requires us to check that the Pod is in the 'Running'
state at least 90% of the time. We will use the provided Python script to verify this.
Since the script checks the Pod status every second, we can run it for 5 seconds to
gather sufficient data for validation.

The 'example-service-http-response-state' requires us to ensure that 95% of HTTP
requests return a 200 OK status. We will use the K6 script to simulate HTTP requests to
 the service. The script is configured to run for 5 seconds with 5 virtual users, which
 should provide enough data to validate this steady state.

Both unit tests will be executed simultaneously to maximize the use of the 10-second
window. This approach ensures that we efficiently validate both steady states within
the given time constraint, allowing us to proceed confidently to the fault injection
phase.
- Verified Steady State #0: ```example-pod-running-state```
  - Workflow Name: ```pre-unittest-example-pod-running-state```
  - Grace Period: ```0s```
  - Duration: ```5s```
- Verified Steady State #1: ```example-service-http-response-state```
  - Workflow Name: ```pre-unittest-example-service-http-response-state```
  - Grace Period: ```0s```
  - Duration: ```5s```
```

---

**Example text embedded to `fault_injection_overview`**

In this fault-injection phase, we aim to simulate a cyber attack by injecting two types
 of faults: PodChaos and NetworkChaos. The goal is to observe how these faults impact
the system's steady states. Given the 40-second time constraint, we will stagger the
fault injections to maximize the observation of their effects.

First, we will inject the PodChaos fault to simulate a Pod failure. This will directly
test the 'example-pod-running-state' steady state, as the Pod's restart policy is set
to 'Never', meaning it will not automatically recover. We will start this fault
injection immediately at the beginning of the phase and let it run for 20 seconds.

Simultaneously, we will run the unit test for the 'example-pod-running-state' to verify
 the impact of the PodChaos fault on the Pod's availability.\n\nAfter 20 seconds, we
will inject the NetworkChaos fault to simulate network latency. This will test the '
example-service-http-response-state' steady state by introducing delays in HTTP
responses. We will let this fault run for the remaining 20 seconds of the phase.

During the NetworkChaos fault injection, we will also run the unit test for the '
example-service-http-response-state' to verify the impact of network latency on the
Service's ability to respond to HTTP requests.

This staggered approach allows us to isolate the effects of each fault on the
respective steady states, providing clear insights into the system's vulnerabilities
and resilience.
- Verified Steady State #0: ```example-pod-running-state```
  - Workflow Name: ```fault-unittest-example-pod-running-state```
  - Grace Period: ```0s```
  - Duration: ```20s```
- Verified Steady State #1: ```example-service-http-response-state```
  - Workflow Name: ```fault-unittest-example-service-http-response-state```
  - Grace Period: ```20s```
  - Duration: ```20s```

- Injected Faults #0: ```PodChaos```
  - Workflow Name: ```fault-podchaos```
  - Grace Period: ```0s```
  - Duration: ```20s```

- Injected Faults #1: ```NetworkChaos```
  - Workflow Name: ```fault-networkchaos```
  - Grace Period: ```20s```
  - Duration: ```20s```

---

**Example text embedded to `post_validation_overview`**

In the post-validation phase, we need to ensure that the system has returned to its
steady states after the fault injection. Given the 10-second total time for this phase,
 we will conduct unit tests to verify each steady state. The tests will be executed
sequentially due to the short duration, ensuring that each steady state is verified
independently. The first test will check the 'example-pod-running-state', ensuring the
Pod is running. The second test will verify the 'example-service-http-response-state',
ensuring the Service responds with a 200 OK status. Each test will have a brief grace
period to allow the system to stabilize after the fault injection, followed by a short
duration to perform the checks. This approach ensures that both steady states are
validated within the available time, confirming the system's recovery.
- Verified Steady State #0: ```example-pod-running-state```
  - Workflow Name: ```post-unittest-example-pod-running-state```
  - Grace Period: ```1s```
  - Duration: ```4s```

- Verified Steady State #1: ```example-service-http-response-state```
  - Workflow Name: ```post-unittest-example-service-http-response-state```
  - Grace Period: ```5s```
  - Duration: ```4s```

---

# 2-3: Agent for adjusting a failure scope

**System:**
You are a helpful AI assistant for Chaos Engineering.

Given a previous K8s manifests, a Chaos-Engineering-experiment plan for it, and the
current K8s manifests, you will determine whether we need to adujst the scope of fault
injections for the current K8s manifests.
Always keep the following rules:
- Cosider how you must change or keep the scope (i.e., target) of the fault injecttion
comparing the previous K8s manifests and the current K8s manifests.
- You only make minor adjustments related to resource changes, metadata change, etc, so
 NEVER make any scope changes that alter the original goal of the chaos experiment.
- The output should be formatted as a JSON instance that conforms to the JSON schema
below.

As an example, for the schema {\"properties\": {\"foo\": {\"title\": \"Foo\", \"
description\": \"a list of strings\", \"type\": \"array\", \"items\": {\"type\": \"
string\"}}}, \"required\": [\"foo\"]}\nthe object {\"foo\": [\"bar\", \"baz\"]} is a
well-formatted instance of the schema. The object {\"properties\": {\"foo\": [\"bar\",
\"baz\"]}} is not well-formatted.

Here is the output schema:
```
{\"properties\": {\"thought\": {\"title\": \"Thought\", \"description\": \"Describe why
 you need to change/keep the scope of the fault injection for the current K8s manifests
.\", \"type\": \"string\"}, \"selector\": {\"title\": \"Selector\", \"description\": \"
Adjust the scope (target) of the fault injection comparing the differeneces between the
 current and previous manifests. \", \"allOf\": [{\"$ref\": \"#/definitions/Selectors
\"}]}}, \"required\": [\"thought\", \"selector\"], \"definitions\": {\"
SetBasedRequirements\": {\"title\": \"SetBasedRequirements\", \"type\": \"object\", \"
properties\": {\"key\": {\"title\": \"Key\", \"description\": \"Label key\", \"type\":
\"string\"}, \"operator\": {\"title\": \"Operator\", \"description\": \"Select an
operator.\", \"enum\": [\"In\", \"NotIn\", \"Exists\", \"DoesNotExist\"], \"type\": \"
string\"}, \"values\": {\"title\": \"Values\", \"description\": \"Label values. The
values set must be non-empty in the case of In and NotIn.\", \"type\": \"array\", \"
items\": {\"type\": \"string\"}}}, \"required\": [\"key\", \"operator\", \"values\"]},
\"Selectors\": {\"title\": \"Selectors\", \"type\": \"object\", \"properties\": {\"
namespaces\": {\"title\": \"Namespaces\", \"description\": \"Specifies the namespace of
 the experiment's target Pod. If this selector is None, Chaos Mesh will set it to the
namespace of the current Chaos experiment.\", \"type\": \"array\", \"items\": {\"type
\": \"string\"}}, \"labelSelectors\": {\"title\": \"Labelselectors\", \"description\":
\"Specifies the label-key/value pairs that the experiment's target Pod must have. If
multiple labels are specified, the experiment target must have all the labels specified
 by this selector.\", \"type\": \"object\", \"additionalProperties\": {\"type\": \"
string\"}}, \"expressionSelectors\": {\"title\": \"Expressionselectors\", \"description
\": \"Specifies a set of expressions that define the label's rules to specifiy the
experiment's target Pod.\", \"example\": [{\"key\": \"tier\", \"operator\": \"In\", \"
values\": [\"cache\"]}, {\"key\": \"environment\", \"operator\": \"NotIn\", \"values\":
 [\"dev\"]}], \"type\": \"array\", \"items\": {\"$ref\": \"#/definitions/
SetBasedRequirements\"}}, \"annotationSelectors\": {\"title\": \"Annotationselectors\",
 \"description\": \"Specifies the annotation-key/value pairs that the experiment's
target Pod must have. If multiple annotations are specified, the experiment target must
 have all annotations specified by this selector.\", \"type\": \"object\", \"
additionalProperties\": {\"type\": \"string\"}}, \"fieldSelectors\": {\"title\": \"
Fieldselectors\", \"description\": \"Specifies the field-key/value pairs of the
experiment's target Pod. If multiple fields are specified, the experiment target must
have all fields set by this selector.\", \"example\": {\"metadata.name\": \"my-pod\",
\"metadata.namespace\": \"dafault\"}, \"type\": \"object\", \"additionalProperties\":
{\"type\": \"string\"}}, \"podPhaseSelectors\": {\"title\": \"Podphaseselectors\", \"
description\": \"Specifies the phase of the experiment's target Pod. If this selector
is None, the target Pod's phase is not limited.\", \"type\": \"array\", \"items\": {\"
enum\": [\"Pending\", \"Running\", \"Succeeded\", \"Failed\", \"Unknown\"], \"type\":
\"string\"}}, \"nodeSelectors\": {\"title\": \"Nodeselectors\", \"description\": \"
Specifies the node-label-key/value pairs to which the experiment's target Pod belongs
.\", \"type\": \"object\", \"additionalProperties\": {\"type\": \"string\"}}, \"nodes
\": {\"title\": \"Nodes\", \"description\": \"Specifies the node to which the
experiment's target Pod belongs. The target Pod can only belong to one node in the
configured node list. If multiple node labels are specified, the node to which the
experiment's target Pod belongs must have all labels specified by this selector.\", \"
type\": \"array\", \"items\": {\"type\": \"string\"}}, \"pods\": {\"title\": \"Pods\",
\"description\": \"Specifies the namespaces and list of the experiment's target Pods.
If you have specified this selector, Chaos Mesh ignores other configured selectors.\",
\"example\": {\"default\": [\"pod-0\", \"pod-2\"]}, \"type\": \"object\", \"
additionalProperties\": {\"type\": \"array\", \"items\": {\"type\": \"string\"}}}}}}}\n
```

**Human:**
# Here is the previous K8s manifests of my system:
**{prev_k8s_yamls}**

# Here is a planned Chaos Engineering:
**{experiment_plan_summary}**

```
# Here is the current K8s menifests of my system:
{curr_k8s_yamls}

# Here is the scope of a fault injection for the previous manifests.
{curr_fault_injection}

Now, please adjust the scope of the fault injection for the current manifests. Note
that you here focus on the 'selector' parameter (i.e., scope).
{format_instructions}
```

# 2-4: Agent for adjusting a VaC script

**System:**
You are a helpful AI assistant for Chaos Engineering.
Given the previous K8s manifests, a previous unit test to verify whether the steady
state satisfies the threshold, and the reconfigured K8s manifests, you will determine
whether the unit test requires adjustment to account for the changes in the
reconfigured manifests, and adjust it as necessary.
Always keep the following rules:
- First, consider which K8s manifest resource is the target of the unit test. If there
are changes to that manifest, update the unit test as necessary. If there are no
changes, the unit test should not require modification.
- You may only make minor adjustments to K8s API, HTTP, or DNS request to account for
changes in resource types, parameter seetings, metadata, etc.
- The reconfiguration was made so that the system satisfy the threshold value in the
previous unit test, so the threshold value or other parameters must remain unchanged in
 the new unit test. For example, suppose the number of replicas was reconfigured from 1
 to 3 in order to maintain a steady state with more than 1 active pod at all times. In
such cases, changing the threshold value from 1 to 3 would alter the intent of this
steady state, so the threshold value must remain unchanged (i.e., more than 1 active
pod)."
- If redundancy has been newly added, the unit test should verify whether the steady
state is maintained by the entire redundancy.
- If the unit test's content needs no changes and only function or variable names need
to be changed, leave them as they are to save output costs.
- The output should be formatted as a JSON instance that conforms to the JSON schema
below.

As an example, for the schema {\"properties\": {\"foo\": {\"title\": \"Foo\", \"
description\": \"a list of strings\", \"type\": \"array\", \"items\": {\"type\": \"
string\"}}}, \"required\": [\"foo\"]}\nthe object {\"foo\": [\"bar\", \"baz\"]} is a
well-formatted instance of the schema. The object {\"properties\": {\"foo\": [\"bar\",
\"baz\"]}} is not well-formatted.

Here is the output schema:
```
{\"properties\": {\"thought\": {\"title\": \"Thought\", \"description\": \"Describe
your thought process for determining whether the unit test requires adjustment to
account for the changes in the reconfigured manifests: First, consider which K8s
manifest resource is the target of the unit test. If there are changes to that manifest
, update the unit test as necessary. If there are no changes, the unit test should not
require modification. If the unit test needs updating, describe also how you modify the
 inspection method according to the differences between the previous and reconfigured
manifests. If the modification is not required, describe the reason.\", \"type\": \"
string\"}, \"code\": {\"title\": \"Code\", \"description\": \"If the unit test needs
updating, write a new unit test code with the inspection method modified. Write only
the content of the code without enclosing it in a code block. If not, this field is not
 required.\", \"type\": \"string\"}}, \"required\": [\"thought\"]}
```

**Human:**
# Here is the previous K8s manifests of my system:
{prev_k8s_yamls}

# Here is the reconfigured K8s manifests of my system:
{curr_k8s_yamls}

# Here is the unit test for the previous manifests.
{prev_unittest}

Now, please determine whether the unit test requires adjustment to account for the
changes in the reconfigured manifests, and adjust it as necessary.

```
 ----- In the verification loop, the prompts below will be stacked as history -----

AI:
{output}

Human:
Your current unit test causes errors when conducted.
The error message is as follows:
{error_message}

This unit test should be succeeded.
Please analyze the reason why the errors occur, then fix the errors.
Always keep the following rules:
- NEVER repeat the same fixes that have been made in the past.
- Fix only the parts related to the errors without changing the original intent.
- {format_instructions}
```

## Example text embedded to **experiment_plan_summary**

The Chaos Engineering experiment is structured into three phases: pre-validation, fault
 injection, and post-validation, all to be completed within a total of 1 minute.

In the pre-validation phase, which lasts for 10 seconds, two unit tests are executed
simultaneously to verify the system's steady states before any faults are introduced.
The 'example-pod-running-state' is checked using a Python script to ensure the Pod is
in the 'Running' state at least 90% of the time. This test runs for 5 seconds.
Concurrently, the 'example-service-http-response-state' is verified using a K6 script
to simulate HTTP requests, ensuring 95% of requests return a 200 OK status. This test
also runs for 5 seconds. Both tests start immediately at the beginning of the phase.

The fault injection phase spans 40 seconds and involves two staggered fault injections.
 Initially, the PodChaos fault is injected to simulate a Pod failure, running for the
first 20 seconds. Simultaneously, the 'example-pod-running-state' unit test is
conducted to observe the impact of this fault. After 20 seconds, the NetworkChaos fault
 is introduced to simulate network latency, running for the remaining 20 seconds.
During this period, the 'example-service-http-response-state' unit test is executed to
assess the effect of network delays. This staggered approach allows for isolated
observation of each fault's impact on the system.

Finally, the post-validation phase, lasting 10 seconds, ensures the system returns to
its steady states after fault removal. The tests are conducted sequentially. The '
example-pod-running-state' is verified first, with a 1-second grace period followed by
a 4-second test duration. Subsequently, the 'example-service-http-response-state' is
checked, starting after a 5-second grace period and running for 4 seconds. This
sequence confirms the system's recovery to its expected steady states.

### B.1.4 ANALYSIS

## # 3-0: Agent for analyzing an experiment results

```
System:
You are a helpful AI assistant for Chaos Engineering.
Given K8s manifests for a network system, its hypothesis, the overview of a Chaos-
Engineeering experiment, and the experimental results, you will analyze the
experimental results.
Always keep the following rules:
- Analyze step by step why the test(s) failed, based on the system configurations (
manifests) and the flow of the experiment.
- Specify the cause while mentioning the corresponding system configurations and the
corresponding phenomena in the Chaos-Engineering experiment.
- The analysis report here will be used for reconfiguring the system later to avoid the
 failures and improve resiliency. Therefore, make carefully the report rich in insights
 so that it will be helpful at that time.
- When providing insights and reconfiguration recommendations, limit them to areas
related to the failed test.
- The output should be formatted as a JSON instance that conforms to the JSON schema
below.

As an example, for the schema {\"properties\": {\"foo\": {\"title\": \"Foo\", \"
description\": \"a list of strings\", \"type\": \"array\", \"items\": {\"type\": \"
```

```
string\"}}}, \"required\": [\"foo\"]}\nthe object {\"foo\": [\"bar\", \"baz\"]} is a
well-formatted instance of the schema. The object {\"properties\": {\"foo\": [\"bar\",
\"baz\"]}} is not well-formatted.

Here is the output schema:
```
{\"properties\": {\"report\": {\"title\": \"Report\", \"description\": \"Analysis of
the experiment result.\", \"type\": \"string\"}}, \"required\": [\"report\"]}
```

Human:
# Here is the overview of my system:
{user_input2}

# Here is the hypothesis for my system:
The hypothesis is "The steady states of the sytem are maintained even when the fault
scenario occurs (i.e., when the faults are injected)".
The steady states here are as follows:
{steady_states}

The fault scenario here is as follows:
{detailed_fault_scenario}

# Here is the overview of my Chaos-Engineering experiment to verify the hypothesis:
{experiment_plan_summary}

--------------------------------------------------------------------------------
 For the first analysis, the following prompt is added
--------------------------------------------------------------------------------

# The experiment's results are as follows:
{experiment_result}

Now, please analyze the results and provide an analysis report rich in insights.

--------------------------------------------------------------------------------
 For the second and subsequent analyses, the following prompt is added
--------------------------------------------------------------------------------

# The update history for the above K8s manifests is the following:
{reconfig_history}

# The experiment's results in the latest K8s manifests are as follows:
{experiment_result}

Now, please analyze the results and provide an analysis report rich in insights.
```

## Example text embedded to `experiment_result`

```
Passed unittests:
- pre-unittest-example-pod-running-state
- pre-unittest-example-service-http-response-state

Failed unittests:
- fault-unittest-example-pod-running-state
```log
Exception when calling CoreV1Api->read_namespaced_pod: (404)
Reason: Not Found\nHTTP response headers: HTTPHeaderDict({'Audit-Id': '8a1e6c00-ebd9-43
ee-9522-6399ce015252', 'Cache-Control': 'no-cache, private', 'Content-Type': '
application/json', 'X-Kubernetes-Pf-Flowschema-Uid': 'c4624bd9-7fc7-42c6-bcb8-4235110
a860d', 'X-Kubernetes-Pf-Prioritylevel-Uid': '4706085f-6263-43ae-93f5-b4a61de8b6be', '
Date': 'Sun, 24 Nov 2024 12:06:18 GMT', 'Content-Length': '190'})
HTTP response body: {\"kind\":\"Status\"...', 'X-Kubernetes-Pf-Flowschema-Uid': '
c4624bd9-7fc7-42c6-bcb8-4235110a860d', 'X-Kubernetes-Pf-Prioritylevel-Uid': '4706085f
-6263-43ae-93f5-b4a61de8b6be', 'Date': 'Sun, 24 Nov 2024 12:06:37 GMT', 'Content-Length
': '190'})\nHTTP response body: {\"kind\":\"Status\",\"apiVersion\":\"v1\",\"metadata
\":{},\"status\":\"Failure\",\"message\":\"pods \\\"example-pod\\\" not found\",\"
reason\":\"NotFound\",\"details\":{\"name\":\"example-pod\",\"kind\":\"pods\"},\"code
\":404}

Pod was running 0 out of 20 seconds, which is 0.00% of the time.
```

```
```

- fault-unittest-example-service-http-response-state
```log
time=\"2024-11-24T12:06:38Z\" level=warning msg=\"Request Failed\" error=\"Get \\\"http
:\/\/example-service.default.svc.cluster.local:80\\\": dial tcp 10.96.255.84:80:
connect: connection refused\"\ntime=\"2024-11-24T12:06:38Z\" level=warning msg=\"
Request Failed\" error=\"Get \\\"http:\/\/example-service.default.svc.cluster.local
:80\\\": dial tcp 10.96.255.84:80: connect: connection refused\"\ntime=\"2024-11-24T12
:06:38Z\" level=warning msg=\"Request Failed\" error=\"Get \\\"http:\/\/example-service
.default.svc.cluster.local:8... level=error msg=\"thresholds on metrics '
http_req_failed' have been crossed
```
```

### B.1.5 IMPROVEMENT

#### # 4-0: Agent for reconfiguring K8s manifests

**System:**
You are a helpful AI assistant for Chaos Engineering.
Given K8s manifests that define a network system, its hypothesis, the overview of a
Chaos-Engineeering experiment, and the experiment's results, you will reconfigure the
system based on analysis of the experiment's results.
Always keep the following rules:
- NEVER change the original intention (its description) of the original version of the
system.
- NEVER do the same reconfiguration as in the history.
- Start with simple reconfiguration, and if the hypothesis is still not satisfied,
gradually try more complex reconfigurations.
- The output should be formatted as a JSON instance that conforms to the JSON schema
below.

As an example, for the schema {\"properties\": {\"foo\": {\"title\": \"Foo\", \"
description\": \"a list of strings\", \"type\": \"array\", \"items\": {\"type\": \"
string\"}}}, \"required\": [\"foo\"]}\nthe object {\"foo\": [\"bar\", \"baz\"]} is a
well-formatted instance of the schema. The object {\"properties\": {\"foo\": [\"bar\",
\"baz\"]}} is not well-formatted.

Here is the output schema:
```
{\"properties\": {\"thought\": {\"title\": \"Thought\", \"description\": \"Describe
your plan to modify the K8s manifests.\", \"type\": \"string\"}, \"modified_k8s_yamls
\": {\"title\": \"Modified K8S Yamls\", \"description\": \"The list of modified K8s
manifests (yamls). If you create a new manifest to modify resources in an existing
manifest, make sure to delete the existing manifest before creating the new one.\", \"
type\": \"array\", \"items\": {\"$ref\": \"#/definitions/ModK8sYAML\"}}}, \"required\":
 [\"thought\", \"modified_k8s_yamls\"], \"definitions\": {\"ModK8sYAML\": {\"title\":
\"ModK8sYAML\", \"type\": \"object\", \"properties\": {\"mod_type\": {\"title\": \"Mod
Type\", \"description\": \"Modification type. Select from ['replace', 'create', 'delete
']. The 'replace' replaces/overwites the content of an exisiting yaml. The 'create'
creates a new yaml. The 'delete' deletes an existing yaml.\", \"enum\": [\"replace\",
\"create\", \"delete\"], \"type\": \"string\"}, \"fname\": {\"title\": \"Fname\", \"
description\": \"The file name of the modified yaml. If mod_type is 'replace' or '
delete', the name must match an existing yaml's name. If mod_type='create', name the
file appropriately to avoid overlapping with existing yamls' names.\", \"type\": \"
string\"}, \"explanation\": {\"title\": \"Explanation\", \"description\": \"If mod_type
 is 'delete', explain why you need to delete the yaml. If mod_type is 'replace',
explain which part you should modify from the original conde and why. If mod_type is '
create', explain whether it is a completely new resource or a replacement resouce for
an existing resource. If it is a replacement, also explain the differences and the
reasons for them, just like with 'replace'.\", \"type\": \"string\"}, \"code\": {\"
title\": \"Code\", \"description\": \"If mod_type is 'delete', this field is not
required. Otherwise, write the content of a K8s YAML manifest modified to pass all the
unit tests. Write only the content of the code, and for dictionary values, enclose them
 within a pair of single double quotes (\\\").\", \"type\": \"string\"}}, \"required\":
 [\"mod_type\", \"fname\", \"explanation\"]}}}
```

**Human:**
# Here is the overview of my system (original version):
{user_input2}

# Here is the hypothesis for my system:
```

```
The hypothesis is "The steady states of the sytem are maintained even when the fault
scenario occurs (i.e., when the faults are injected)".
The steady states here are as follows:
{steady_states}

The fault scenario here is as follows:
{detailed_fault_scenario}

# Here is the overview of my Chaos-Engineering experiment to verify the hypothesis:
{experiment_plan_summary}

# The experiment's results of the original system are as follows:
{experiment_result}

First, please analyze the results and provide an analysis report rich in insights.

AI:
# Here is my analysis report:
{analysis_report}

Human:
Then, please reconfigure the system to avoid the fails (improve resiliency).

-----------------------------------------------------------------------------------
 In the improvement loop, the prompts below will be stacked as improvement history
-----------------------------------------------------------------------------------

AI:
```json
{output}
```"""

Human:
# Here is the K8s menifests of the modified system (version={mod_version}):
{k8s_yamls_mod}

# The experiment's results of the modified system were as follows:
{experiment_result_mod}

Please analyze the results and provide an analysis report rich in insights again.

AI:
# Here is my analysis report:
{analysis_report_mod}

Human:
Then, please reconfigure the system to avoid the fails (improve resiliency).

-----------------------------------------------------------------------------------
 In the verification loop, the prompts below will be stacked as verification history
 Verification history is kept within a single verification loop and is reset in every
 improvement iteration
-----------------------------------------------------------------------------------

AI:
```json
{output}
```

User:
Your current unittest causes errors when conducted.
The error message is as follows:
{error_message}

Please analyze the reason why the errors occur, then fix the errors.
Always keep the following rules:
- Ensure that the implementation supports variable durations again.
- NEVER repeat the same fixes that have been made in the past.
- Fix only the parts related to the errors without changing the original content.
- the same format instruction as in the System role
```

---

### Example text embedded to `analysis_report`

The Chaos Engineering experiment aimed to test the system's resilience by simulating a
cyber attack through PodChaos and NetworkChaos. The experiment results indicate
failures in maintaining the defined steady states during and after the fault injection
phase. Here is a detailed analysis of the results:

1. **Pre-Validation Phase:**
   - Both pre-unittests, `example-pod-running-state` and `example-service-http-response-
   state`, passed successfully. This indicates that the system was in a healthy state
   before the fault injection, with the Pod running 100% of the time and the Service
   responding with a 200 OK status for all requests.

2. **Fault Injection Phase:**
   - **PodChaos Fault:** The `fault-unittest-example-pod-running-state` failed because
   the Pod was not found (404 error) during the test. This is expected due to the
   PodChaos action 'pod-kill', which terminated the Pod. Since the Pod's restart policy
   is set to 'Never', it did not restart, leading to 0% uptime during the test. This
   highlights the critical issue of having a single Pod with no redundancy or automatic
   recovery mechanism.
   - **NetworkChaos Fault:** The `fault-unittest-example-service-http-response-state`
   also failed. The logs show repeated connection refused errors, indicating that the
   Service could not route traffic to the Pod, as it was not running. This failure is a
   direct consequence of the Pod being unavailable, demonstrating the lack of redundancy
    and the impact of a single point of failure.

3. **Post-Validation Phase:**
   - Both post-unittests, `example-pod-running-state` and `example-service-http-response
   -state`, failed. The Pod was still not found, and the Service continued to refuse
   connections. This indicates that the system did not recover to its steady states
   after the faults were removed, primarily due to the Pod's restart policy and the
   absence of a controller to manage Pod lifecycle and redundancy.

**Insights and Recommendations:**
   - **Pod Restart Policy:** The Pod's restart policy should be changed from 'Never' to
   'Always' or 'OnFailure' to ensure automatic recovery in case of failures.
   - **Redundancy and Scalability:** Implement a Deployment or ReplicaSet to manage the
   Pod. This will provide redundancy and ensure that a minimum number of Pods are always
    running, improving the system's resilience to failures.
   - **Service Availability:** Ensure that the Service can handle traffic even if one
   Pod fails by having multiple replicas. This can be achieved by scaling the Deployment
    to have more than one replica.
   - **Monitoring and Alerts:** Implement monitoring and alerting mechanisms to detect
   and respond to Pod failures promptly, minimizing downtime.

By addressing these issues, the system can improve its resilience and maintain its
steady states even during fault scenarios.\nHuman: Then, please reconfigure the system
to avoid the fails (improve resiliency).

### B.1.6 POST-PROCESSING

---

### # EX: Agent for summarizing a completed CE cycle

**System:**
You are a helpful AI assistant for Chaos Engineering.
Given a summary of a Chaos Engineering cycle, please elaborate the summary.
The output should be formatted as a JSON instance that conforms to the JSON schema
below.

As an example, for the schema {\"properties\": {\"foo\": {\"title\": \"Foo\", \"
description\": \"a list of strings\", \"type\": \"array\", \"items\": {\"type\": \"
string\"}}}, \"required\": [\"foo\"]}\nthe object {\"foo\": [\"bar\", \"baz\"]} is a
well-formatted instance of the schema. The object {\"properties\": {\"foo\": [\"bar\",
\"baz\"]}} is not well-formatted.

Here is the output schema:
```
{\"properties\": {\"summary\": {\"title\": \"Summary\", \"type\": \"string\"}}, \"
required\": [\"summary\"]}
```

**Human:**
Here is the overview of a Chaos Engineering Cycle:
# Here is a Chaos Engineering cycle

---

```
## Step 0. User-input understanding
### Here is the overview of user inputs:
{user_input2}

## Step 1. Hypothesis definition
### Here is the overview of the hypothesis for the system:
The hypothesis is "The steady states of the system are maintained even when the fault
scenario occurs (i.e., when the faults are injected)".
The steady states here are as follows:
{steady_states}

The fault scenario here is as follows:
{detailed_fault_scenario}

## Step 2.1. Chaos-Engineering experiment
### Here is the overview of my Chaos-Engineering experiment to verify the hypothesis:
{experiment_plan_summary}

## Step 2.2, 3, 4. Experiment execution, analysis and improvement (reconfiguring the
system to satisfy the hypothesis)
### Here is the improvement history:
{improvement_history}

Please elaborate the above summary of a Chaos Engineering Cycle.
```

## Example text embedded to **improvement_history**

```
### Experiment result (1st try)
the same content as experiment_result

### Analysis report (1st try)
the same content as analysis_report

### Improvement result (1st try)
1 K8s manifests are modified:
- The K8s manifest 'nginx/pod.yaml' was replaced.

Replace the Pod manifest with a Deployment manifest to provide redundancy and automatic
 recovery. The Deployment will manage the Pod lifecycle, ensuring that a specified
number of replicas are always running, which addresses both the restart policy and
redundancy issues.
```yaml
apiVersion: apps/v1
kind: Deployment
metadata:
  name: example-deployment
  labels:
    app: example
spec:
  replicas: 3
  selector:
    matchLabels:
      app: example
    template:
      metadata:
        labels:
          app: example
      spec:
        containers:
        - name: example-container
          image: nginx:1.17.1
          ports:
          - containerPort: 80
```

### Experiment result (2nd try)
Passed unittests:
- pre-unittest-example-pod-running-state
- pre-unittest-example-service-http-response-state
- fault-unittest-example-pod-running-state
- fault-unittest-example-service-http-response-state
- post-unittest-example-pod-running-state
- post-unittest-example-service-http-response-state

Failed unittests:
```

## C  FULL RESULTS OF CASE STUDIES

In this section, we provide full inputs and outputs for each run studied in the case study. The inputs include instructions, Skaffold configuration files, K8s manifests. The outputs include VaC scripts, Chaos Mesh workflow manifests, reconfigured K8s manifests, adjusted VaC scripts in the experiment replanning phase. In the case study, we pick up the fourth run for NGINX and the fifth run for SOCKSHOP. For complete dialogues, see casestudy_complete_dialogues/Nginx/ChaosEater_GPT4o_nginx_4.pdf for NGINX and casestudy_complete_dialogues/Sockshop/ChaosEater_GPT4o_sockshop_5.pdf in the Supplementary Material. The results of other runs can also be found in the same folders. Table 3 shows the statistics of NGINX and SOCKSHOP.

Table 3: Statistics of the systems for the case study. Tokens are counted by the tokenizer `cl100k_base`.

| System | # manifests | # lines | # words | # tokens |
|--------|-------------|---------|---------|----------|
| NGINX | 2 | 24 | 373 | 115 |
| SOCKSHOP | 29 | 869 | 17696 | 4605 |

### C.1  NGINX

#### C.1.1  INPUTS

**Instructions**

```
The Chaos-Engineering experiment must be completed within 1 minute.
```

**skaffold.yaml**

```
1 apiVersion: skaffold/v3
2 kind: Config
3 metadata:
4   name: nginx-app
5 manifests:
6   rawYaml:
7     - pod.yaml
8     - service.yaml
```

**pod.yaml**

```
1  apiVersion: v1
2  kind: Pod
3  metadata:
4    name: example-pod
5    labels:
6      app: example
7  spec:
8    restartPolicy: Never
9    containers:
10   - name: example-container
11     image: nginx:1.17.1
12     ports:
13     - containerPort: 80
```

service.yaml

```
1  apiVersion: v1
2  kind: Service
3  metadata:
4    name: example-service
5  spec:
6    selector:
7      app: example
8    ports:
9      - protocol: TCP
10       port: 80
11       targetPort: 80
```

## C.1.2  OUTPUTS

VaC script for the steady state #1: The Pod should be running at least 90% of the time during the check period

```
1  import os
2  import time
3  import argparse
4  from kubernetes import client, config
5  from unittest_base import K8sAPIBase
6
7  class TestPodRunningState(K8sAPIBase):
8      def __init__(self):
9          super().__init__()
10
11     def check_pod_status(self, namespace, pod_name):
12         try:
13             pod = self.v1.read_namespaced_pod(name=pod_name, namespace=namespace)
14             return pod.status.phase == 'Running'
15         except client.exceptions.ApiException as e:
16             print(f"Exception when calling CoreV1Api->read_namespaced_pod: {e}")
17             return False
18
19     def test_pod_running_state(self, duration):
20         namespace = 'default'
21         pod_name = 'example-pod'
22         running_count = 0
23
24         # Check the pod status every second for the specified duration
25         for _ in range(duration):
26             if self.check_pod_status(namespace, pod_name):
27                 running_count += 1
28             time.sleep(1)
29
30         # Calculate the running percentage
31         running_percentage = (running_count / duration) * 100
32
33         # Assert that the running percentage is at least 90%
34         assert running_percentage >= 90, f"Pod '{pod_name}' running percentage is below
             threshold: {running_percentage}%"
35
36         print(f"Pod '{pod_name}' running status checked {duration} times. Running
             percentage: {running_percentage}%.")
37
38
39 def main():
40     parser = argparse.ArgumentParser(description='Test if a pod is running at least 90%
            of the time.')
41     parser.add_argument('--duration', type=int, default=5, help='Duration to check the
            pod status in seconds.')
42     args = parser.parse_args()
43
44     test = TestPodRunningState()
45     test.test_pod_running_state(args.duration)
46
47
48 if __name__ == '__main__':
49     main()
```

VaC script for the steady state #2: Service availability should be at least 99.9% with a response status of 200

```javascript
1  import http from 'k6/http';
2  import { check } from 'k6';
3
4  export const options = {
5    vus: 1,
6    duration: '5s',
7    thresholds: {
8      // Ensure that the service availability is at least 99.9%
9      'http_req_failed': ['rate<=0.001'], // 0.1% failure rate corresponds to 99.9%
       availability
10    },
11  };
12
13  export default function () {
14    const res = http.get('http://example-service.default.svc.cluster.local:80');
15    check(res, {
16      'status is 200': (r) => r.status === 200,
17    });
18  }
```

Chaos Mesh workflow manifest

```yaml
1  apiVersion: chaos-mesh.org/v1alpha1
2  kind: Workflow
3  metadata:
4    name: chaos-experiment-20241124-132854
5  spec:
6    entry: the-entry
7    templates:
8      #-----------------------------
9      # entry point of whole workflow
10     #-----------------------------
11     - name: the-entry
12       templateType: Serial
13       deadline: 30m51s
14       children:
15         - pre-validation-phase
16         - fault-injection-phase
17         - post-validation-phase
18
19     #-------------------------------------
20     # Entry point of pre-validation-phase
21     #-------------------------------------
22     - name: pre-validation-phase
23       templateType: Serial
24       deadline: 10m10s
25       children:
26         - pre-validation-overlapped-workflows
27
28     - name: pre-validation-suspend-workflow
29       templateType: Serial
30       deadline: 5m10s
31       children:
32         - pre-validation-suspend
33         - pre-unittest-example-service-availability
34
35     - name: pre-validation-suspend
36       templateType: Suspend
37       deadline: 5s
38
39     - name: pre-validation-overlapped-workflows
40       templateType: Parallel
41       deadline: 5m10s
42       children:
43         - pre-unittest-example-pod-running
44         - pre-validation-suspend-workflow
45
46     # Definitions of children of pre-validation-phase
47     - name: pre-unittest-example-pod-running
48       templateType: Task
49       deadline: 5m5s
```

```
 50       task:
 51         container:
 52           name: pre-unittest-example-pod-running-container
 53           image: chaos-eater/k8sapi:1.0
 54           imagePullPolicy: IfNotPresent
 55           command: ["/bin/bash", "-c"]
 56           args: ["python /chaos-eater/sandbox/cycle_20241124_132128/unittest_example-
                pod-running_mod0.py --duration 5"]
 57           volumeMounts:
 58             - name: pvc-volume
 59               mountPath: /chaos-eater
 60         volumes:
 61           - name: pvc-volume
 62             persistentVolumeClaim:
 63               claimName: pvc
 64
 65     - name: pre-unittest-example-service-availability
 66       templateType: Task
 67       deadline: 5m5s
 68       task:
 69         container:
 70           name: pre-unittest-example-service-availability-container
 71           image: grafana/k6:latest
 72           command: ["k6", "run", "--duration", "5s", "--quiet", "/chaos-eater/sandbox/
                cycle_20241124_132128/unittest_example-service-availability_mod0.js"]
 73           volumeMounts:
 74             - name: pvc-volume
 75               mountPath: /chaos-eater
 76         volumes:
 77           - name: pvc-volume
 78             persistentVolumeClaim:
 79               claimName: pvc
 80
 81     #------------------------------------
 82     # Entry point of fault-injection-phase
 83     #------------------------------------
 84     - name: fault-injection-phase
 85       templateType: Serial
 86       deadline: 10m30s
 87       children:
 88         - fault-injection-overlapped-workflows
 89
 90     - name: fault-injection-parallel-workflow
 91       templateType: Parallel
 92       deadline: 5m10s
 93       children:
 94         - fault-unittest-example-pod-running
 95         - fault-podchaos
 96
 97     - name: fault-injection-suspend-workflow
 98       templateType: Serial
 99       deadline: 5m30s
100       children:
101         - fault-injection-suspend
102         - fault-injection-parallel-workflows
103
104     - name: fault-injection-suspend
105       templateType: Suspend
106       deadline: 10s
107
108     - name: fault-injection-parallel-workflows
109       templateType: Parallel
110       deadline: 5m20s
111       children:
112         - fault-unittest-example-service-availability
113         - fault-networkchaos
114
115     - name: fault-injection-overlapped-workflows
116       templateType: Parallel
117       deadline: 5m30s
118       children:
119         - fault-injection-parallel-workflow
120         - fault-injection-suspend-workflow
121
122     # Definitions of children of pre-validation-phase
123     # unit tests
124     - name: fault-unittest-example-pod-running
125       templateType: Task
```

```
126        deadline: 5m10s
127        task:
128          container:
129            name: fault-unittest-example-pod-running-container
130            image: chaos-eater/k8sapi:1.0
131            imagePullPolicy: IfNotPresent
132            command: ["/bin/bash", "-c"]
133            args: ["python /chaos-eater/sandbox/cycle_20241124_132128/unittest_example-
                   pod-running_mod0.py --duration 10"]
134            volumeMounts:
135              - name: pvc-volume
136                mountPath: /chaos-eater
137          volumes:
138            - name: pvc-volume
139              persistentVolumeClaim:
140                claimName: pvc
141
142      - name: fault-unittest-example-service-availability
143        templateType: Task
144        deadline: 5m20s
145        task:
146          container:
147            name: fault-unittest-example-service-availability-container
148            image: grafana/k6:latest
149            command: ["k6", "run", "--duration", "20s", "--quiet", "/chaos-eater/sandbox/
                   cycle_20241124_132128/unittest_example-service-availability_mod0.js"]
150            volumeMounts:
151              - name: pvc-volume
152                mountPath: /chaos-eater
153          volumes:
154            - name: pvc-volume
155              persistentVolumeClaim:
156                claimName: pvc
157
158    # fault_injections
159    - name: fault-podchaos
160      templateType: PodChaos
161      deadline: 10s
162      podChaos:
163        action: pod-kill
164        mode: one
165        selector:
166          labelSelectors:
167            app: example
168          namespaces:
169            - default
170
171
172    - name: fault-networkchaos
173      templateType: NetworkChaos
174      deadline: 20s
175      networkChaos:
176        action: delay
177        delay:
178          correlation: '50'
179          jitter: 10ms
180          latency: 100ms
181        device: eth0
182        direction: to
183        mode: all
184        selector:
185          labelSelectors:
186            app: example
187          namespaces:
188            - default
189        target:
190          mode: all
191          selector:
192            labelSelectors:
193              app: example
194            namespaces:
195              - default
196
197
198    #-------------------------------------
199    # Entry point of post-validation phase
200    #-------------------------------------
201    - name: post-validation-phase
```

```
202        templateType: Serial
203        deadline: 10m11s
204        children:
205          - post-validation-overlapped-workflows
206
207      - name: post-validation-suspend-workflow
208        templateType: Serial
209        deadline: 5m8s
210        children:
211          - post-validation-suspend
212          - post-unittest-example-pod-running
213
214      - name: post-validation-suspend
215        templateType: Suspend
216        deadline: 2s
217
218      - name: post-validation-suspend-workflow2
219        templateType: Serial
220        deadline: 5m11s
221        children:
222          - post-validation-suspend2
223          - post-unittest-example-service-availability
224
225      - name: post-validation-suspend2
226        templateType: Suspend
227        deadline: 6s
228
229      - name: post-validation-overlapped-workflows
230        templateType: Parallel
231        deadline: 5m11s
232        children:
233          - post-validation-suspend-workflow
234          - post-validation-suspend-workflow2
235
236      # Definitions of children of pre-validation-phase
237      - name: post-unittest-example-pod-running
238        templateType: Task
239        deadline: 5m6s
240        task:
241          container:
242            name: post-unittest-example-pod-running-container
243            image: chaos-eater/k8sapi:1.0
244            imagePullPolicy: IfNotPresent
245            command: ["/bin/bash", "-c"]
246            args: ["python /chaos-eater/sandbox/cycle_20241124_132128/unittest_example-
                   pod-running_mod0.py --duration 6"]
247            volumeMounts:
248              - name: pvc-volume
249                mountPath: /chaos-eater
250          volumes:
251            - name: pvc-volume
252              persistentVolumeClaim:
253                claimName: pvc
254
255      - name: post-unittest-example-service-availability
256        templateType: Task
257        deadline: 5m5s
258        task:
259          container:
260            name: post-unittest-example-service-availability-container
261            image: grafana/k6:latest
262            command: ["k6", "run", "--duration", "5s", "--quiet", "/chaos-eater/sandbox/
                   cycle_20241124_132128/unittest_example-service-availability_mod0.js"]
263            volumeMounts:
264              - name: pvc-volume
265                mountPath: /chaos-eater
266          volumes:
267            - name: pvc-volume
268              persistentVolumeClaim:
269                claimName: pvc
```

## Reconfigured pod.yaml

```
1 apiVersion: apps/v1
2 kind: Deployment
```

```
3  metadata:
4    name: example-deployment
5    labels:
6      app: example
7  spec:
8    replicas: 3
9    selector:
10     matchLabels:
11       app: example
12   template:
13     metadata:
14       labels:
15         app: example
16     spec:
17       containers:
18       - name: example-container
19         image: nginx:1.17.1
20         ports:
21         - containerPort: 80
```

Adjusted VaC script for the steady state #1: The Pod should be running at least 90% of the time during the check period

```python
1  import os
2  import time
3  import argparse
4  from kubernetes import client, config
5  from unittest_base import K8sAPIBase
6
7  class TestDeploymentRunningState(K8sAPIBase):
8      def __init__(self):
9          super().__init__()
10
11     def check_deployment_pods_status(self, namespace, deployment_name):
12         try:
13             pods = self.v1.list_namespaced_pod(namespace=namespace, label_selector=f'
                 app={deployment_name}').items
14             running_pods = [pod for pod in pods if pod.status.phase == 'Running']
15             return len(running_pods) > 0
16         except client.exceptions.ApiException as e:
17             print(f"Exception when calling CoreV1Api->list_namespaced_pod: {e}")
18             return False
19
20     def test_deployment_running_state(self, duration):
21         namespace = 'default'
22         deployment_name = 'example'
23         running_count = 0
24
25         # Check the deployment pods status every second for the specified duration
26         for _ in range(duration):
27             if self.check_deployment_pods_status(namespace, deployment_name):
28                 running_count += 1
29             time.sleep(1)
30
31         # Calculate the running percentage
32         running_percentage = (running_count / duration) * 100
33
34         # Assert that the running percentage is at least 90%
35         assert running_percentage >= 90, f"Deployment '{deployment_name}' running
             percentage is below threshold: {running_percentage}%"
36
37         print(f"Deployment '{deployment_name}' running status checked {duration} times.
             Running percentage: {running_percentage}%.")
38
39
40 def main():
41     parser = argparse.ArgumentParser(description='Test if a deployment has at least one
            pod running at least 90% of the time.')
42     parser.add_argument('--duration', type=int, default=5, help='Duration to check the
            deployment pods status in seconds.')
43     args = parser.parse_args()
44
45     test = TestDeploymentRunningState()
46     test.test_deployment_running_state(args.duration)
47
```

```
48
49 if __name__ == '__main__':
50     main()
```

## C.2 SOCKSHOP

### C.2.1 INPUTS

---

**Instructions**

```
– The Chaos-Engineering experiment must be completed within 1 minute.
– When using k6 in steady-state definition, always select a request URL from the
following options (other requests are invalid):
1. http://front-end.sock-shop.svc.cluster.local/
2. http://front-end.sock-shop.svc.cluster.local/catalogue?size=10
3. http://front-end.sock-shop.svc.cluster.local/detail.html?id=<ID>
   Replace <ID> with an available ID: [03fef6ac-1896-4ce8-bd69-b798f85c6e0b, 3395a43e-2
   d88-40de-b95f-e00e1502085b, 510a0d7e-8e83-4193-b483-e27e09ddc34d, 808a2de1-1aaa-4c25
   -a9b9-6612e8f29a38, 819e1fbf-8b7e-4f6d-811f-693534916a8b, 837ab141-399e-4c1f-9abc-
   bace40296bac, a0a4f044-b040-410d-8ead-4de0446aec7e, d3588630-ad8e-49df-bbd7-3167
   f7efb246, zzz4f044-b040-410d-8ead-4de0446aec7e]
4. http://front-end.sock-shop.svc.cluster.local/category/
5. http://front-end.sock-shop.svc.cluster.local/category?tags=<TAG>
   Replace <TAG> with an available tag: [magic, action, blue, brown, black, sport,
   formal, red, green, skin, geek]
6. http://front-end.sock-shop.svc.cluster.local/basket.html
```

---

**skaffold.yaml**

```
1 apiVersion: skaffold/v3
2 kind: Config
3 metadata:
4   name: sock-shop-app
5 manifests:
6   rawYaml:
7     - manifests/00-sock-shop-ns.yaml
8     - manifests/01-carts-dep.yaml
9     - manifests/02-carts-svc.yaml
10    - manifests/03-carts-db-dep.yaml
11    - manifests/04-carts-db-svc.yaml
12    - manifests/05-catalogue-dep.yaml
13    - manifests/06-catalogue-svc.yaml
14    - manifests/07-catalogue-db-dep.yaml
15    - manifests/08-catalogue-db-svc.yaml
16    - manifests/09-front-end-dep.yaml
17    - manifests/10-front-end-svc.yaml
18    - manifests/11-orders-dep.yaml
19    - manifests/12-orders-svc.yaml
20    - manifests/13-orders-db-dep.yaml
21    - manifests/14-orders-db-svc.yaml
22    - manifests/15-payment-dep.yaml
23    - manifests/16-payment-svc.yaml
24    - manifests/17-queue-master-dep.yaml
25    - manifests/18-queue-master-svc.yaml
26    - manifests/19-rabbitmq-dep.yaml
27    - manifests/20-rabbitmq-svc.yaml
28    - manifests/21-session-db-dep.yaml
29    - manifests/22-session-db-svc.yaml
30    - manifests/23-shipping-dep.yaml
31    - manifests/24-shipping-svc.yaml
32    - manifests/25-user-dep.yaml
33    - manifests/26-user-svc.yaml
34    - manifests/27-user-db-dep.yaml
35    - manifests/28-user-db-svc.yaml
```

**manifests/00-sock-shop-ns.yaml**

```
1  apiVersion: v1
2  kind: Namespace
3  metadata:
4    name: sock-shop
```

**manifests/01-carts-dep.yaml**

```
1  apiVersion: apps/v1
2  kind: Deployment
3  metadata:
4    name: carts
5    labels:
6      name: carts
7    namespace: sock-shop
8  spec:
9    replicas: 2
10   selector:
11     matchLabels:
12       name: carts
13   template:
14     metadata:
15       labels:
16         name: carts
17     spec:
18       containers:
19       - name: carts
20         image: weaveworksdemos/carts:0.4.8
21         env:
22          - name: JAVA_OPTS
23            value: -Xms64m -Xmx128m -XX:+UseG1GC -Djava.security.egd=file:/dev/urandom -
                   Dspring.zipkin.enabled=false
24         resources:
25           limits:
26             cpu: 300m
27             memory: 500Mi
28           requests:
29             cpu: 100m
30             memory: 200Mi
31         ports:
32         - containerPort: 80
33         securityContext:
34           runAsNonRoot: true
35           runAsUser: 10001
36           capabilities:
37             drop:
38               - all
39             add:
40               - NET_BIND_SERVICE
41           readOnlyRootFilesystem: true
42         volumeMounts:
43         - mountPath: /tmp
44           name: tmp-volume
45       volumes:
46         - name: tmp-volume
47           emptyDir:
48             medium: Memory
49       nodeSelector:
50         beta.kubernetes.io/os: linux
```

**manifests/02-carts-svc.yaml**

```
1  apiVersion: v1
2  kind: Service
3  metadata:
4    name: carts
5    annotations:
6        prometheus.io/scrape: 'true'
7    labels:
8      name: carts
```

```
 9   namespace: sock-shop
10 spec:
11   ports:
12     # the port that this service should serve on
13   - port: 80
14     targetPort: 80
15   selector:
16     name: carts
```

**manifests/03-carts-db-dep.yaml**

```
 1 apiVersion: apps/v1
 2 kind: Deployment
 3 metadata:
 4   name: carts-db
 5   labels:
 6     name: carts-db
 7   namespace: sock-shop
 8 spec:
 9   replicas: 2
10   selector:
11     matchLabels:
12       name: carts-db
13   template:
14     metadata:
15       labels:
16         name: carts-db
17     spec:
18       containers:
19       - name: carts-db
20         image: mongo
21         ports:
22         - name: mongo
23           containerPort: 27017
24         securityContext:
25           capabilities:
26             drop:
27               - all
28             add:
29               - CHOWN
30               - SETGID
31               - SETUID
32           readOnlyRootFilesystem: true
33         volumeMounts:
34         - mountPath: /tmp
35           name: tmp-volume
36       volumes:
37         - name: tmp-volume
38           emptyDir:
39             medium: Memory
40       nodeSelector:
41         beta.kubernetes.io/os: linux
```

**manifests/04-carts-db-svc.yaml**

```
 1 apiVersion: v1
 2 kind: Service
 3 metadata:
 4   name: carts-db
 5   labels:
 6     name: carts-db
 7   namespace: sock-shop
 8 spec:
 9   ports:
10     # the port that this service should serve on
11   - port: 27017
12     targetPort: 27017
13   selector:
14     name: carts-db
```

manifests/05-catalogue-dep.yaml

```
1  apiVersion: apps/v1
2  kind: Deployment
3  metadata:
4    name: catalogue
5    labels:
6      name: catalogue
7    namespace: sock-shop
8  spec:
9    replicas: 2
10   selector:
11     matchLabels:
12       name: catalogue
13   template:
14     metadata:
15       labels:
16         name: catalogue
17     spec:
18       containers:
19       - name: catalogue
20         image: weaveworksdemos/catalogue:0.3.5
21         command: ["/app"]
22         args:
23         - -port=80
24         resources:
25           limits:
26             cpu: 200m
27             memory: 200Mi
28           requests:
29             cpu: 100m
30             memory: 100Mi
31         ports:
32         - containerPort: 80
33         securityContext:
34           runAsNonRoot: true
35           runAsUser: 10001
36           capabilities:
37             drop:
38               - all
39             add:
40               - NET_BIND_SERVICE
41           readOnlyRootFilesystem: true
42         livenessProbe:
43           httpGet:
44             path: /health
45             port: 80
46           initialDelaySeconds: 300
47           periodSeconds: 3
48         readinessProbe:
49           httpGet:
50             path: /health
51             port: 80
52           initialDelaySeconds: 180
53           periodSeconds: 3
54       nodeSelector:
55         beta.kubernetes.io/os: linux
```

manifests/06-catalogue-svc.yaml

```
1  apiVersion: v1
2  kind: Service
3  metadata:
4    name: catalogue
5    annotations:
6        prometheus.io/scrape: 'true'
7    labels:
8      name: catalogue
9    namespace: sock-shop
10 spec:
11   ports:
12     # the port that this service should serve on
13   - port: 80
14     targetPort: 80
15   selector:
```

```
16     name: catalogue
```

### manifests/06-catalogue-svc.yaml

```
1 apiVersion: v1
2 kind: Service
3 metadata:
4   name: catalogue
5   annotations:
6       prometheus.io/scrape: 'true'
7   labels:
8     name: catalogue
9   namespace: sock-shop
10 spec:
11   ports:
12     # the port that this service should serve on
13   - port: 80
14     targetPort: 80
15   selector:
16     name: catalogue
```

### manifests/07-catalogue-db-dep.yaml

```
1 apiVersion: apps/v1
2 kind: Deployment
3 metadata:
4   name: catalogue-db
5   labels:
6     name: catalogue-db
7   namespace: sock-shop
8 spec:
9   replicas: 2
10   selector:
11     matchLabels:
12       name: catalogue-db
13   template:
14     metadata:
15       labels:
16         name: catalogue-db
17     spec:
18       containers:
19       - name: catalogue-db
20         image: weaveworksdemos/catalogue-db:0.3.0
21         env:
22           - name: MYSQL_ROOT_PASSWORD
23             value: fake_password
24           - name: MYSQL_DATABASE
25             value: socksdb
26         ports:
27         - name: mysql
28           containerPort: 3306
29       nodeSelector:
30         beta.kubernetes.io/os: linux
```

### manifests/08-catalogue-db-svc.yaml

```
1 apiVersion: v1
2 kind: Service
3 metadata:
4   name: catalogue-db
5   labels:
6     name: catalogue-db
7   namespace: sock-shop
8 spec:
9   ports:
10     # the port that this service should serve on
11   - port: 3306
12     targetPort: 3306
```

```
13   selector:
14     name: catalogue-db
```

**manifests/09-front-end-dep.yaml**

```
 1 apiVersion: apps/v1
 2 kind: Deployment
 3 metadata:
 4   name: front-end
 5   namespace: sock-shop
 6 spec:
 7   replicas: 1
 8   selector:
 9     matchLabels:
10       name: front-end
11   template:
12     metadata:
13       labels:
14         name: front-end
15     spec:
16       containers:
17       - name: front-end
18         image: weaveworksdemos/front-end:0.3.12
19         resources:
20           limits:
21             cpu: 300m
22             memory: 1000Mi
23           requests:
24             cpu: 100m
25             memory: 300Mi
26         ports:
27         - containerPort: 8079
28         env:
29         - name: SESSION_REDIS
30           value: "true"
31         securityContext:
32           runAsNonRoot: true
33           runAsUser: 10001
34           capabilities:
35             drop:
36               - all
37           readOnlyRootFilesystem: true
38         livenessProbe:
39           httpGet:
40             path: /
41             port: 8079
42           initialDelaySeconds: 300
43           periodSeconds: 3
44         readinessProbe:
45           httpGet:
46             path: /
47             port: 8079
48           initialDelaySeconds: 30
49           periodSeconds: 3
50       nodeSelector:
51         beta.kubernetes.io/os: linux
```

**manifests/10-front-end-svc.yaml**

```
 1 apiVersion: v1
 2 kind: Service
 3 metadata:
 4   name: front-end
 5   annotations:
 6         prometheus.io/scrape: 'true'
 7   labels:
 8     name: front-end
 9   namespace: sock-shop
10 spec:
11   type: NodePort
12   ports:
13   - port: 80
```

```
14      targetPort: 8079
15      nodePort: 30001
16    selector:
17      name: front-end
```

### manifests/11-orders-dep.yaml

```
1  apiVersion: apps/v1
2  kind: Deployment
3  metadata:
4    name: orders
5    labels:
6      name: orders
7    namespace: sock-shop
8  spec:
9    replicas: 2
10   selector:
11     matchLabels:
12       name: orders
13   template:
14     metadata:
15       labels:
16         name: orders
17     spec:
18       containers:
19       - name: orders
20         image: weaveworksdemos/orders:0.4.7
21         env:
22          - name: JAVA_OPTS
23            value: -Xms64m -Xmx128m -XX:+UseG1GC -Djava.security.egd=file:/dev/urandom -
                   Dspring.zipkin.enabled=false
24         resources:
25           limits:
26             cpu: 500m
27             memory: 500Mi
28           requests:
29             cpu: 100m
30             memory: 300Mi
31         ports:
32         - containerPort: 80
33         securityContext:
34           runAsNonRoot: true
35           runAsUser: 10001
36           capabilities:
37             drop:
38               - all
39             add:
40               - NET_BIND_SERVICE
41           readOnlyRootFilesystem: true
42         volumeMounts:
43         - mountPath: /tmp
44           name: tmp-volume
45       volumes:
46         - name: tmp-volume
47           emptyDir:
48             medium: Memory
49       nodeSelector:
50         beta.kubernetes.io/os: linux
```

### manifests/12-orders-svc.yaml

```
1  apiVersion: v1
2  kind: Service
3  metadata:
4    name: orders
5    annotations:
6        prometheus.io/scrape: 'true'
7    labels:
8      name: orders
9    namespace: sock-shop
10 spec:
11   ports:
```

```
12      # the port that this service should serve on
13    - port: 80
14      targetPort: 80
15    selector:
16      name: orders
```

manifests/13-orders-db-dep.yaml

```
 1 apiVersion: apps/v1
 2 kind: Deployment
 3 metadata:
 4   name: orders-db
 5   labels:
 6     name: orders-db
 7   namespace: sock-shop
 8 spec:
 9   replicas: 2
10   selector:
11     matchLabels:
12       name: orders-db
13   template:
14     metadata:
15       labels:
16         name: orders-db
17     spec:
18       containers:
19       - name: orders-db
20         image: mongo
21         ports:
22         - name: mongo
23           containerPort: 27017
24         securityContext:
25           capabilities:
26             drop:
27               - all
28             add:
29               - CHOWN
30               - SETGID
31               - SETUID
32           readOnlyRootFilesystem: true
33         volumeMounts:
34         - mountPath: /tmp
35           name: tmp-volume
36       volumes:
37       - name: tmp-volume
38         emptyDir:
39           medium: Memory
40       nodeSelector:
41         beta.kubernetes.io/os: linux
```

manifests/14-orders-db-svc.yaml

```
 1 apiVersion: v1
 2 kind: Service
 3 metadata:
 4   name: orders-db
 5   labels:
 6     name: orders-db
 7   namespace: sock-shop
 8 spec:
 9   ports:
10     # the port that this service should serve on
11   - port: 27017
12     targetPort: 27017
13   selector:
14     name: orders-db
```

**manifests/15-payment-dep.yaml**

```
1  apiVersion: apps/v1
2  kind: Deployment
3  metadata:
4    name: payment
5    labels:
6      name: payment
7    namespace: sock-shop
8  spec:
9    replicas: 2
10   selector:
11     matchLabels:
12       name: payment
13   template:
14     metadata:
15       labels:
16         name: payment
17     spec:
18       containers:
19       - name: payment
20         image: weaveworksdemos/payment:0.4.3
21         resources:
22           limits:
23             cpu: 200m
24             memory: 200Mi
25           requests:
26             cpu: 99m
27             memory: 100Mi
28         ports:
29         - containerPort: 80
30         securityContext:
31           runAsNonRoot: true
32           runAsUser: 10001
33           capabilities:
34             drop:
35               - all
36             add:
37               - NET_BIND_SERVICE
38           readOnlyRootFilesystem: true
39         livenessProbe:
40           httpGet:
41             path: /health
42             port: 80
43           initialDelaySeconds: 300
44           periodSeconds: 3
45         readinessProbe:
46           httpGet:
47             path: /health
48             port: 80
49           initialDelaySeconds: 180
50           periodSeconds: 3
51       nodeSelector:
52         beta.kubernetes.io/os: linux
```

**manifests/16-payment-svc.yaml**

```
1  apiVersion: v1
2  kind: Service
3  metadata:
4    name: payment
5    annotations:
6        prometheus.io/scrape: 'true'
7    labels:
8      name: payment
9    namespace: sock-shop
10 spec:
11   ports:
12     # the port that this service should serve on
13   - port: 80
14     targetPort: 80
15   selector:
16     name: payment
```

manifests/17-queue-master-dep.yaml

```
1  apiVersion: apps/v1
2  kind: Deployment
3  metadata:
4    name: queue-master
5    labels:
6      name: queue-master
7    namespace: sock-shop
8  spec:
9    replicas: 2
10   selector:
11     matchLabels:
12       name: queue-master
13   template:
14     metadata:
15       labels:
16         name: queue-master
17     spec:
18       containers:
19       - name: queue-master
20         image: weaveworksdemos/queue-master:0.3.1
21         env:
22          - name: JAVA_OPTS
23            value: -Xms64m -Xmx128m -XX:+UseG1GC -Djava.security.egd=file:/dev/urandom -
                   Dspring.zipkin.enabled=false
24         resources:
25           limits:
26             cpu: 300m
27             memory: 500Mi
28           requests:
29             cpu: 100m
30             memory: 300Mi
31         ports:
32         - containerPort: 80
33       nodeSelector:
34         beta.kubernetes.io/os: linux
```

manifests/18-queue-master-svc.yaml

```
1  apiVersion: v1
2  kind: Service
3  metadata:
4    name: queue-master
5    annotations:
6        prometheus.io/scrape: 'true'
7    labels:
8      name: queue-master
9    namespace: sock-shop
10 spec:
11   ports:
12     # the port that this service should serve on
13   - port: 80
14     targetPort: 80
15   selector:
16     name: queue-master
```

manifests/19-rabbitmq-dep.yaml

```
1  apiVersion: apps/v1
2  kind: Deployment
3  metadata:
4    name: rabbitmq
5    labels:
6      name: rabbitmq
7    namespace: sock-shop
8  spec:
9    replicas: 2
10   selector:
11     matchLabels:
12       name: rabbitmq
```

```
13    template:
14      metadata:
15        labels:
16          name: rabbitmq
17        annotations:
18          prometheus.io/scrape: "false"
19      spec:
20        containers:
21        - name: rabbitmq
22          image: rabbitmq:3.6.8-management
23          ports:
24          - containerPort: 15672
25            name: management
26          - containerPort: 5672
27            name: rabbitmq
28          securityContext:
29            capabilities:
30              drop:
31                - all
32              add:
33                - CHOWN
34                - SETGID
35                - SETUID
36                - DAC_OVERRIDE
37            readOnlyRootFilesystem: true
38        - name: rabbitmq-exporter
39          image: kbudde/rabbitmq-exporter
40          ports:
41          - containerPort: 9090
42            name: exporter
43        nodeSelector:
44          beta.kubernetes.io/os: linux
```

---

**manifests/20-rabbitmq-svc.yaml**

```
1  apiVersion: v1
2  kind: Service
3  metadata:
4    name: rabbitmq
5    annotations:
6        prometheus.io/scrape: 'true'
7        prometheus.io/port: '9090'
8    labels:
9      name: rabbitmq
10   namespace: sock-shop
11 spec:
12   ports:
13      # the port that this service should serve on
14   - port: 5672
15     name: rabbitmq
16     targetPort: 5672
17   - port: 9090
18     name: exporter
19     targetPort: exporter
20     protocol: TCP
21   selector:
22     name: rabbitmq
```

---

**manifests/21-session-db-dep.yaml**

```
1  apiVersion: apps/v1
2  kind: Deployment
3  metadata:
4    name: session-db
5    labels:
6      name: session-db
7    namespace: sock-shop
8  spec:
9    replicas: 2
10   selector:
11     matchLabels:
12         name: session-db
```

```
13   template:
14     metadata:
15       labels:
16         name: session-db
17       annotations:
18         prometheus.io.scrape: "false"
19     spec:
20       containers:
21       - name: session-db
22         image: redis:alpine
23         ports:
24         - name: redis
25           containerPort: 6379
26         securityContext:
27           capabilities:
28             drop:
29               - all
30             add:
31               - CHOWN
32               - SETGID
33               - SETUID
34           readOnlyRootFilesystem: true
35       nodeSelector:
36         beta.kubernetes.io/os: linux
```

manifests/22-session-db-svc.yaml

```
1  apiVersion: v1
2  kind: Service
3  metadata:
4    name: session-db
5    labels:
6      name: session-db
7    namespace: sock-shop
8  spec:
9    ports:
10     # the port that this service should serve on
11   - port: 6379
12     targetPort: 6379
13   selector:
14     name: session-db
```

manifests/23-shipping-dep.yaml

```
1  apiVersion: apps/v1
2  kind: Deployment
3  metadata:
4    name: shipping
5    labels:
6      name: shipping
7    namespace: sock-shop
8  spec:
9    replicas: 2
10   selector:
11     matchLabels:
12       name: shipping
13   template:
14     metadata:
15       labels:
16         name: shipping
17     spec:
18       containers:
19       - name: shipping
20         image: weaveworksdemos/shipping:0.4.8
21         env:
22          - name: ZIPKIN
23            value: zipkin.jaeger.svc.cluster.local
24          - name: JAVA_OPTS
25            value: -Xms64m -Xmx128m -XX:+UseG1GC -Djava.security.egd=file:/dev/urandom -
                   Dspring.zipkin.enabled=false
26         resources:
27           limits:
```

```
28              cpu: 300m
29              memory: 500Mi
30            requests:
31              cpu: 100m
32              memory: 300Mi
33          ports:
34          - containerPort: 80
35          securityContext:
36            runAsNonRoot: true
37            runAsUser: 10001
38            capabilities:
39              drop:
40                - all
41              add:
42                - NET_BIND_SERVICE
43            readOnlyRootFilesystem: true
44          volumeMounts:
45          - mountPath: /tmp
46            name: tmp-volume
47        volumes:
48          - name: tmp-volume
49            emptyDir:
50              medium: Memory
51        nodeSelector:
52          beta.kubernetes.io/os: linux
```

### manifests/24-shipping-svc.yaml

```
1  apiVersion: v1
2  kind: Service
3  metadata:
4    name: shipping
5    annotations:
6        prometheus.io/scrape: 'true'
7    labels:
8      name: shipping
9    namespace: sock-shop
10 spec:
11   ports:
12     # the port that this service should serve on
13   - port: 80
14     targetPort: 80
15   selector:
16     name: shipping
```

### manifests/25-user-dep.yaml

```
1  apiVersion: apps/v1
2  kind: Deployment
3  metadata:
4    name: user
5    labels:
6      name: user
7    namespace: sock-shop
8  spec:
9    replicas: 2
10   selector:
11     matchLabels:
12       name: user
13   template:
14     metadata:
15       labels:
16         name: user
17     spec:
18       containers:
19       - name: user
20         image: weaveworksdemos/user:0.4.7
21         resources:
22           limits:
23             cpu: 300m
24             memory: 200Mi
25           requests:
```

```
26              cpu: 100m
27              memory: 100Mi
28          ports:
29          - containerPort: 80
30          env:
31          - name: mongo
32            value: user-db:27017
33          securityContext:
34            runAsNonRoot: true
35            runAsUser: 10001
36            capabilities:
37              drop:
38                - all
39              add:
40                - NET_BIND_SERVICE
41            readOnlyRootFilesystem: true
42          livenessProbe:
43            httpGet:
44              path: /health
45              port: 80
46            initialDelaySeconds: 300
47            periodSeconds: 3
48          readinessProbe:
49            httpGet:
50              path: /health
51              port: 80
52            initialDelaySeconds: 180
53            periodSeconds: 3
54        nodeSelector:
55          beta.kubernetes.io/os: linux
```

manifests/26-user-svc.yaml

```
1 apiVersion: v1
2 kind: Service
3 metadata:
4   name: user
5   annotations:
6       prometheus.io/scrape: 'true'
7   labels:
8     name: user
9   namespace: sock-shop
10 spec:
11   ports:
12     # the port that this service should serve on
13   - port: 80
14     targetPort: 80
15   selector:
16     name: user
```

manifests/27-user-db-dep.yaml

```
1 apiVersion: apps/v1
2 kind: Deployment
3 metadata:
4   name: user-db
5   labels:
6     name: user-db
7   namespace: sock-shop
8 spec:
9   replicas: 2
10   selector:
11     matchLabels:
12       name: user-db
13   template:
14     metadata:
15       labels:
16         name: user-db
17     spec:
18       containers:
19       - name: user-db
20         image: weaveworksdemos/user-db:0.3.0
```

```
21          ports:
22          - name: mongo
23            containerPort: 27017
24          securityContext:
25            capabilities:
26              drop:
27                - all
28              add:
29                - CHOWN
30                - SETGID
31                - SETUID
32            readOnlyRootFilesystem: true
33          volumeMounts:
34          - mountPath: /tmp
35            name: tmp-volume
36        volumes:
37        - name: tmp-volume
38          emptyDir:
39            medium: Memory
40        nodeSelector:
41          beta.kubernetes.io/os: linux
```

manifests/28-user-db-svc.yaml

```
1  apiVersion: v1
2  kind: Service
3  metadata:
4    name: user-db
5    labels:
6      name: user-db
7    namespace: sock-shop
8  spec:
9    ports:
10     # the port that this service should serve on
11   - port: 27017
12     targetPort: 27017
13   selector:
14     name: user-db
```

### C.2.2 OUTPUTS

VaC script for the steady state #1 (carts-db-dep.yaml): At least 1 ready replica 100% of the time and 2 ready replicas 80% of the time during the monitoring period

```
1  import os
2  import time
3  import argparse
4  from kubernetes import client, config
5  from unittest_base import K8sAPIBase
6
7  class TestCartsDBReplicas(K8sAPIBase):
8      def __init__(self, namespace='sock-shop', deployment_name='carts-db', duration=5):
9          super().__init__()
10         self.namespace = namespace
11         self.deployment_name = deployment_name
12         self.duration = duration
13         # Use the correct API client for deployments
14         self.apps_v1 = client.AppsV1Api()
15
16     def test_replicas_threshold(self):
17         # Initialize counters for ready replicas
18         total_checks = 0
19         ready_replicas_count = 0
20         fully_ready_replicas_count = 0
21
22         for _ in range(self.duration):
23             try:
24                 # Read the deployment status using the correct API client
25                 deployment = self.apps_v1.read_namespaced_deployment(self.
                     deployment_name, self.namespace)
```

```
26                      replicas = deployment.status.replicas
27                      ready_replicas = deployment.status.ready_replicas
28                      print(f"Total replicas: {replicas}, Ready replicas: {ready_replicas}")
29
30                      # Increment the total checks
31                      total_checks += 1
32
33                      # Check if at least 1 replica is ready
34                      if ready_replicas >= 1:
35                          ready_replicas_count += 1
36
37                      # Check if both replicas are ready
38                      if ready_replicas == 2:
39                          fully_ready_replicas_count += 1
40
41                  except client.exceptions.ApiException as e:
42                      print(f"Exception when calling AppsV1Api->read_namespaced_deployment: {
                        e}")
43
44                  # Wait for 1 second before the next check
45                  time.sleep(1)
46
47          # Calculate the percentage of time conditions are met
48          one_ready_percentage = (ready_replicas_count / total_checks) * 100
49          two_ready_percentage = (fully_ready_replicas_count / total_checks) * 100
50
51          # Assert the threshold conditions
52          assert one_ready_percentage == 100, "At least 1 ready replica was not available
             100% of the time."
53          assert two_ready_percentage >= 80, "2 ready replicas were not available at
            least 80% of the time."
54
55          print("Test passed: Steady state conditions are satisfied.")
56
57
58 def main():
59     parser = argparse.ArgumentParser(description='Test carts-db replicas threshold')
60     parser.add_argument('--duration', type=int, default=5, help='Duration to check the
           replicas')
61     args = parser.parse_args()
62
63     # Create an instance of the test class with the specified duration
64     test = TestCartsDBReplicas(duration=args.duration)
65     # Run the test
66     test.test_replicas_threshold()
67
68
69 if __name__ == '__main__':
70     main()
```

VaC script for the steady state #2 (front-end-dep.yaml): At least 1 ready replica must be
present 100% of the time during the monitoring period

```
1  import os
2  import time
3  import argparse
4  from kubernetes import client, config
5  from unittest_base import K8sAPIBase
6
7  class TestFrontEndReplica(K8sAPIBase):
8      def __init__(self, namespace, deployment_name, duration):
9          super().__init__()
10         self.namespace = namespace
11         self.deployment_name = deployment_name
12         self.duration = duration
13         # Use AppsV1Api to interact with deployments
14         self.apps_v1 = client.AppsV1Api()
15
16     def test_steady_state(self):
17         ready_replicas_count = 0
18
19         # Loop for the specified duration
20         for _ in range(self.duration):
21             # Get the deployment status
```

```
22              resp = self.apps_v1.read_namespaced_deployment_status(self.deployment_name,
                 self.namespace)
23              ready_replicas = resp.status.ready_replicas or 0
24              print(f"Ready replicas for {self.deployment_name}: {ready_replicas}")
25
26              # Check if the number of ready replicas is at least 1
27              if ready_replicas >= 1:
28                  ready_replicas_count += 1
29
30              # Wait for 1 second before the next check
31              time.sleep(1)
32
33          # Calculate the percentage of time the deployment was ready
34          readiness_percentage = (ready_replicas_count / self.duration) * 100
35          print(f"{self.deployment_name} was ready {ready_replicas_count}/{self.duration}
             times.")
36
37          # Assert that the deployment was ready 100% of the time
38          assert readiness_percentage == 100, f"{self.deployment_name} readiness was {
             readiness_percentage}%, expected 100%."
39
40
41  def main():
42      parser = argparse.ArgumentParser(description='Test front-end replica readiness')
43      parser.add_argument('--duration', type=int, default=5, help='Duration to check the
            replicas')
44      args = parser.parse_args()
45
46      # Create a test instance and run the test
47      test = TestFrontEndReplica('sock-shop', 'front-end', args.duration)
48      test.test_steady_state()
49
50
51  if __name__ == '__main__':
52      main()
```

### Chaos Mesh workflow manifest

```
1  apiVersion: chaos-mesh.org/v1alpha1
2  kind: Workflow
3  metadata:
4    name: chaos-experiment-20241127-045539
5  spec:
6    entry: the-entry
7    templates:
8      #-----------------------------
9      # entry point of whole workflow
10     #-----------------------------
11     - name: the-entry
12       templateType: Serial
13       deadline: 30m45s
14       children:
15         - pre-validation-phase
16         - fault-injection-phase
17         - post-validation-phase
18
19     #------------------------------------
20     # Entry point of pre-validation-phase
21     #------------------------------------
22     - name: pre-validation-phase
23       templateType: Serial
24       deadline: 10m20s
25       children:
26         - pre-validation-parallel-workflows
27
28     - name: pre-validation-parallel-workflows
29       templateType: Parallel
30       deadline: 5m20s
31       children:
32         - pre-unittest-carts-db-replicas
33         - pre-unittest-front-end-replica
34
35     # Definitions of children of pre-validation-phase
36     - name: pre-unittest-carts-db-replicas
37       templateType: Task
```

```
38      deadline: 5m20s
39      task:
40        container:
41          name: pre-unittest-carts-db-replicas-container
42          image: chaos-eater/k8sapi:1.0
43          imagePullPolicy: IfNotPresent
44          command: ["/bin/bash", "-c"]
45          args: ["python /chaos-eater/sandbox/cycle_20241127_043136/unittest_carts-db-
                  replicas_mod0.py --duration 20"]
46          volumeMounts:
47            - name: pvc-volume
48              mountPath: /chaos-eater
49        volumes:
50          - name: pvc-volume
51            persistentVolumeClaim:
52              claimName: pvc
53
54    - name: pre-unittest-front-end-replica
55      templateType: Task
56      deadline: 5m20s
57      task:
58        container:
59          name: pre-unittest-front-end-replica-container
60          image: chaos-eater/k8sapi:1.0
61          imagePullPolicy: IfNotPresent
62          command: ["/bin/bash", "-c"]
63          args: ["python /chaos-eater/sandbox/cycle_20241127_043136/unittest_front-end-
                  replica_mod0.py --duration 20"]
64          volumeMounts:
65            - name: pvc-volume
66              mountPath: /chaos-eater
67        volumes:
68          - name: pvc-volume
69            persistentVolumeClaim:
70              claimName: pvc
71
72    #-------------------------------------
73    # Entry point of fault-injection-phase
74    #-------------------------------------
75    - name: fault-injection-phase
76      templateType: Serial
77      deadline: 10m15s
78      children:
79        - fault-injection-overlapped-workflows
80
81    - name: fault-injection-parallel-workflow
82      templateType: Parallel
83      deadline: 5m10s
84      children:
85        - fault-unittest-carts-db-replicas
86        - fault-stresschaos
87
88    - name: fault-injection-suspend-workflow
89      templateType: Serial
90      deadline: 5m15s
91      children:
92        - fault-injection-suspend
93        - fault-injection-parallel-workflows
94
95    - name: fault-injection-suspend
96      templateType: Suspend
97      deadline: 10s
98
99    - name: fault-injection-parallel-workflows
100     templateType: Parallel
101     deadline: 5m5s
102     children:
103       - fault-unittest-front-end-replica
104       - fault-podchaos
105
106   - name: fault-injection-overlapped-workflows
107     templateType: Parallel
108     deadline: 5m15s
109     children:
110       - fault-injection-parallel-workflow
111       - fault-injection-suspend-workflow
112
113   # Definitions of children of pre-validation-phase
```

```
114      # unit tests
115      - name: fault-unittest-carts-db-replicas
116        templateType: Task
117        deadline: 5m10s
118        task:
119          container:
120            name: fault-unittest-carts-db-replicas-container
121            image: chaos-eater/k8sapi:1.0
122            imagePullPolicy: IfNotPresent
123            command: ["/bin/bash", "-c"]
124            args: ["python /chaos-eater/sandbox/cycle_20241127_043136/unittest_carts-db-
                 replicas_mod0.py --duration 10"]
125            volumeMounts:
126              - name: pvc-volume
127                mountPath: /chaos-eater
128            volumes:
129              - name: pvc-volume
130                persistentVolumeClaim:
131                  claimName: pvc
132
133      - name: fault-unittest-front-end-replica
134        templateType: Task
135        deadline: 5m5s
136        task:
137          container:
138            name: fault-unittest-front-end-replica-container
139            image: chaos-eater/k8sapi:1.0
140            imagePullPolicy: IfNotPresent
141            command: ["/bin/bash", "-c"]
142            args: ["python /chaos-eater/sandbox/cycle_20241127_043136/unittest_front-end-
                 replica_mod0.py --duration 5"]
143            volumeMounts:
144              - name: pvc-volume
145                mountPath: /chaos-eater
146            volumes:
147              - name: pvc-volume
148                persistentVolumeClaim:
149                  claimName: pvc
150
151      # fault_injections
152      - name: fault-stresschaos
153        templateType: StressChaos
154        deadline: 10s
155        stressChaos:
156          containerNames:
157            - carts-db
158          mode: all
159          selector:
160            labelSelectors:
161              name: carts-db
162            namespaces:
163              - sock-shop
164          stressors:
165            cpu:
166              load: 80
167              workers: 2
168
169
170      - name: fault-podchaos
171        templateType: PodChaos
172        deadline: 5s
173        podChaos:
174          action: pod-kill
175          mode: one
176          selector:
177            labelSelectors:
178              name: front-end
179            namespaces:
180              - sock-shop
181          value: '1'
182
183
184      #-------------------------------------
185      # Entry point of post-validation phase
186      #-------------------------------------
187      - name: post-validation-phase
188        templateType: Serial
189        deadline: 10m10s
```

```
190        children:
191          - post-validation-parallel-workflows
192
193     - name: post-validation-parallel-workflows
194       templateType: Parallel
195       deadline: 5m10s
196       children:
197          - post-unittest-carts-db-replicas
198          - post-unittest-front-end-replica
199
200     # Definitions of children of pre-validation-phase
201     - name: post-unittest-carts-db-replicas
202       templateType: Task
203       deadline: 5m10s
204       task:
205         container:
206           name: post-unittest-carts-db-replicas-container
207           image: chaos-eater/k8sapi:1.0
208           imagePullPolicy: IfNotPresent
209           command: ["/bin/bash", "-c"]
210           args: ["python /chaos-eater/sandbox/cycle_20241127_043136/unittest_carts-db-
                  replicas_mod0.py --duration 10"]
211           volumeMounts:
212             - name: pvc-volume
213               mountPath: /chaos-eater
214           volumes:
215             - name: pvc-volume
216               persistentVolumeClaim:
217                 claimName: pvc
218
219     - name: post-unittest-front-end-replica
220       templateType: Task
221       deadline: 5m10s
222       task:
223         container:
224           name: post-unittest-front-end-replica-container
225           image: chaos-eater/k8sapi:1.0
226           imagePullPolicy: IfNotPresent
227           command: ["/bin/bash", "-c"]
228           args: ["python /chaos-eater/sandbox/cycle_20241127_043136/unittest_front-end-
                  replica_mod0.py --duration 10"]
229           volumeMounts:
230             - name: pvc-volume
231               mountPath: /chaos-eater
232           volumes:
233             - name: pvc-volume
234               persistentVolumeClaim:
235                 claimName: pvc
```

### Reconfigured 09-front-end-dep.yaml

```
1 apiVersion: apps/v1
2 kind: Deployment
3 metadata:
4   name: front-end
5   namespace: sock-shop
6 spec:
7   replicas: 2
8   selector:
9     matchLabels:
10       name: front-end
11   template:
12     metadata:
13       labels:
14         name: front-end
15     spec:
16       containers:
17       - name: front-end
18         image: weaveworksdemos/front-end:0.3.12
19         resources:
20           limits:
21             cpu: 300m
22             memory: 1000Mi
23           requests:
24             cpu: 100m
```

```
25            memory: 300Mi
26        ports:
27        - containerPort: 8079
28        env:
29        - name: SESSION_REDIS
30          value: "true"
31        securityContext:
32          runAsNonRoot: true
33          runAsUser: 10001
34          capabilities:
35            drop:
36              - all
37          readOnlyRootFilesystem: true
38        livenessProbe:
39          httpGet:
40            path: /
41            port: 8079
42          initialDelaySeconds: 300
43          periodSeconds: 3
44        readinessProbe:
45          httpGet:
46            path: /
47            port: 8079
48          initialDelaySeconds: 30
49          periodSeconds: 3
50      nodeSelector:
51        beta.kubernetes.io/os: linux
```