# OpenReview forum: "ChaosEater: Fully Automating Chaos Engineering with Large Language Models"
_ICLR.cc/2025/Conference — Submitted to ICLR 2025_

### Official Review · Reviewer_aLZw · 2024-10-28

**Soundness:** 2
**Presentation:** 4
**Contribution:** 2
**Rating:** 6
**Confidence:** 2

**Summary:**

This paper proposes ChaosEater, an automatic framework for Chaos Engineering (CE) operators. In particular, each CE operator is automated using one or multiple LLM agents, equipped with carefully crafted system/user/AI prompts. The experimental results demonstrate that the proposed system significantly reduces both time and monetary costs within the CE cycle.

**Strengths:**

- The paper presents an intriguing application of LLM agents in improving distributed system resilience.
- The framework effectively automates the CE process, minimizing the need for manual intervention.
- The authors provide the implementation of the proposed methods, which could be beneficial to the research community.
- The paper is well-organized, with a clear and logical flow.

**Weaknesses:**

- It is unclear to what extent ChaosEater reduces the reliance on human expertise. For example, steady-state selection still requires experts to define measurable states, with the agent only used for state selection.
- Both steady-state selection and failure injection are determined by LLM agents, whose inherent biases could hinder the discovery of new issues.
- Figure 7 is difficult to comprehend; it could be better to include a summarized version in the main text and move the detailed figure to the Appendix.
- The experiments seem limited to a toy example, which may not fully demonstrate the effectiveness of ChaosEater.

**Questions:**

- Is the LLM primarily used in most phases to adjust parameters within predefined templates (e.g., fault scopes)? Could a smaller model be employed in the experiment instead?
- The prompts in ChaosEater appear to be carefully designed, resulting in well-structured LLM responses. I suggest providing some detailed prompts in the Appendix.
- In addition, what is the cost of prompt tuning, and can the agent maintain robustness when using other LLM models or in different environments?

---

> ### Author Response · Authors · 2024-11-19
> **Response to Reviewer aLZw [1/2]**
>
> Dear reviewer aLZw
>
> Thank you for your time, valuable suggestions, and important questions. We have answered your concerns in the following. We hope our answers address all your concerns. If you have remaining questions/concerns, please feel free to raise them for further discussion.
>
> ---
> > **W1**: It is unclear to what extent ChaosEater reduces the reliance on human expertise. For example, steady-state selection still requires experts to define measurable states, with the agent only used for state selection.
>
> **Answer (A1)**
>
> First of all, please let us clarify again that ChaosEater can complete all the operations in a CE cycle without any user intervention. The case study demonstrates that ChaosEater can autonomously define the hypothesis and reconfigure problematic K8s manifests following general best practices. Therefore, ChaosEater enables the users to complete a generally reasonable CE cycle without requiring any human expertise. However, to improve its personalization, a user intervention may be required. For example, if the manifests include implicit user intentions or conventions and constraints that are not explicitly stated and are deviating from the general practices, the user must explicitly include such intentions in their input instructions to guide the LLM agent inside of ChaosEater.
>
> ---
> > **W2**: Both steady-state selection and failure injection are determined by LLM agents, whose inherent biases could hinder the discovery of new issues.
>
> **Answer (A2)**
>
> We agree that with a single CE cycle, those biases could hinder the discovery of new issues. However, we believe that multiple CE cycles would mitigate those biases. We can easily come up with some strategies to increase the diversity of hypotheses (i.e., steady states + failure injections) through multiple CE cycles. For example, in each CE cycle, include a user instruction to propose a hypothesis in the next CE cycle that differs from those proposed in the previous cycles. As a result, the diversity of proposed hypotheses is expected to increase over multiple CE cycles. By forcibly exploring various directions in this manner, we believe it is possible to increase the diversity of hypotheses and reduce biases that tend to favor specific hypotheses. By the way, we discussed a similar topic regarding diversity in the response to reviewer ujQS (W1+Q2). Please see it as well if necessary.
>
> ---
> > **W3**: Figure 7 is difficult to comprehend; it could be better to include a summarized version in the main text and move the detailed figure to the Appendix.
>
> **Answer (A3)**
>
> Thank you for your suggestion, and we are sorry for the inconvenience. We are revising that part to improve the presentation of the case study section, and moving the full version to the Appendix. Please wait a little longer.

---

> > ### Author Response · Authors · 2024-11-19
> > **Response to Reviewer aLZw [2/2]**
> >
> > > **W4**: The experiments seem limited to a toy example, which may not fully demonstrate the effectiveness of ChaosEater.
> >
> > **Answer (A4)**
> >
> > We are currently evaluating ChaosEater on the sock-shop server [1], which is a much lager, practical system consisting of 28 manifests (resources) and over 800 lines in total.  We will add the results to the revised manuscript, so please wait a little longer.
> >
> > ---
> > > **Q1**: Is the LLM primarily used in most phases to adjust parameters within predefined templates (e.g., fault scopes)? Could a smaller model be employed in the experiment instead?
> >
> > **Answer (A5)**
> >
> > LLMs are used to adjust parameters for predefined templates in the following four cases:
> >
> > - when determining the duration of inspecting the current states in the inspection phase
> > - when specifying the failure types and their detailed parameters in the failure-definition phase
> > - when determining detailed schedules of failure injections and steady-state validation, such as the ```grace_period``` and ```duration```, in the experiment-planning phase
> > - when adjusting the failure scopes in the experiment-replanning phase
> >
> > Inspection scripts, VaC scripts, and reconfigured K8s manifests are generated/debugged from scratch. Here, generating inspection scripts and VaC scripts from scratch allows LLMs to flexibly determine actions (i.e., inspection and steady-state validation) using specified tools such as the K8s API and k6 through code. Other sentences, such as thoughts, descriptions, and summaries, are freely written in specified formats. Therefore, in most phases, the LLMs’ responses are less constrained than parameter adjustment, where values are selected from a predefined range.
> >
> > ChaosEater requires LLM agent capabilities, such as tool usage and JSON output, as well as a long context window. In general, relatively smaller LLMs tend to have less agent capabilities [2], and require additional finetuning for specific tool usages. Therefore, we believe that it is difficult for smaller models to complete a CE cycle with satisfactory quality without additional finetuning. We are not sure if we can make it during this discussion period, but we would try existing smaller models (≤ 13B) as well when trying different LLMs to answer Q3.
> >
> > ---
> > > **Q2**: The prompts in ChaosEater appear to be carefully designed, resulting in well-structured LLM responses. I suggest providing some detailed prompts in the Appendix.
> >
> > **Answer (A6)**
> >
> > Thank you for your suggestion. We are adding all our system prompts to the Appendix, so please wait a little longer.
> >
> > ---
> > > **Q3**: In addition, what is the cost of prompt tuning, and can the agent maintain robustness when using other LLM models or in different environments?
> >
> > **Answer (A7)**
> >
> > Our system prompts are manually tuned using actual LLMs’ responses as feedback, so the cost would be the API billing incurred to obtain those responses. As we did not precisely track all API usage for this project, we cannot provide the exact cost estimate, but it is very roughly over $100.
> >
> > In the production environment, some additional components are required, such as high-level monitoring functions to oversee ChaosEater's operations to allow ChaosEater to be immediately stopped in case of emergencies, as well as more mature designs for the impact scope of fault injections (i.e., the blast radius design) to avoid affecting the actual services. However, our system prompts can be commonly used in both development and production environments, and the behavior of the ChaosEater should remain the same in both environments.
> >
> > On the other hand, the behavior of ChaosEater is expected to change depending on the LLMs used. To evaluate our system's robustness to the LLMs used, we plan to add the results when using different LLMs, such as Claude, Gemini, etc., to the revised manuscript. Please wait a little longer.
> > Additionally, please let us excuse the robustness in advance. From an engineering perspective, focusing on a specific tool (i.e., LLM) is an effective strategy to reduce the cost of managing system components (i.e., prompts for better responses). Our system prompts are, in fact, highly tuned for GPT-4o.  Therefore, the results would degrade to some extent when using different LLMs. From the user side and research perspectives, it is, of course, preferable to ensure robustness across various LLMs. However, we would like to emphasize that the lack of such robustness does not mean that our system is useless in practice.
> >
> > ---
> > [1] https://github.com/microservices-demo/microservices-demo
> >
> > [2] Y. Li et al., Personal LLM Agents: Insights and Survey about the Capability, Efficiency and Security, https://arxiv.org/abs/2401.05459

---

> > > ### Comment · Reviewer_aLZw · 2024-11-24
> > >
> > > Thank you for your clarification. I believe it provides a strong example of LLM agents in software engineering, and as a result, I have increased the score to 6.

---

> ### Author Response · Authors · 2024-11-25
> **Thank you for your response**
>
> Dear reviewer aLZw
>
> Thank you for reading our response and raising our score!
> We are very pleased that you have recognized our work as a promising example of LLM agents in software engineering!
> If you have any additional questions or concerns, please feel free to ask at any time (As for the revision of the manuscript, please wait a little longer).
>
> Sincerely,
> Authors

---

> ### Author Response · Authors · 2024-11-30
> **Notification of the Revised Manuscript**
>
> Dear reviewer aLZw
>
> We apologize for bothering you repeatedly, but please allow us to inform you about the revised manuscript and the discussion summary.
> The revised manuscript includes an additional case study of larger system and our system prompte tempates, which are related to your concerns/suggestions (W4 and Q2).
>
> Regarding the robustness across different LLMs, unfortunately, ChaosEater encounters runtime errors with different LLMs such as Calude Sonnet 3.5 and Gemini 1.5 pro. In the revised manuscript, we conclude that ChaosEater does not currently support other LLMs in the Limitations section, and discuss the necessity of automatic prompt tuning in the Future directions section.
>
> If you have time, we would appreciate it if you could also see the revised manuscript and our general response. Of course, we are always open to any additional questions or feedback you may have!
>
> Sincerely,
> Authors

---

### Official Review · Reviewer_iWjm · 2024-10-29

**Soundness:** 2
**Presentation:** 2
**Contribution:** 2
**Rating:** 5
**Confidence:** 4

**Summary:**

The author proposes a system called ChaosEater, primarily designed to automate the Chaos Engineering (CE) workflow using LLMs. It provides an efficient and low-cost solution for maintaining system resilience, offering potential for automated resilience testing and fault remediation in future complex distributed systems.

**Strengths:**

The article leverages LLMs to automate various stages of chaos experiments, thereby reducing manual operations. This innovative application demonstrates the potential of LLMs in infrastructure-as-code and fault injection testing, showcasing a degree of novelty.

**Weaknesses:**

1.The current version of ChaosEater relies on LLMs to handle complex input and output, particularly in multi-stage and multi-dependency chaos experiments, which can be susceptible to context length limitations. Even with models like GPT-4, longer contexts may still lead to information truncation, affecting the accuracy and comprehensiveness of the experiments.
2.The types of failures and injection methods currently supported by ChaosEater may be too limited to cover all potential faults in distributed systems. For example, the system may lack support for specific failure injections related to storage systems, such as disk latency or database locking, as well as certain network issues like packet loss and jitter. Furthermore, the paper does not mention support for various complex and dynamic failure scenarios, such as cross-injection of multiple faults, cascading failures, or the failure of partially dependent components, all of which are quite common in complex systems.
3.Chaos experiments typically require real-time monitoring and response to system states to promptly terminate the experiment in the event of severe anomalies or catastrophic failures. However, the paper does not mention that ChaosEater has such real-time monitoring capabilities, which limits its ability to dynamically adjust strategies during the experiment.

**Questions:**

The main points and issues have been outlined in the "Weaknesses" section.

---

> ### Author Response · Authors · 2024-11-19
> **Response to Reviewer iWjm [1/3]**
>
> Dear reviewer iWjm
>
> Thank you for your time and important questions. In the following, we have answered your concerns and corrected some misunderstandings. We hope our answers address your concerns. If you have remaining questions/concerns, please feel free to raise them for further discussion.
>
> ---
> > **W1**: The current version of ChaosEater relies on LLMs to handle complex input and output, particularly in multi-stage and multi-dependency chaos experiments, which can be susceptible to context length limitations. Even with models like GPT-4, longer contexts may still lead to information truncation, affecting the accuracy and comprehensiveness of the experiments.
>
> **Short Answer (A1)**
>
> ChaosEater does not fully resolve the long context issues but mitigates them through workflow and prompt-engineering-level approaches. As a result, ChaosEater is capable of completing a reasonable CE cycle for the systems tested, where numerous stages exist and the input context grows gradually.
>
> **Details**
>
> Issues such as information loss in long contexts have been actively studied in the LLM community, and we understand that no complete solution has yet been established. ChaosEater has inherited those issues as it uses existing LLMs without additional fine-tuning. However, ChaosEater mitigates those issues by employing some workflow- and prompt-engineering-level approaches:
>
> - Placing critical information at the beginning or end of the input context.
> - Dividing tasks into detailed sub-tasks for LLM agents (lines 133-134). It contributes to reducing input context.
> - Designed chat history management: Instead of simply appending all previous data and agent outputs to the conversation history to create the next agent’s prompt, we create a new conversation for each agent every time and embed the organized previous data and agent outputs within it (lines 159-161).
>
> As a result, ChaosEater demonstrates that it can complete a reasonable CE cycle, where many stages exist, and the input context becomes increasingly long. While the initial manuscript focused on a simple system, we have additionally tested ChaosEater on a much larger, real-world system (which significantly increases the input context) and confirmed that it works in those cases as well. We are currently revising the manuscript to include this, so please wait a little longer.
>
> Overall, ChaosEater does not completely solve the long context issues, and there is a possibility that those issues may become pronounced in unseen systems. However, the current version of ChaosEater is already capable of handling long contexts at a level that demonstrates its potential for the future.
>
> ---
> > **W2-1**: The types of failures and injection methods currently supported by ChaosEater may be too limited to cover all potential faults in distributed systems. For example, the system may lack support for specific failure injections related to storage systems, such as disk latency or database locking, as well as certain network issues like packet loss and jitter.
>
> **Short Answer (rebuttal) (A2)**
>
> ChaosEater already supports a sufficient variety of failure types to simulate actual failure scenarios, including the failures that were pointed out as missing support in the raised concerns.
>
> **Details**
>
> ChaosEater supports all the failure types supported by Chaos Mesh, except for kernelChaos. Seven types of failures are supported in Chaos Mesh/ChaosEater:
>
> - ```PodChaos```: simulates Pod failures, such as Pod node restart, Pod's persistent unavailability, and certain container failures in a specific Pod.
> - ```NetworkChaos```: simulates network failures, such as network latency, packet loss, packet disorder, and network partitions.
> - ```DNSChaos```: simulates DNS failures, such as the parsing failure of DNS domain name and the wrong IP address returned.
> - ```HTTPChaos```: simulates HTTP communication failures, such as HTTP communication latency.
> - ```StressChaos```: simulates CPU race or memory race.
> - ```IOChaos```: simulates the I/O failure of an application file, such as I/O delays, read and write failures.
> - ```TimeChaos```: simulates the time jump exception.
>
> ```IOChaos``` and ```NetworkChaos``` can simulate disk latency/locking and network issues respectively, which were pointed out as missing support in the raised concerns.
>
> Moreover, Chaos Mesh provides more detailed parameters for each of the seven failures, which can more flexibly control the failure behaviors.  ChaosEater supports all the detailed parameters as well; therefore, we believe that ChaosEater already supports sufficiently comprehensive failures to simulate various failure scenarios.
>
> **Additional comments**
>
> In the supplementary code, the Python scripts in the ```chaos_eater/chaosmesh/faults/``` directory define the supported Chaos Mesh failures and their detailed parameters as Pydantic objects (which are used to extract the JSON outputs of the detailed parameters from LLMs). Please also check them if you are interested.

---

> > ### Comment · Reviewer_jkoh · 2024-11-25
> > **Failure types**
> >
> > Although ChaosEater supports creation of all the failure types listed, it seems to me that it cannot propose solutions for some of these failure types. It can only suggest solutions via changes to K8s manifest files as you noted in your response to me.
> > For example, although ChaosEater can simulate IOChaos, this failure may not be solvable without changing application or other backend layer code.
> >
> > In your response to me, you seemed to imply that K8s manifest changes can handle all the backend resiliency issues. I don't believe this is true. It depends a lot on how the system is architected and what resiliency parts are relegated to K8s vs what is handled inside the backend system itself. It also depends on how resiliency is defined. In your own example, the resiliency definition of "the Pod being in the Running state for more than 95% of the monitoring period" excluded one suggested approach. Similarly resiliency definitions may require solutions that change the backend system code rather than just K8s config. ChaosEater would not be able to suggest solutions to such failures.

---

> ### Author Response · Authors · 2024-11-19
> **Response to Reviewer iWjm [2/3]**
>
> > **W2-1**: Furthermore, the paper does not mention support for various complex and dynamic failure scenarios, such as cross-injection of multiple faults, cascading failures, or the failure of partially dependent components, all of which are quite common in complex systems.
>
> **Short answer (rebuttal) (A3)**
>
> ChaosEater already supports complex failure injections, which are realized by combinations of parallel and sequential multi-failure injections.
>
> **Details**
>
> The ‘Failure definition’ section (lines 222-226) and ‘Experiment planning’ section (lines 265-280 + Figure 5) discuss how ChaosEater realizes complex failure injections. Here, please let us redescribe this implementation briefly:
>
> - First, an LLM agent outputs a 2D list representing the sequence of failure injections.
> The inner lists involve failures injected concurrently, and the outer list represents the injection order of each concurrent failure set. For example, [[```StressChaos```, ```NetworkChaos```], [```PodChaos```]] represents that ```PodChaos``` is injected after simultaneously injecting both ```StressChaos``` and ```NetworkChaos```.
> - Second, another LLM agent details the sequence of failure injections by determining parameters such as ```grace_period``` and ```duration``` (VaC script execution is also scheduled here). They are determined based on the injection order defined by the 2D list. The ```grace_period``` determines the timing for starting each fault injection, while ```duration``` determines the duration of each fault injection. Here, there are no restrictions on overlaps in injection timing, allowing for highly flexible planning of the fault injection schedule. For example, it is also possible to inject ```PodChaos``` a certain amount of time after ```NetworkChaos``` was first injected, while it remains active.
> - Finally, our rule-based algorithm converts the LLM agent’s plan into a Chaos Mesh workflow manifest, which enables automatic execution of all the scheduled failures and steady-state validation (i.e., VaC script execution).
>
> As described above, ChasEater already supports flexible failure injection patterns, such as Intertwined sequential and parallel fault injections.
>
> Additionally, for “the failure of partially dependent components” in the raised concern, Chaos Mesh can control the failure injection scope, and can inject failures focused on a specific part. The scope can be configured in detail at various levels, including namespaces, resource types, metadata, statuses, etc. (See ```chaos_eater/chaosmesh/faults/selectors.py``` for more details). Therefore, ChaosEater, which supports most of the functions of Chaos Mesh, can also inject such failures by setting an appropriate scope in the failure-definition phase (lines 227-239).
>
> **Additional comments**
>
> We believe that the abovementioned implementation already covers most types of failure injection patterns. However, we do not have 100% confidence in understanding the exact definitions of the "cross-injection" and "cascading failures" that you mentioned. So, we are willing to discuss them further if our current answer does not cover your concerns. In that case, we would appreciate it if you would provide their more detailed definitions so that we can respond in a way that meets your expectations.

---

> ### Author Response · Authors · 2024-11-19
> **Response to Reviewer iWjm [3/3]**
>
> > **W3**: Chaos experiments typically require real-time monitoring and response to system states to promptly terminate the experiment in the event of severe anomalies or catastrophic failures. However, the paper does not mention that ChaosEater has such real-time monitoring capabilities, which limits its ability to dynamically adjust strategies during the experiment.
>
> **Clarification Notice**
>
> We are sorry, but we could not understand which case corresponds to the "real-time monitoring" you mentioned: while the first sentence seems to be Case A, the second one seems to be Case B. Therefore, we answer your concerns from the perspectives of both Case A and Case B. We would appreciate it if you would provide their more detailed definitions if we are still misunderstanding its meaning.
>
> - Case A: higher-level real-time monitoring of the CE cycles that ChaosEater conducts
> - Case B: real-time monitoring of steady states in the experiment phase
>
> **Answer: Case A (A4-a)**
>
> ChaosEater does not have a higher-level real-time monitoring function to oversee its own operations. However, ChaosEater currently supports only development environments, so this is not a significant issue. A development environment is always an isolated, resettable sandbox. Therefore, there is no need to use such a higher-level monitoring function to ensure security.
>
> On the other hand, it will be required when adapting ChaosEater to production environments. If fault injections through Chaos Engineering impact an unexpected scope in a production environment, it could significantly affect the actual service and its users. In such cases, having a monitoring function for ChaosEater's operations is necessary to enable an emergency stop if necessary. As discussed in the 'feature directions' section, we are considering adding these functions to ensure that ChaosEater can be safely used even in production environments.
>
> **Answer: Case B (A4-b)**
>
> ChaosEater does not support the dynamical adjustment of monitoring and validation strategies during the experiment phase. However, to complete a CE cycle SYSTEMATICALLY, the hypothesis (including monitoring and validation strategies) defined at the beginning phase should remain fixed throughout the CE cycle. Monitoring and validation strategies correspond to the processes to rigorously validate whether a hypothesis is satisfied. Therefore, if they change dynamically during the experiment, it means that the goal of a single CE cycle is altered midway. This is not appropriate as a SYSTEMATIC hypothesis testing process. A hypothesis defined at the beginning of a cycle should be maintained until the end of that cycle. If there is an issue with the hypothesis, a new one (i.e., new monitoring and validation strategies) should be defined in the next cycle, and this consistent process should be repeated. Therefore, ChaosEater does not support such dynamical adjustment. Note that during the hypothesis phase, it is already ensured that monitoring and validation strategies can be executed without any runtime errors.
>
> **Additional comments for Case B**
>
> Note that ChaosEater will not change the monitoring and validation strategies dynamically during the experiment. However, the monitoring and steady-state validation are executed in REAL TIME during the experiment phase using Validation as Code (VaC) scripts.

---

> ### Author Response · Authors · 2024-11-25
> **Response to reviewer jkoh: Can K8s manifest configure all backend settings? Can all Chaos Mesh failures be solved solely through K8s manifest reconfiguration?**
>
> Dear reviewer jkoh
>
> We appreciate your further discussion!
> As the title says, we answer your concerns by separating two topics:
>
> - Can K8s manifest configure all backend settings?
> - Can all Chaos Mesh failures be solved solely through K8s manifest reconfiguration?
>
> ---
>
> > Can K8s manifest configure all backend settings?
> >
>
> ChaosEater focuses on K8s-based systems in K8s clusters managed by kind, where most backend settings can be configured through K8s manifests, ranging from environment (i.e., cluster-node settings) to service mesh, permission management, volume mount, and deployed resources (e.g., Pod, Service). For example,  we can configure service mesh using K8s manifests (CRD) of Istio. we can also configure cluster nodes as follows:
>
> ```
> # This cluster has a single node (i.e., master node).
> # This K8s yaml can be found at `chaos-eater/k8s/kind_config.yaml` in the supplementary code.
> apiVersion: kind.x-k8s.io/v1alpha4
> kind: Cluster
> nodes:
> - role: control-plane
>   extraMounts:
>     - hostPath: ${PWD}
>       containerPath: /chaos-eater
>
> ```
>
> ChaosEater currently supports reconfigurations only for resources deployed in the K8s clusters, not for the clusters themselves. However, to the best of our knowledge, we understand that K8s manifests can TECHNICALLY (re-)configure almost everything in the backend, including the abovementioned examples, to improve the system's resiliency. Note that other cloud infrastructures provided by AWS, MS Azure, etc. might require additional configurations that cannot be managed through code.
>
> ---
>
> > Can all Chaos Mesh failures be solved solely through K8s manifest reconfiguration?
> >
>
> Since the types of failure patterns (i.e., failure types x resource types) are vast, we can not certainly say that K8s manifest reconfiguration for deployed resources can solve all failures. However, we believe that most failures can be addressed by somehow increasing the redundancy of the resources (even if it is not the optimal solution).
> For example, `IOChaos` is injected into a database Pod to simulate the I/O failure of an application file in the database, such as I/O delays, and read and write failures. We can handle the failure only by increasing the redundancy of the database Pod as follows:
>
> 1. Define a Deployment resource managing more than two replicas (i.e., database Pods)
> 2. In the Deployment's manifests, define ```readinessProbe``` that verifies whether a query to the database returns a valid response within a specified time limit.
> 3. If the verification fails, the replica is automatically removed from the load balancing targets, and another replica that is not affected by the failure will be routed instead.
>
> Although it is not guaranteed that ChaosEater always reach this solution, some measures can be taken solely through K8s manifest reconfiguration for deployed resources.
>
> Of course, reconfiguring the application and environment layers would also be necessary to improve the system's resiliency in the most optimal way. However, solely reconfiguring the deployed resources is often the most flexible and sufficient solution.
> Therefore, given that addressing all layers would require substantial effort and time, we prioritize the resource layer as the first step and leave the other layers for future work.
>
> We hope this answer addresses your concerns. If you still have any questions or concerns (e.g., exceptions that always require modifications beyond K8s manifests), please feel free to ask again!
>
> Sincerely,
> Authors

---

> > ### Comment · Reviewer_jkoh · 2024-11-28
> > **Thank you for your response**
> >
> > Thanks to the authors for their response.

---

> ### Author Response · Authors · 2024-11-30
> **Gentle Reminder (Deadline: Dec. 2nd (AoE))**
>
> Dear reviewer iWjm
>
> We appreciate your feedback. Since the discussion period is closing soon (Dec. 2nd (AoE)), we know you are busy, but we would greatly appreciate it if you could take the time to respond to our response. In our response, we have described:
> - ChaosEater already supports various failure types and failure modes to simulate (A2 and A3)
> - The lack of dynamic changes to the monitoring strategy is intentional to systematically perform CE cycles (A4-a)
>
> The revised manuscript includes a case study of a larger system. We believe this result supports that ChaosEater can mitigate the long-context issues (W1/A1).
>
> If you have time, please also take a look at our general response to catch up on the current state of our work.
>
> We are still looking forward to your response!
>
> Sincerely,
> Authors

---

> ### Author Response · Authors · 2024-12-03
> **Final Gentle Reminder (Deadline: Within Half a Day)**
>
> Dear reviewer iWjm
>
> We apologize for bothering you repeatedly.
> As the reviewer response period is ending in half a day, we would like to hear your thoughts on our response.
> We believe that our response addresses most of your concerns.
>
> If possible, could you kindly take a moment to provide your feedback?
>
> Sincerely,
> Authors

---

### Official Review · Reviewer_jtMF · 2024-10-31

**Soundness:** 2
**Presentation:** 3
**Contribution:** 2
**Rating:** 5
**Confidence:** 4

**Summary:**

The paper proposes a system, called CHAOSEATER, that automates the entire Chaos Engineering (CE) cycle using LLMs to enhance the resilience of distributed systems. It defines a structured CE process with five phases: hypothesis setting, chaos experimentation, analysis, improvement, and final review. CHAOSEATER using Infrastructure as Code (IaC) to manage system configurations. The paper show the proposed system help in reducing the time and cost compared to manual CE

**Strengths:**

1- The paper demonstrates how LLMs contribute to reducing time and costs in Chaos Engineering (CE).

2- The paper is well-written and includes clear figures and visualizations.

**Weaknesses:**

1- Clearly differentiate your fully automated technique for Chaos Engineering (CE) using LLMs from existing approaches like self-refinement and self-debugging. Provide specific details on what sets your method apart and its unique contributions.

2- Include accuracy metrics in Table 1 to present success and failure rates alongside time and cost. This will offer a more comprehensive view of the technique's effectiveness.

3- To position this work as a foundational effort in automating system resilience improvement, construct a larger dataset, establish robust evaluation metrics for correctness, and include baseline comparisons with prior automated approaches using LLMs.

4- Add a qualitative analysis and human evaluation component to assess the effectiveness of LLMs in Chaos Engineering tasks. This will strengthen the paper by showing how well LLMs perform in real-world scenarios.

**Questions:**

If the paper proposes an automated system for Chaos Engineering (CE), consider constructing a benchmark to evaluate multiple LLMs and compare different automated approaches. This could provide a more comprehensive assessment of the system's effectiveness and highlight its unique contributions.

---

> ### Author Response · Authors · 2024-11-19
> **Response to Reviewer jtMF [1/2]**
>
> Dear reviewer jtMF
>
> Thank you for your time, valuable suggestions, and important questions. We have answered your concerns in the following. We hope our answers address your concerns. If you have remaining questions/concerns, please feel free to raise them for further discussion.
>
> ---
> > **W1**: Clearly differentiate your fully automated technique for Chaos Engineering (CE) using LLMs from existing approaches like self-refinement and self-debugging. Provide specific details on what sets your method apart and its unique contributions.
>
> **Answer (A1)**
>
> First of all, please let us clarify our understanding and definitions of self-refinement and self-debugging. Self-refinement is a strategy where LLMs generate more refined outputs for a task by leveraging its previous outputs and any feedback provided on them. Self-debugging is a strategy where LLMs debug programming code by leveraging the previously implemented code having bugs and the error message encountered when executing the code (e.g., raw error messages, unit-test results, etc.) [1, 2]. Therefore, self-refinement is a more general concept, while self-debugging is a specific case of self-refinement specialized to programming code, where the previous outputs correspond to previously implemented code and the feedback corresponds to error messages. However, for the sake of convenience, the following discussion will refer to self-refinement as a strategy for refining non-code outputs such as thoughts and explanations, while self-debugging will retain its original meaning.
>
> In ChaosEater, self-refinement is used for the step-by-step definition of steady states and failure injections, while self-debugging is used for both the verification loops of debugging inspection scripts, VaC scripts, and failure injection manifests and the improvement loop of reconfiguring K8s manifests. The general concepts of self-refinement and self-debugging in ChaosEater are similar to existing works. However, the types of code being debugged (K8s manifests and k6 JavaScript) and the tasks being conducted in Chaos Engineering (steady-state definition, failure definition, etc.) are totally new.
>
> Moreover, self-refinement and self-debugging are merely subsets of ChaosEater, which also has additional components related to them. In particular, ChaosEater significantly differs from self-debugging in that it autonomously sets its own goals. In self-debugging, the goal (i.e., unit test) and the specifications of the function to be implemented (or already implemented) are provided in advance. It then debugs the code to satisfy the goal and the specification. In contrast, given a system, ChaosEater sets the goal (i.e., hypotheses/VaC scripts) appropriate for that system by itself, and debug the system to satisfy those goals. This additional task requires a more advanced level of automation, but ChaosEater was able to complete it reasonably on the tested system. We believe that this demonstration, showing the LLM's capabilities in debugging combined with self-goal setting, contributes not only to the Chaos Engineering community but also to the broader software engineering community. For example, when LLMs autonomously generate attested code completely from scratch, they need to determine their own goals and generate both code and unit tests to validate if the code satisfies the goals.
>
> [1] X. Chen et al., Teaching Large Language Models to Self-Debug, ICLR 2024: https://openreview.net/forum?id=KuPixIqPiq
>
> [2] L. Zhong et al., Debug like a Human: A Large Language Model Debugger via Verifying Runtime Execution Step by Step,  ACL 2024: https://aclanthology.org/2024.findings-acl.49/
>
> ---
> > **W2**: Include accuracy metrics in Table 1 to present success and failure rates alongside time and cost. This will offer a more comprehensive view of the technique's effectiveness.
>
> **Answer (A2)**
>
> Thank you for your suggestion. We plan to include two different accuracy metrics, the “completion rate” and “reconfiguration rate”, calculated from multiple runs, in Table 1. The former refers to the percentage of cases where ChaosEater successfully completes the CE cycle without runtime errors, while the latter represents the percentage of cases where ChaosEater not only completes the CE cycle but also successfully reconfigures the input system. We are revising our manuscript to include this, so please wait a little longer.

---

> ### Author Response · Authors · 2024-11-19
> **Response to Reviewer jtMF [2/2]**
>
> > **W3**: To position this work as a foundational effort in automating system resilience improvement, construct a larger dataset, establish robust evaluation metrics for correctness, and include baseline comparisons with prior automated approaches using LLMs.
>
> > **Q1**: If the paper proposes an automated system for Chaos Engineering (CE), consider constructing a benchmark to evaluate multiple LLMs and compare different automated approaches. This could provide a more comprehensive assessment of the system's effectiveness and highlight its unique contributions.
>
> **Answer (A3)**
>
> Thank you for your suggestion. We understand the importance of constructing evaluation frameworks, such as large datasets, novel metrics, and baselines. However, considering the page limit\* and the importance of promptly sharing non-trivial efforts, we decided to separate our contributions into two different papers: a system-architecture side paper and an evaluation-framework side paper.
>
> This paper corresponds to the former, and focuses on presenting the system architecture and showing its potential in the new field through case studies. Even without comprehensive analysis by sophisticated evaluation frameworks, sharing the detailed system architecture and discussing its effectiveness and limitations through case studies can sufficiently contribute to the recognition of this emerging field and provide valuable guidance for subsequent research. Additionally, sharing these efforts promptly is crucial to maximizing its effectiveness. This is why we have first submitted the system-architecture side paper to this conference.
>
> As mentioned earlier, we also understand the importance of constructing evaluation frameworks. In fact, we are working on them to provide more solid contributions to the emerging community. On the other hand, we would greatly appreciate it if you could recognize our contributions from the perspective outlined above.
>
> \* This conference does not limit the Appendix pages. However, due to its importance, we believe that an effort to construct such evaluation frameworks must be presented as the main content.
>
> ---
> > **W4**: Add a qualitative analysis and human evaluation component to assess the effectiveness of LLMs in Chaos Engineering tasks. This will strengthen the paper by showing how well LLMs perform in real-world scenarios.
>
> **Answer (A4)**
>
> We hope that the case study offers some qualitative insights on ChaosEater: the reasonabilities of each operation in the CE cycle. However, we understand that user studies are necessary to evaluate its practicality more deeply. We are currently planning to conduct a large user study for infrastructure engineers on a cloud-sourcing platform. Thank you for your suggestion.

---

> ### Author Response · Authors · 2024-11-30
> **Gentle Reminder (Deadline: Dec. 2nd (AoE))**
>
> Dear reviewer jtMF
>
> We appreciate your feedback. Since the discussion period is closing soon (Dec. 2nd (AoE)), we know you are busy, but we would greatly appreciate it if you could take the time to respond to our response.
> In our response, we have described:
> - our problem differs from existing self-debugging in that it requires debugging code with self-goal setting, as well as addressing new types of topics and code (A1)
> - our paper already provides sufficient insights into the potentials and limitations of applying LLMs to CE, offering a sufficient basis to promote subsequent work in this new filed (A3)
>
> Additionally, the revised manuscript includes an additional case study of a larger system and the "complete rate" and "reconfiguration rate" obtained from multiple runs. We believe that this addresses W2.
>
> If you have time, please also take a look at our general response to catch up on the current state of our work.
>
> We are still looking forward to your response!
>
> Sincerely,
> Authors

---

> > ### Comment · Reviewer_jtMF · 2024-12-02
> > **Response to authors**
> >
> > Thank you for your gentle reminder. I generally appreciate this paper as it presents a new application for LLMs. However, my primary concern is the lack of novelty in the work.

---

> > > ### Author Response · Authors · 2024-12-02
> > > **Thank you for your response. Let us share our thoughts.**
> > >
> > > Dear reviewer jtMF
> > >
> > > We appreciate your response. As you pointed out, the individual techniques used in our system rely on existing methods, which makes the novelty, from a general research perspective (e.g., proposals for a new paradigm or the addition of components to baseline methods), somewhat limited.
> > >
> > > However, we believe that our novelty can be found in providing a reasonable combination of individual techniques that effectively solves a real-world problem in a new way, such as a new agent workflow for CE and the unit test-based validation for ensuring the consistency of LLM agents' steady-state validation. In fact, finding such a combination is non-trivial, and we have invested significant effort into it.
> > > Given that our proposed combination reduces the effort required by subsequent researchers to discover such combinations from scratch and provides a clear guide (limitations and future directions), we believe that our system deserves recognition as a novel contribution.
> > >
> > > What we shared above is our subjective opinion, so we would appreciate it if you could consider it as another perspective on novelty, just for your reference.
> > >
> > > Anyway, we really thank you for your constructive feedback and for sharing your concerns!
> > > If you have additional questions or concerns, please feel free to ask again!
> > >
> > > Sincerely,
> > > Authors

---

> > > > ### Comment · Reviewer_jtMF · 2024-12-02
> > > > **Response to authors**
> > > >
> > > > Thank you for your response. I'll increase my score to 5.

---

> ### Author Response · Authors · 2024-12-02
> **Thank you for your response**
>
> Dear reviewer jtMF
>
> Thank you for your response and for raising your score!
> We are really pleased that you have recognized our work more positively!
>
> We still take your important feedback seriously and will continue to improve our work accordingly.
>
> Sincerely,
> Authors

---

### Official Review · Reviewer_jkoh · 2024-11-03

**Soundness:** 3
**Presentation:** 3
**Contribution:** 3
**Rating:** 6
**Confidence:** 3

**Summary:**

The paper presents a framework for automating Chaos Engineering (CE) workflows using LLMs. The system, ChaosEater uses multi-step, multi-agent workflow to generate hypotheses, inject failures, run experiments, analyze output, suggest improvements and repeat until Validation as Code passes. Chaos Eater may automate CE in Kubernetes (K8s) environments managed with Infrastructure as Code (IaC).

**Architecture**: The system consists of five primary phases:
   - **Pre-processing**: Processes configuration dependencies.
   - **Hypothesis Definition**: Defines system resilience criteria (steady states) and specifies failure scenarios for testing.
   - **Experimentation**: Conducts chaos experiments by injecting pre-defined failures while validating steady states in real-time.
   - **Analysis**: Reviews the experiment results to assess whether the system meets the resilience hypothesis.
   - **Improvement**: If the hypothesis is not satisfied, reconfigures the system accordingly and repeats the experimentation phase.

**Strengths:**

1. **CE Automation**: ChaosEater attempts full automation of Chaos Engineering, covering hypothesis generation, fault injection, and iterative system improvement using LLMs and agentic architecture.
2. **Well Defined Architecture**: Authors clearly define the architecture of the system and LLMs/agents that work in each of the parts.
3. **Demonstrated Case Study**: System is demonstrated on a case study with a Kubernetes-managed Nginx server system.
4. **Cost and Time Efficiency**: The system would significantly reduce the time and costs associated with manual CE processes.
5. **Validation as Code (VaC)**: Provides a transparent and consistent method for validating system resilience.

**Weaknesses:**

### Weaknesses
1. **Demonstrated on Toy System Only**: System has been demonstrated on a toy system with a very simple failure. It is not known if system will work well on actual large systems. Could you show results of experiments on a benchmark set or on larger systems? Also see weakness 2.
2. **Challenge in Vulnerability Discovery**: The system may not be able to identify issues in already resilient systems, where fault discovery requires deeper analysis.
3. **Limited to Development Environments**: Currently operates only in development environments. Additionally, CHAOSEATER struggles to uncover vulnerabilities in systems that are already resilient.
4. **Limited to Configuration Improvements**: If I understand correctly, only K8s configuration scripts can be changed in the system improvement step. In other words, the system does not automatically change the tested system code. This is a significant limitation, although I understand that currently LLMs may not be able to automatically change the tested system code to improve resiliency.

**Questions:**

Figure 7 is unreadable while taking a whole page – please replace with something that serves better in presenting the system.

---

> ### Author Response · Authors · 2024-11-19
> **Response to Reviewer jkoh**
>
> Dear reviewer jkoh
>
> Thank you for your time, valuable suggestions, and important questions. We have answered your concerns in the following. We hope our answers address all your concerns. If you have remaining questions/concerns, please feel free to raise them for further discussion.
>
> ---
> > **W1**: System has been demonstrated on a toy system with a very simple failure. It is not known if system will work well on actual large systems. Could you show results of experiments on a benchmark set or on larger systems? Also see weakness 2.
>
> **Answer (A1)**
>
> Thank you for your suggestion. We are currently evaluating ChaosEater on the sock-shop server [1], which is a much lager, practical system consisting of 28 manifests (resources) and over 800 lines in total.  We will add the results to the revised manuscript, so please wait a little longer.
>
> ---
> > **W2**: The system may not be able to identify issues in already resilient systems, where fault discovery requires deeper analysis.
>
> > **W3**: Currently operates only in development environments. Additionally, CHAOSEATER struggles to uncover vulnerabilities in systems that are already resilient.
>
> **Answer (A2)**
>
> As you pointed out, we can deploy ChaosEater only in development environments, not in production environments. ChaosEater has a limited ability to discover vulnerabilities in mature systems through a CE cycle. As discussed in the ‘Discussion’ section, we plan to address these limitations in future work: we will improve the security aspect for the production deployment; we will improve some components and conduct long-term and multiple CE cycles to discover vulnerabilities in mature systems that already have good resiliency, including those that have not been identified by human engineers.
>
> ---
> > **W4**: If I understand correctly, only K8s configuration scripts can be changed in the system improvement step. In other words, the system does not automatically change the tested system code. This is a significant limitation, although I understand that currently LLMs may not be able to automatically change the tested system code to improve resiliency.
>
> **Clarification Notice**
>
> We are sorry, but we do not have 100% confidence in understanding the definition of the "tested system code" you mentioned. K8s manifests manage the backend of systems and correspond to system architectures themselves (i.e., Infrastructure as Code). Therefore,  as code related to the system other than K8s manifests, we imagined that "tested system code" refers to the code in the application layer of the system, such as frontend code (HTML/CSS/JS). In the following, we respond based on this assumption. If we are misunderstanding, we would appreciate it if you would provide its more detailed definition, so that we can respond in a way that meets your expectations.
>
> **Answer (A3)**
>
> As you pointed out, ChaosEater can change only the K8s manifests of a given system, and cannot change other types of code, such as frontend code (e.g., HTML/CSS/JS). This is because ChaosEater focuses on improving the resiliency of the system on the backend side, without changing the original content of the system application. The changes in the K8s manifests are restricted to those related to the backend's resiliency, so there is little possibility of these changes impacting the frontend code.
>
> On the other hand, the frontend code would affect the system resilience, and mutually improving both the frontend and backend sides would be important. For example, poor implementation of frontend code would lead to excessive resource consumption, causing issues in K8s resources. ChaosEater does currently not support such advanced mutual reconfiguration, but we would like to consider it as it is practically important and technically exciting. Personally, we believe that it would be necessary for the integration of other LLM-based systems for the web application creation with our system, which is mentioned briefly in ‘broader impacts’ section.
>
> ---
> > **Q1**: Figure 7 is unreadable while taking a whole page – please replace with something that serves better in presenting the system.
>
> **Answer (A4)**
>
> Thank you for your suggestion, and we are sorry for the inconvenience. We are revising that part to improve the presentation of the case study section, so please wait a little longer.
>
> ---
> [1] https://github.com/microservices-demo/microservices-demo

---

> > ### Comment · Reviewer_jkoh · 2024-11-23
> > **Thank you for your responses**
> >
> > Thank you for your responses.

---

> ### Author Response · Authors · 2024-11-30
> **Thank you for your response and further discussion**
>
> Dear reviewer jkoh
>
> Thank you for reading our response and for pointing out an important aspect in the discussion with the reviewer iWjm!
> We apologize for bothering you repeatedly, but we would like to inform you about the revised manuscript and the discussion summary. The manuscript includes an additional case study of a larger system and a summarized figure (the figure number has been changed from 7 to 6), which are related to your concerns/suggestions (W1 and Q1).
>
> If you have time, we would appreciate it if you could also see the revised manuscript and our general response. Of course, we are always open to any additional questions or feedback you may have!
>
> Sincerely,
> Authors

---

> > ### Comment · Reviewer_jkoh · 2024-12-03
> > **Thank you for manuscript revision**
> >
> > See my response in the general response section.
> >
> > I am going to maintain my score. Even though I would suggest to possibly accept this paper, I do not think I can champion it for acceptance.
> >
> > Thanks.

---

> ### Author Response · Authors · 2024-12-03
> **Thank you for your feedback again!**
>
> Dear reviewer jkoh
>
> We greatly appreciate your active discussion and constructive feedback!!! We are also really pleased that you have recognized our work in a positive way while understanding our weaknesses to be addressed in future work!
>
> We have addressed your feedback in the General Response, and we hope it helps your understanding.
>
> Although the deadline is approaching, we would still be happy to address any additional questions or concerns until the very last moment! In such cases, we will use the grace period for authors to provide a response.
>
> Sincerely,
> Authors

---

### Official Review · Reviewer_ujQS · 2024-11-04

**Soundness:** 3
**Presentation:** 3
**Contribution:** 3
**Rating:** 6
**Confidence:** 2

**Summary:**

The paper presents a framework for automated testing and improving Infrastructure-as-Code systems based on Kubernetes. It follows the Chaos Engineering (CE) approach, which observes how the system reacts to artificially injected deficiencies. Based on configuration files, manifests, etc., the framework automatically analyzes to identify the normal behavior, then modifies the configuration and checks whether the system behavior degrades significantly, and, if necessary, modifies the system configuration to make the system more resilient.

The approach essentially boils down to using scripts to connect existing CE tools and having LLMs figure out parameters and code modifications as required. I'm not an expert in CE, so I don't know how hard that is. I welcome that the authors did make an effort to insert verification and validation steps to ensure the soundness of the approach. All in all, this seems like a promising step in applying LLMs to an interesting problem in software engineering.

**Strengths:**

The approach essentially replaces a human CE engineer with a combination of scripts and LLM input to implement the required tests and modifications. The entire workflow is highly automated so that with enough computational resources, one could imagine the system finding and fixing a variety of problems all by itself. This might bring considerable cost reductions, at least under the assumption that compute costs will decline rapidly in the future (and LLM power will increase).

**Weaknesses:**

The paper says little about the exhaustiveness of the approach. Given infinite time, would the system be expected to find the majority of relevant faults? The case study seems relatively simple, and even in such an idealized case, two different solutions are proposed, and it does not seem clear which one is preferable. I can imagine that this problem of choosing between alternatives is much worse in more complex systems. The authors admit in the discussion that it is not trivial to scale this to challenging tasks, and I applaud their intellectual honesty.

**Questions:**

- How are thresholds calculated from the inspected values (margin for natural fluctuations…)?
- Why is the temperature of the LLM set to zero? Doesn't this limit the creativity (or exhaustiveness, depending on the time budget) in devising chaos experiments?
- How can you be sure that VaC scripts work as intended (e.g., rather than just always giving positive results)? For example, you could voluntarily use thresholds that are too low to check that they give negative results when necessary.
- In the case study, the analysis phase presented two solutions: changing the restart policy or using a Deployment resource. Why was the latter solution chosen in the improvement phase? What are the perceived upsides/downsides? How about the actual upsides/downsides of this solution? E.g., does introducing three replicas increase the overall cost of operating the system?

---

> ### Author Response · Authors · 2024-11-19
> **Response to Reviewer ujQS [1/3]**
>
> Dear reviewer ujQS
> Thank you for your time, valuable suggestions, and insightful questions. We have answered your concerns in the following. We hope our answers address all your concerns. If you have remaining concerns/questions, please feel free to raise them for further discussion.
>
> ---
> > **W1**: The paper says little about the exhaustiveness of the approach. Given infinite time, would the system be expected to find the majority of relevant faults?
>
> > **Q2**: “Why is the temperature of the LLM set to zero? Doesn't this limit the creativity (or exhaustiveness, depending on the time budget) in devising chaos experiments?”, “The paper says little about the exhaustiveness of the approach. Given infinite time, would the system be expected to find the majority of relevant faults?”
>
> **Answer (A1)**
>
> The reason why we set the temperature to zero is to improve the reproducibility of CE cycles conducted by LLM agents. Even when fixing the seed value with ```temperature=0```, it is not possible to completely reproduce the outputs from LLM agents (GPT-4o) every time. However, we found that the output patterns remain constrained within a certain range. Therefore, we adopt ```temperature=0``` to allow other researchers and engineers to approximately reproduce the results presented in our paper.
>
> On the other hand, as you pointed out, setting the temperature to zero would limit the diversity of proposed steady states and failure injections (i.e., hypothesis). To address both reproducibility and diversity simultaneously, we believe that controlling diversity through instructions is effective. Fortunately, ChaosEater can take user instructions along with a Skaffold project folder. When running multiple CE cycles, we can instruct ChaosEater to propose appropriate hypotheses that differ from those proposed in previous cycles. For example, the input instruction would be “In the previous CE cycle, you proposed a hypothesis with a steady state of A and a failure of B. For the next cycle, please propose a different hypothesis to explore various scenarios”. By doing so, the diversity of proposed hypotheses is expected to increase over multiple CE cycles\*. Moreover, rather than simply setting a higher temperature for random exploration, explicitly guiding ChaosEater to propose new hypotheses in this manner is a more efficient way to achieve sufficient exhaustiveness/diversity. Of course, this approach would work for both ```temperature=0``` and ```temperature>0``` settings.
>
> The paper focused on discussing a single cycle, so this topic was not mentioned in detail. However, as described above, we believe that the approach can be used to propose a wide range of Chaos experiments across multiple CE cycles. We are not sure that we can provide a deeper analysis of the exhaustiveness/diversity during this discussion period, but we will conduct it later to more deeply understand ChaosEater's behavior and effectiveness.
>
> \*As discussed in the ‘Discussion’ section, how to manage the history (i.e., previously defined hypothesis) of a large number of CE cycles and reduce duplication in the proposed hypotheses for each cycle is still unresolved.
>
> ---
> > **W2-1**: The case study seems relatively simple
>
> **Answer (A2)**
>
> We are currently evaluating ChaosEater on the sock-shop server [1], which is a much larger, practical system consisting of 28 manifests (resources) and over 800 lines in total. We will add the results to the revised manuscript, so please wait a little longer.
>
> [1] https://github.com/microservices-demo/microservices-demo

---

> ### Author Response · Authors · 2024-11-19
> **Response to Reviewer ujQS [2/3]**
>
> > **W2-2**: The case study seems relatively simple, and even in such an idealized case, two different solutions are proposed, and it does not seem clear which one is preferable.
>
> > **Q4**: In the case study, the analysis phase presented two solutions: changing the restart policy or using a Deployment resource. Why was the latter solution chosen in the improvement phase? What are the perceived upsides/downsides? How about the actual upsides/downsides of this solution? E.g., does introducing three replicas increase the overall cost of operating the system?
>
> **Answer (A3)**
>
> In the following, we answer your concerns and questions by separating two topics in order: the differences between Pod with ```restartPolicy=Always``` and Deployment with three replicas, and our insights on the decisions made by LLMs.
>
> As a premise, restarting a Pod typically takes several to tens of seconds (it depends on systems though). Therefore, a Pod with ```restartPolicy=Always``` will automatically restart when killed, but downtime occurs until the Pod is fully recovered. On the other hand, in a Deployment with three replicas, even if one of the Pods managed by the Deployment is killed, the remaining two Pods (replicas) can compensate for it, effectively eliminating downtime. In the case study, a steady state is defined as “the Pod being in the ```Running``` state for more than 95% of the monitoring period”.
> While a Pod with ```restartPolicy=Always``` might initially seem reasonable, it is difficult to satisfy this steady state when downtime is taken into account. More specifically, during the fault-injection phase of the chaos experiment, a 30-second monitoring is conducted. Therefore, to meet the 95% requirement, the Pod must be in the ```Running``` state for at least 28.5 seconds out of the 30 seconds. Furthermore, since monitoring is performed at 1-second intervals, the Pod effectively needs to be fully recovered within 1 second, which is difficult to achieve with a single Pod. Overall, while a Pod with ```restartPolicy=Always``` has the advantage of saving more resources, maintaining the steady state in this case is challenging without using a Deployment with three replicas.
>
> While the facts of each option are as described above, we cannot say that LLMs consistently apply the same logic to select the Deployment resource in similar cases. In fact, we also observed the case where the LLM agent selected the Pod with ```restartPolicy=Always``` instead of the Deployment, and the first reconfiguration failed. However, considering the errors from the first reconfiguration, the agent was eventually able to arrive at the Deployment with multiple replicas. Therefore, for your question “Why the LLM agent selected the Deployment in the case study?”, all we can say here is that a Deployment was chosen in the case study as a result of probabilistic selection. On the other hand, it has been confirmed that through trial and error in chaos experiments (i.e., the improvement loop), the conclusion eventually converges to a similar reconfiguration (the Deployment In this case). We believe that this process embodies the very Chaos Engineering, where the validity cannot be determined without actual execution, and ChaosEater successfully puts it into practice.
>
> ---
> > **Q1**: How are thresholds calculated from the inspected values (margin for natural fluctuations…)?
>
> **Answer (A4)**
>
> As you imagined, the threshold is defined as (current state value + tolerance), where the tolerance accounts for natural fluctuations. Note that we assume here that the current state value is the system's normal state.

---

> ### Author Response · Authors · 2024-11-19
> **Response to Reviewer ujQS [3/3]**
>
> > **Q3**: How can you be sure that VaC scripts work as intended (e.g., rather than just always giving positive results)? For example, you could voluntarily use thresholds that are too low to check that they give negative results when necessary.
>
> **Answer (A5)**
>
> Thank you for pointing out that important mechanism. Unfortunately, the current version of ChaosEater does not have a mechanism to verify whether the generated VaC scripts are non-trivial. Although it has a verification loop for VaC scripts, it just verifies whether VaC scripts pass (the threshold should be satisfied by the current state value).
>
> In the case studies (Nginx server+ sock-shop server (to be added later)), we did not observe such trivial VaC scripts that always pass. We consider that this is due to the design of our workflow and the benefits of the LLM's in-context learning capabilities. Regarding the former, the thresholds and the VaC scripts to validate them are generated by different LLM agents. The thresholds are determined by adding tolerance to the current state value. The VaC scripts are then generated by simply integrating the predefined thresholds into inspection scripts (which are generated in step 2 of the steady-state definition). Therefore, there is little room for the LLM agents to engage in cheating behavior and generate trivial VaC scripts that always pass. As for the latter, In-context learning, which dynamically adapts outputs to the input context of Chaos Engineering, makes such meaningless optimizations less likely. This is significantly different from the conventional reinforcement learning agents, which focus only on the reward without context and may exhibit cheating behavior if the reward design contains loopholes.
>
> On the other hand, we agree that such a mechanism is necessary to guarantee that LLMs are aligned to generate our intended VaC scripts. Implementing VaC scripts in a way that allows the threshold to be changed via command-line options and verifying if the pass/fail status switches at appropriate boundaries when the threshold value is shifted would be an excellent idea to verify that the scripts are non-trivial. We can probably not include the new mechanism during this discussion period, but we will certainly add it to the next version of ChaosEater. Thank you very much for pointing it out and for your suggestion.

---

> > ### Comment · Reviewer_ujQS · 2024-11-26
> >
> > I thank the authors for their detailed comments and explanations. I maintain my positive review of the paper.

---

> ### Author Response · Authors · 2024-11-30
> **Thank you for your response**
>
> Dear reviewer ujQS
>
> Thank you for reading our response and recognzing our work positively!
> We have revised the manuscript and summarized our discussions with the reviewers.
> The revised manuscript includes an additional case study of a larger system, which is related to your concerns (W2-1).
>
> If you have time, we would appreciate it if you could also see the revised manuscript and our general response.
> Of course, we are always open to any additional questions or feedback you may have!
>
> Sincerely,
> Authors

---

### Author Response · Authors · 2024-11-30
**General Response by Authors: Current Discussion Summary and Revised Manuscript [1/3]**

Dear all the reviewers

Thank all the reviewers for their valuable time and constructive feedback!
We have uploaded the final revised manuscript.
In the following, we summarize:

- Selected discussion with the reviewers so far
- Changes in the revised manuscript

We hope this summary helps the reviewers catch up on the current state of our work.
For details, please refer to the threads for each reviewer.
If you have additional questions or concerns, please feel free to ask at any time!

---
## Discussion

**D1. Only Small System Evaluated**

The reviewers ```ujQS```, ```jkoh```, and ```aLZw``` were concerned that Nginx studied in the case study is too simple to sufficiently demonstrate the effectiveness of ChaosEater. To address this, we added the case study of SockShop [1], which is a practical and large-scale e-commerce system consisting of 29 manifests. Compared to Nginx, the number of manifests is approximately 15 times, the code lines 35 times, and the tokens 40 times greater (see Table 3 in Appendix C for this statistic). Additionally, the intentional resiliency issue included in SockShop is "Single replica of Deployment resource (Deployment has already restarting mechanism)" and is more challenging.

The additional case study showed that even for the large-scale system with relatively redundant configurations, ChaosEater can identify the downtime issue of the single replica and improve the system by increasing the number of replicas. The time and monetary costs are 25 mins and USD 0.8. Intuitively, they are still significantly lower than the costs incurred by human engineers.

The reviewer ```jtMF``` suggested an additional metric obtained from multiple runs. Therefore, we also ran the case studies for Nginx and SockShop five times and calculated the "completion rate" and "reconfiguration rate". These results confirmed that ChaosEater can stably complete reasonable single CE cycles as mentioned above.

We believe that these additional case studies enhance the demonstration of ChaosEater's effectiveness.
Additionally, the successes in SockShop support that ChaosEater can effectively mitigate long-context issues, a concern raised by the reviewer ```iWjm```.

[1] https://github.com/microservices-demo/microservices-demo

---
**D2. Support for various types of failures and their modes**

The reviewer ```iWjm``` was concerned that supported failure types and failure modes are too limited in ChaosEater.
However, we believe that the current version of ChaosEater already covers most types of failures and failure modes.
ChaosEater supports all failure types of Chaos Mesh except for kernelChaos. There are 7 general failure types\*, such as PodChaos, NetworkChaos, and IOChaos, covering the majority of failures.
ChaosEater also supports complex failure modes by flexibly combining multiple Chaos Mesh failures sequentially or in parallel. Such complex failure combinations are realized by the LLM agent planning and our rule-based algorithm to convert the plan to a Chaos Mesh workflow manifest.

Please see our response to reviewer ```iWjm``` (A2, A3) for more details on this topic.

\* Including subtypes and detailed parameters, an enormous number of failure types can be generated.

---
**D3. Can Reconfiguring K8s Manifests Alone Address All Failures?**

Regarding Discussion 2, the reviewer ```jkoh``` pointed out an important point that reconfiguring K8s Manifests alone would not be sufficient to address all failure modes. Systems are constructed based on not only K8s manifests but also lower-layer settings (e.g., cluster settings) and application code (e.g., HTML/CSS/JS/Python). Therefore, the reviewer ```jkoh``` was concerned that even if ChaosEater can inject most types of failures, it would not be able to provide solutions for the cases that require modification on layers other than K8s manifests.

We agree that reconfiguring other layers than K8s manifests is necessary to improve the system resiliency in an optimal way.
On the other hand, we still believe that solely K8s manifest reconfiguration can somehow solve the majority of failures (even if it is not the optimal solution). K8s manifests manage system resource settings, and their settings can handle most failures by increasing the redundancy/resiliency and improving the error handling of the corresponding resources.

Therefore, the current version of ChaosEater supports only K8s manifest reconfiguration as the top priority feature. We believe that this is sufficient to improve the system’s resiliency in most cases. However, to optimize the system resiliency improvement, we plan to add the other layer reconfiguration to the next version.

---

> ### Author Response · Authors · 2024-11-30
> **General Response by Authors: Current Discussion Summary and Revised Manuscript [2/3]**
>
> **D4. Diversity of Proposed Hypotheses**
> The reviewers ```ujQS``` and ```aLZw``` were concerned about the diversity of proposed hypotheses (steady states + failures). With only a single CE cycle, as they pointed out, the proposed hypotheses may be also biased due to temperature settings or biases inherent to the LLMs. However, by instructing ChaosEater to propose hypotheses different from those proposed in previous cycles in each of the multiple CE cycles, it is possible to forcibly increase the diversity of proposed hypotheses and mitigate biases.
>
> To comprehensively evaluate the diversity and validity of hypotheses, it is necessary to construct datasets of various systems that covers a wide range of steady states and potential failures. It is also necessary to newly define evaluation methods, such as how to calculate the coverage rate for each system\*. Similar to Discussion 6, we plan to analyze the diversity of proposed hypotheses using newly constructed datasets in future work.
>
> \* For each system, the steady states and potential failures are somewhat limited (for example, IOChaos is irrelevant for an Nginx server without a database). However, this limited range is unknown in general. Therefore, calculating the coverage rate itself is a challenge that needs to be addressed.
>
> ---
> **D5. Support for Different LLMs**
>
> The reviewer ```aLZw``` was concerned about the robustness of ChaosEater (prompt templates) for different LLMs. We replaced GPT-4o with Claude Sonnet 3.5 and Gemini 1.5 pro and checked whether ChaosEater works correctly with these different LLMs. Unfortunately, the results reveal that ChaosEater encounters runtime errors in most cases (see ```casestudy_complete_dialogues/Nginx/ChaosEater_{claude, gemini}_nginx_N.pdf``` in the Supplementary Material). The runtime errors include JSON output format errors, debugging counts exceed in the verification loop of unit test, etc. We believe that this is due to manual prompt tuning, where we tune our prompts specifically for GPT-4o to prevent it from generating inappropriate responses.
>
> Specializing in a single LLM (GPT-4o) is an effective engineering strategy for reducing prompt management costs in complex LLM systems like ours. Therefore, the current version of ChaosEater focuses on GPT-4o, one of the SoTA models. However, from the user and research perspectives, it is also true that supporting a variety of LLMs is important. So, we plan to explore auto prompt tuning methods to optimize our prompts for each LLM using our current prompts as seeds.

---

> ### Author Response · Authors · 2024-11-30
> **General Response by Authors: Current Discussion Summary and Revised Manuscript [3/3]**
>
> **D6. Lack of Evaluation Frameworks**
>
> The reviewer ```jtMF``` suggested constructing evaluation frameworks, such as large datasets, new metrics, user studies, and baselines, as the foundation of this field. We understand its importance, but after considering the page limit and the importance of promptness, we have submitted this paper as a system paper to promote the application of LLMs to this new field.
>
> In fact, constructing datasets and metrics for CE requires as much effort as creating a dataset from scratch. Compared to other types of programming code, the number of open-source licensed K8s manifests for microservices is quite limited. Therefore, we should create K8s manifests for evaluation by ourselves using our time, cloud-sourcing, LLMs, etc. After collecting K8s manifests, we should intentionally introduce various resiliency issues into the K8s manifests to create a ground truth for whether the issue is addressed. As the quality of CE involves a somewhat philosophical perspective, we also believe that quantitative metrics based on that ground truth are not enough. Even if the issue is not resolved, a valid CE cycle also serves as a guarantee that "the system satisfies the hypothesis defined within it." Therefore, it is necessary to qualitatively evaluate the content of the CE cycle, regardless of whether the issue has been resolved. This qualitative evaluation requires conducting large-scale crowdsourcing-based user studies targeting K8s/CE engineers and building an LLM-as-a-judge system refined using actual data from such studies.
>
> As discussed above, constructing new evaluation frameworks for CE requires significant efforts. We first believe that such significant efforts must be presented in the main content. However, due to the page limit, presenting both system architecture and new datasets/metrics in a paper is impracticable. Additionally, even if we were to publish a paper focusing on the evaluation frameworks, it would take a significant amount of time.  Therefore, in the meantime, we aim to promote this new field by promptly demonstrating the potential of LLMs for CE through ChaosEater; ChaosEater significantly reduces time and monetary costs while completing reasonable single CE cycles for both small and large systems. Of course, additional features and more solid evaluation are required to further advance this field. However, we believe that this paper, as a first attempt, provides sufficient evidence of its effectiveness and current limitations, which could promote subsequent research in this new field, including improved systems and diverse benchmarks developed by other researchers and engineers.
>
> ---
> ## Revision
>
> In the revised manuscript, changes or additions are highlighted in red (except for typos and minor corrections). Changes or additions to entire sections or figures/tables are highlighted in red in their titles or captions. We thank all reviewers for suggesting the revisions!
>
> - **R1**: Regarding Discussion 1, we added the results of SockShop. We also replaced the values in Table 1 with ones averaged across five runs and added Table 2 for completion and reconfiguration rates.
> - **R2**: Regarding Discussion 5, we added the limitation for different LLMs. In the future directions, we added auto prompt tuning as a solution for this limitation.
> - **R3**: We added a higher-level monitoring system that monitors our system as an example of emergency measures in the future directions.
> - **R4**: We replaced the K8s analysis agent with an agent that directly identifies the weaknesses in the input K8s manifests. We found that the number of edges in the dependency graph becomes enormous for large-scale systems, and it is inefficient. As a result, we shifted our focus toward more direct methods of identifying weaknesses. We plan to reintegrate the K8s analysis agent into ChaosEater after implementing sub-graph extraction.
> - **R5**: We replaced the snapshots of ChaosEater with the highlighted outputs for Nginx and ShockShop to improve its presentation (Figure 6). The full dialogues are moved to separated PDFs. See ```casestudy_complete_dialogues``` in the Supplementary Material.
> - **R6**: We moved the Related Work section to Appendix A due to the page limit.
> - **R7**: We added our system prompt templates to Appendix B.
> - **R8**: We added full inputs and outputs to Appendix C.

---

> ### Comment · Reviewer_jkoh · 2024-12-03
> **Thank you for manuscript revision**
>
> I appreciate the work authors put into revising the manuscript and adding experiments with the SockShop application.
> It is great to see that the system worked on a larger example. However, as authors have observed, this required a change in ChaosEater where authors replaced K8s analysis agent with an agent that directly identifies the weaknesses in the input K8s manifests. On one hand, this change allows ChaosEater to deal with larger more realistic systems. On the other hand, we don't know if this approach limits the capabilities of the system.
>
> I remain somewhat concerned that ChaosEater can only find issues with K8s manifests. This seems limiting.
>
> Minor issue: Figure 6 is unreadable. It may be barely readable at 3x or more zoom, but that's not the expected clarity for figures in the manuscript. If the paper gets accepted, please replace Figure 6 with something that is illustrative and can be read at normal size.

---

> ### Author Response · Authors · 2024-12-03
> **Thank you for checking our revision**
>
> Dear reviewer jkoh
>
> Thank you for checking our revision and for pointing out important points again!
> In the following, we answered your questions and concerns by separating two topics.
>
> - The intention and potential impacts of the replacement of the K8s analysis agent
> - ChaosEater can only find issues with K8s manifests
>
> PS: Regarding Figure 6, we apologize for the inconvenience again.  We will certainly increase the size of the figures and text so that readers can easily read them without zooming.
>
> ---
>
> > The intention and potential impacts of the replacement of the K8s analysis agent
>
> Both agents share the same goal; they aim to generate an implicit context of K8s manifests to assist with proposing effective failure injections targeting the system’s weaknesses in the hypothesis phase and analyzing root causes in the analysis phase.
>
> The K8s analysis agent generates descriptions of dependencies between K8s resources based on the dependency graph produced by kubectl-graph. This description enables the proposal of failure injections exploiting the dependencies. In the analysis phase, by comparing the results of chaos experiments with the dependencies, it becomes possible to analyze the root causes of complex failures that propagate through those dependencies.
>
> On the other hand, the new agent that directly identifies the weaknesses literally generates a report of potential weaknesses in the input K8s manifests. The report enables the proposal of failure injections exploiting the weaknesses and appropriate root cause analysis by comparing the results of chaos experiments with the potential weaknesses.
>
> The K8s analysis agent appears to enable slightly more fine-grained failure injection and analysis. However, we found that for large-scale systems, such as SockShop, the length of dependency descriptions increases rapidly with the number of edges in the dependency graph. This raises concerns about promoting long context issues and inefficiency in terms of time and redundant context. On the other hand, the new agent summarizes only the information relevant to the original goal, so such concerns do not arise even for large-scale systems. Therefore, we have replaced the K8s analysis agent with the new agent, which is simple yet effective in achieving the original goal.
>
> However, removing the K8s analysis agent could introduce some limitations. For example, without explicit dependency descriptions, the LLM agents might struggle with proposing failures that exploit the dependencies and identifying root causes of complex failures that propagate through them. Such complex issues are beyond the current scope, but it is necessary to bring ChaosEater to a practical level. Therefore we plan to reintegrate the K8s analysis agent into the next version of ChaosEater after addressing the issue of the rapid increase in dependency descriptions. We believe that a solution would be subgraph(edge)-extraction, where we use the proposed failure scenario or the results of chaos experiments as queries to retrieve the relevant parts of the dependency descriptions (i.e., edges) from the entire set. These retrieved parts are then embedded into the agent’s input context. This would enable explicitly presenting the relevant dependencies to the agent while mitigating the long-context issues (bad time efficiency is unavoidable). This discussion is related to the fifth future direction in our manuscript.
>
> ---
>
> > ChaosEater can only find issues with K8s manifests.
>
> As we discussed before, ChaosEater currently supports only K8s manifest reconfiguration. In other words, ChaosEater is limited to finding issues within K8s manifests. Personally, we still believe that K8s manifests reflect most aspects of the backend, and reconfiguring them alone can somehow handle most failure scenarios. However, it is also true that reconfiguration for other types of code is necessary to improve the system’s resiliency in the optimal way and cover all failure scenarios. Therefore, as you pointed out, this becomes one of the limitations of the current ChaosEater (we will include this topic in the limitation section later).
>
> However, we believe that ChaosEater can support the other code types without major updates. Since front-end, application code, etc. can be input in the same way as K8s manifests, the remaining change is to make slight adjustments to the system prompt templates to accommodate code other than K8s manifests. At this moment, we cannot provide concrete evidence that it can be easily achieved, and new challenges might arise during implementation. However, we remain confident that it is at least feasible. Therefore, based on your feedback, we plan to expand the range of supported code types in the next version of ChaosEater.

---

### Author Response · Authors · 2024-12-04
**Final Comment by Authors**

Dear AC and Reviewers

We greatly appreciate all the reviewers for their valuable time and constructive feedback.
The feedback helps us not only improve our manuscript and demonstration but also identify our system’s limitations to be addressed in future work.

Our current system still has room for improvement, including support for various LLMs and code types, more comprehensive analysis, and enhanced diversity of hypotheses through multiple CE cycles.
However, we believe our work has successfully established an initial foundation by presenting a new system architecture that can fully automate CE cycles for both small- and large-scale systems at low cost. This non-trivial architecture design, along with the potential, limitations, and future directions suggested through case studies, provides sufficient guidance for subsequent works in the new field of LLM applications. This field is expected to become increasingly important not only for infrastructure but also for ML communities.

Finally, we would like to leave a comment to once again emphasize how our contributions are related to the ML community.
We hope these additional comments and our discussion summary serve as helpful references for your final decision.
We sincerely thank the AC and reviewers in advance for your valuable time and additional feedback during the final decision period.

Sincerely,
Authors


---

### Relation between our contributions and ML community
This paper proposed an LLM-based system that fully automates CE cycles, thereby reducing the time and monetary costs of CE. We believe our system innovates the conventional operations in the infrastructure of software-based systems, enabling anyone to build highly resilient systems at low cost.

On the other hand, why does it matter to the ML community? The answer is related to the broader impact section. In the ML community, the automatic generation of software applications using LLMs has been actively explored. However, few consider their infrastructure and resiliency, which are crucial for building practical services. Our proposed system addresses this gap. Therefore, we believe that our contributions to the ML community lie in demonstrating the importance of the applications’ infrastructure and suggesting that even improving its resiliency can be automated using LLMs.

Additionally, the automation of software engineering (SE) using LLMs has been actively studied. Since CE can be regarded as SE, our work is considered a part of this trend. Unlike existing benchmarks for solving GitHub issues, CE requires raising issues independently and resolving them. We believe our work suggests the existence of such a new and challenging SE task and demonstrates the potential of LLMs in addressing this task to the ML community.

Therefore, we believe our contributions are meaningful to both (software-based) infrastructure and ML communities.

---

### Meta-Review · Area_Chair_HW65 · 2024-12-26

**Metareview:**

This paper concerns automating Chaos Engineering (CE) workflows, and proposes an LLM-based framework ChaosEater. ChaosEater uses a multi-agent 6-phase workflow ranging from hypothesis generation, failure injection to improvement suggestion and validation as code. ChaosEater is specifically designed to automate CE in Kubernetes (K8s) environments. The problem of using LLMs to automate CE workflows is novel and interesting. The main concerns shared by the reviewers and the AC are twofold. First, whether sufficient contributions have been made to the machine learning community; the paper is written in simply describing a workflow of applying LLMs for automate CE in K8s, which certainly has values for CE community but is less clear to the ML community. CE workflows may serve as a valuable benchmark for LLMs, however, the current work is more like a system work rather than setting up a benchmark. Second, whether the evaluation of using a toy example (i.e., a simple Nginx-server system that consists of two K8s manifests) provides convincing conclusions.

**Additional Comments On Reviewer Discussion:**

There were active discussions between the authors and reviewers during the rebuttal. Given Chaos Engineering is a less familiar topic to the ML community, the authors made significant effort in elaborating and clarifying confusions raised by reviewers, which are much appreciated.

---

### Decision · Program_Chairs · 2025-01-22

Reject